# A Black Hole Eddy Dataset of North Pacific Ocean Based on Satellite Altimetry

Fenglin Tian[1, 2], Yingying Zhao[1], Lan Qin[1], Shuang Long[3], Ge Chen[1, 2, *]

[1] State Key Laboratory of Physical Oceanography, College of Marine Technology, Ocean University of China, Qingdao, 266100, China
[2] Laboratory for regional Oceanography and Numerical Modelling, Department of Ocean Big Data and Prediction, Laoshan Laboratory, Qingdao, 266237, China
[3] Guangxi Key Laboratory of Beibu Gulf Marine Resources, Environment and Sustainable Development, Fourth Institute of Oceanography, Ministry of Natural Resources, 536000, China

*Correspondence to*: Ge Chen (gechen@ouc.edu.cn)

**Abstract.** The methodologies employed for the identification of coherent ocean eddies can be categorized as either Eulerian and Lagrangian. Among Lagrangian structures, Black Hole Eddies (BHEs) exhibit the highest degree of material coherence and conservation, making them particularly suitable for studying the transport and retention of oceanic materials. This study presents an efficient Graphics Processing Unit (GPU) -based BHE identification algorithm for the North Pacific (0-50°N, 130°-270°E), enhancing computational efficiency by approximately 13 times compared to the existing methods. Using this algorithm, a Black Hole Eddy dataset (BHE v1.0) is constructed for the first time in the North Pacific Ocean, based on satellite-derived surface geostrophic velocity data from January 1, 1993 to May 5, 2023 (Tian et al., 2025a). BHE v1.0 contains 18387 eddies with radius larger than 20 km and lifetimes longer than 4 weeks and captures both the spatial-temporal characteristics and the trajectories of coherent eddies throughout lifetimes. Through the advection of Lagrangian particles in Eulerian eddy, rotationally coherent Lagrangian vortices (RCLVs) and BHEs, it is confirmed that BHEs retain material coherence throughout their lifecycle without significant mixing. Additionally, approximately 6% of BHEs, which do not overlap with any RCLVs or Eulerian eddies, are identified and referred to as the Naked Black Hole Eddy and further analyze its coherence through advection. Transport analysis shows that BHEs induce westward transport of about 1.5 Sv, three times weaker than the transport attributed to the broader RCLVs, suggesting it likely represents a more accurate estimate of the coherent, non-leaky component of oceanic transport, as BHEs isolate the materially coherent core of the eddies. These findings address the existing gap in Black Hole Eddy datasets within the field of oceanography and provides a novel perspective for studying the interactions between coherent eddies and oceanic physical phenomena.

## 1 Introduction

Mesoscale eddies, one of the most prominent processes in the ocean, typically range from several tens to hundreds of kilometers in spatial scale (Chelton et al., 2011a; Chaigneau et al., 2008). In the North Pacific, mesoscale eddies generally persist for about 4 weeks to one year, with longer-lived structures occasionally exceeding this range, particularly in energetic

regions such as the Kuroshio Extension (Chelton et al., 2011a; Chen and Han, 2019). These eddies play a crucial role in the transport of heat, salinity, and other various variables, further influencing the marine material cycle, large-scale water body transport, and biological activities (Chen et al., 2012; Chen et al., 2021; Faghmous et al., 2015; Dong et al., 2014). In recent

years, mesoscale eddies have become a major focus of research within the field of oceanography(Chelton et al., 2011b).

Existing mesoscale eddy identification algorithms are primarily categorized into Eulerian and Lagrangian methods. The Eulerian approaches are based on the instantaneous state of the flow field and relies on a fixed reference coordinate system (Serra and Haller, 2016), which can be categorized into three major categories: (1) extraction methods that utilize physical parameters, including the Okubo-Weiss (OW) parameter method (Chelton et al., 2007; Henson and Thomas, 2008), and the

Winding-Angle (WA) method (Chaigneau et al., 2008; Sadarjoen and Post, 2000); (2) extraction methods that are based on the direction of ocean currents (Nencioli et al., 2010; Williams et al., 2011); and (3) extraction methods based on sea surface height (SSH) (Chelton et al., 2011b; Faghmous et al., 2015; Mason et al., 2014). However, the Eulerian methods are frame-dependent and may yield inconsistent results under coordinate transformations, as the ocean flow lacks a fixed reference frame. Moreover, due to the transient nature of ocean currents, water parcels within Eulerian-defined boundaries can quickly

stretch into filaments and disperse, leading to a possible overestimation of eddy transport.

The Lagrangian methods are proposed based on the accumulation of fluid states within a flow field over a specified duration (Haller, 2015). In contrast to the Eulerian approaches, the Lagrangian methods conceptualize eddies as coherent structures within the oceanic environment. Due to the boundaries' ability to maintain stability over a finite time (Provenzale, 1999), Lagrangian eddies are demonstrated to exhibit material coherence. Therefore, Lagrangian eddies provide more accurate

representations of structures of oceanic mesoscale eddies, particularly within unstable flow fields. Furthermore, Lagrangian eddies are effective in quantifying the material transport capacity of these eddies (Tian et al., 2019; Tian et al., 2022).

Before Lagrangian coherent structures (LCSs) were rigorously defined, heuristic criteria methods such as the Finite-Time Lyapunov Exponent (FTLE) (Shadden et al., 2005; Haller, 2011) and the Finite-Size Lyapunov Exponent (FSLE) (D'ovidio et al., 2004; Joseph and Legras, 2002; Aurell et al., 1997) have been proposed to identify coherent structure. But these

heuristic methods lack material invariance, depend sensitively on thresholds and resolution, and are inadequate for capturing coherent structures in chaotic or finite-time dynamical systems (Beron-Vera et al., 2010; Beron-Vera et al., 2012; Haller, 2015; Peacock and Haller, 2013; Onu et al., 2015). To address these limitations, Haller and Beron-Vera (2012) introduced a geodesic framework in which elliptic LCSs are defined as closed material curves that minimize stretching, corresponding to closed geodesics of a Riemannian metric derived from the flow's Cauchy-Green strain tensor. Subsequently, Haller and

Beron-Vera (2013) introduced a rigorous variational framework for detecting coherent Lagrangian vortices, which are defined as closed material loops that remain minimally stretched and filamented over time. Remarkably, the mathematical formulation of these vortex boundaries is analogous to that of photon spheres in general relativity. In this analogy, fluid particles that cross the boundary remain trapped within the eddy's interior, much like light trapped inside a black hole. This striking similarity has led to the term "Black Hole Eddy (BHE)" being used to describe such coherent Lagrangian vortices..

Building on this theoretical foundation, Karrasch et al. (2015) developed an automated numerical algorithm that detects such

vortex boundaries by computing the Cauchy-Green strain tensor from the flow map and extracting closed tensorlines aligned with its eigenvector fields. This approach enables the systematic and frame-independent identification of elliptic LCSs, facilitating practical analysis of coherent vortices in real oceanic flows. Later, Serra and Haller (2017) presented a simplified and efficient algorithm for computing closed null geodesics that define coherent vortex boundaries in two-dimensional flows,

enabling fully automated and robust BHE boundary detection. Using the extraction method of BHE, Beron-Vera et al. (2013) extracted and analyzed the boundaries of the Agulhas eddy, the body of water enveloped by the Lagrangian eddy can continuously maintain independent rotation, and provided a coherent description of its boundary changes. Haller et al. (2018) subsequently introduced the automatically extracted eddy boundary into all randomly diffusing flow fields, considering this eddy boundary as a material barrier for diffusion and random transport. This enhances the objectivity of eddy boundary

identification, representing a significant theoretical advance over previous heuristic Lagrangian methods. And the process of solving BHE is complex (Karrasch and Schilling, 2020), but they still maintain superior coherence in the long-term transportation of flow fields. However, the practical implementation of these algorithms remains computationally intensive. Compared to Lagrangian approaches, Eulerian eddy datasets were developed earlier on a global scale due to the relative simplicity of their detection methods. In recent years, the number of published Eulerian eddy datasets has grown

significantly, driven by the increasing availability of satellite altimeter products and the growing interest in mesoscale ocean dynamics. Using 16 years of SSH data constructed by merging the measurements from two simultaneously operating altimeters, Chelton et al. (2011b) employed the closed contour method to generate the first global eddy dataset and analyze the general features of mesoscale eddies. Mason et al. (2014) proposed an open-source mesoscale eddy identification and automatic tracking algorithm, known as py-eddy-tracker (PET). Based on sea level anomaly (SLA), PET effectively tracks

eddy trajectories within user-defined regions. The method introduces key innovations by utilizing interpolated SLA contours, incorporating rigorous shape tests, and applying a speed-based criterion to define eddy boundaries and centers, thereby enhancing tracking accuracy and consistency compared to CSS11. Later, based on the most recent version of postprocessed daily SLA estimates that better resolves mesoscale features than the version used by CSS11, Faghmous et al. (2015) presented a global daily mesoscale ocean eddy dataset that contains ~45 million mesoscale eddies and 3.3 million eddy

trajectories. Based on CSS11 code, the global Mesoscale Eddy Trajectory Atlases (META 2.0) was produced by SSALTO/DUACS and distributed by Archiving, Validation and Interpretation of Satellite Oceanographic (AVISO), with support from the National Centre for Space Studies, in collaboration with Oregon State University and support from the National Aeronautics and Space Administration. Furthermore, Tian et al. (2020) developed a parallelized identification and tracking framework for global mesoscale eddies based on merged multi-satellite altimeter data. The approach employs

refined contour-based boundary extraction and introduces a synthetic "fake eddy" mechanism to accommodate short-term surface signal loss, thereby enhancing the temporal continuity and robustness of eddy trajectories. In order to better represent the dynamics in the more energetic oceanic regions and in the vicinity of coasts and islands, based on the absolute dynamic topography (ADT) field, instead of the previous SLA maps, Pegliasco et al. (2022) presented META3.1exp, which consists

of eddy identifications and trajectories derived from altimetric maps. The detection methodology utilized is based on the

PET algorithm developed by Mason et al. (2014).

In contrast to the earlier development of Eulerian eddy datasets, the advancement of Lagrangian eddy datasets has been relatively limited due to methodological and computational challenges. Tian et al. (2022) employed the orthogonal parallel architecture algorithm to identify and release a global Lagrangian eddy dataset, achieving a significant enhancement in identification efficiency, with a computational speed increase of up to 500 times, and using advection particles found that

lagrangian eddies have better material coherence and objectivity than Eulerian eddies. Later, Liu and Abernathey (2023) employed the Lagrangian-averaged vorticity deviation (LAVD) method (Haller et al., 2016) to generate a global Lagrangian eddy dataset GLED v1.0 based on altimetry observations. This dataset not only delineates the general characteristics of eddies with T=30-day and 90-day, such as the position of the eddy center, equivalent radius, and rotational properties, but also includes the particle trajectories delineated by the coherent eddy boundaries throughout their entire life cycle. Utilizing

the LAVD method, Xia et al. (2022) detected nine long-lived Agulhas rings and categorized their life cycles into two distinct stages: growth and extinction, which introduced the concept of a material belt and investigated the factors contributing to the longevity of Agulhas rings by establishing a linear relationship between the average strain rate and the square root of kinetic energy. To elucidate biogeochemical developments within eddies, Jones-Kellett and Follows (2024) used a backward-in-time Lagrangian particle tracking approach to identify rotationally coherent Lagrangian vortices (RCLVs) with a 32-day

integration window and 8-day reinitialization intervals. Each vortex was tracked across its lifespan by following the trajectory of its core particle and linking coherent boundaries across consecutive time steps. This method highlights the material retention capacity of Lagrangian eddies, aligning more closely with biogeochemical responses that depend on the water mass. Despite the increasing comprehensiveness of eddy datasets, the majority are constructed under an Eulerian framework. In contrast, the Lagrangian framework faces challenges such as algorithmic complexity, low computational

efficiency, resulting in a scarcity of global datasets pertaining to Lagrangian eddies. Furthermore, there is a notable lack of long-term Lagrangian datasets for specific marine regions, with most being primarily identified through the LAVD method. Consequently, long-term, and large-scale dataset products concerning BHE remain absent. Due to the lack of global or regional BHE data products, there remains a lack of analytical research on the material coherence, formation reasons, longevity, energy, and other characteristics.

Besides, coherent structures that persist for longer durations tend to be fewer in number and smaller in scale. (Abernathey and Haller, 2018; Jones-Kellett and Follows, 2024; Xia et al., 2022; Liu and Abernathey, 2023). However, Jones-Kellett and Follows (2024) compared the atlas of Lagrangian coherent eddies with an atlas of Eulerian eddies and find that RCLVs may not overlap with Eulerian eddies. However, it is unclear that whether the BHE is wrapped by the RCLVs and the Eulerian Eddies. Furthermore, does the RCLVs and Eulerian identification method fail to identify some coherent vortices? Comparing

BHE dataset to RCLVs and Eulerian eddy atlases can answer these questions, revealing how the coherent properties of mesoscale features manifest in space and time (Jones-Kellett and Follows, 2024).

This study presents an efficient GPU-accelerated identification algorithm designed to enhance the geodesic algorithm proposed by Serra and Haller (2017) for identifying BHE. This advancement represents a significant improvement, achieving speeds that are 13 times faster than the original algorithm in the Lagrangian approach to Black Hole Eddy identification. Utilizing this efficient GPU-accelerated identification algorithm, a dataset of Black Hole Eddy in the North Pacific (0-50°N, 130°-270°E) is constructed for the first time spanning from January 1, 1993 to May 5, 2023, which addresses a notable gap in oceanography, as such a dataset previously did not exist. The North Pacific is selected as the study region due to its high eddy activity (Chelton et al., 2011b), extensive altimetric coverage favorable for coherent eddy detection (Chaigneau et al., 2008), and the crucial role of mesoscale eddies in modulating regional oceanic heat transport and biogeochemical dynamics (Qiu and Chen, 2005). Based on this dataset, we compare the Eulerian eddies and RCLVs to analyze the strong coherence of BHE and find the Naked Black Hole Eddy which neither overlap with Eulerian eddies nor with RCLVs. Additionally, we provide fundamental characteristics of BHE, including their lifespan, size, and geographic distribution. This represents first comprehensive analysis of BHE.

The organization of this article is structured as follows: Section 2 introduces the methodology for generating the North Pacific Black Hole Eddy dataset. Section 3 presents essential information about the dataset, including the statistical characteristics of the BHE, as well as comparisons with Eulerian eddies and RCLVs. Section 4 discusses the usability of the eddy dataset. Finally, Section 5 offers a summary and conclusion.

## 2 Data and methods

### 2.1 Data

#### 2.1.1 Satellite altimetry data

This study utilizes the Level 4 global ocean gridded sea level reprocessed (delayed-time) product, version SEALEVEL_GLO_PHY_L4_REP_008_047, which was distributed by the Copernicus Marine Environment Monitoring Service (CMEMS) at the time of our analysis. This dataset provided data on a 0.25° × 0.25° global grid and was generated by the DUACS multi-mission altimeter data processing system. It merges measurements from all available satellite altimeter missions (e.g., TOPEX/Poseidon, Jason series, Sentinel series) through optimal interpolation. For our analysis, we used the ADT and the associated absolute geostrophic velocity fields, covering the period from 1 January 1993 to 5 May 2023. It should be noted that this specific 1/4° resolution product has since been decommissioned by CMEMS and superseded by a higher-resolution 1/8° version (SEALEVEL_GLO_PHY_L4_MY_008_047).

#### 2.1.2 Chlorophyll a data

Chlorophyll *a* (Chla) data from the Ocean Colour Climate Change Initiative (OC_CCI, Ocean Colour Climate Change Initiative) project of the European Space Agency (ESA) is designed to provide high-quality long-term ocean water color data

products for climate change studies fusing data from multiple satellite sensors, such as SeaWiFS, MODIS and MERIS, through band-shifting and bias-correction techniques, and using global measured data for error correction. It includes d datasets with daily to monthly averages, a spatial resolution of $0.25° \times 0.25°$ and a time series covering from September 1997 to the present. The temporal resolution chosen for this paper is 1-day, and the data can be downloaded at: https://www.pml.ac.uk/science/Earth-Observation-Science-and-Applications.

### 2.1.3 Sea surface temperature data

Sea surface temperature (SST) data used in this study are obtained from the National Oceanic and Atmospheric Administration (NOAA) Physical Sciences Laboratory (PSL), specifically from the NOAA Optimum Interpolation Sea Surface Temperature version 2 (NOAA OISST v2) dataset. This product is generated by blending satellite observations (e.g., AVHRR), ship-based measurements, and buoy data using optimal interpolation techniques. The anomaly fields are calculated relative to a long-term climatology, and reflect deviations in SST associated with phenomena such as eddies, El Niño events, or seasonal variability. The data are provided in NetCDF format, with a global spatial resolution of $0.25° \times 0.25°$ and temporal resolution of daily. The dataset can be accessed and downloaded from the NOAA PSL website: https://www.psl.noaa.gov/data/gridded/data.noaa.oisst.v2.highres.html.

### 2.1.4 Sea surface salinity data

Sea surface salinity (SSS) data used in this study are derived from the Global Ocean Ensemble Physics Reanalysis product (GLOBAL_MULTIYEAR_PHY_ENS_001_031), provided by the CMEMS. This ensemble reanalysis product offers global gridded estimates of key ocean physical variables, including temperature, salinity, ocean currents, and ice parameters, with a spatial resolution of $0.25° \times 0.25°$. The dataset spans from January 1, 1993 to December 31, 2023, and provides both daily and monthly averages over 75 vertical depth levels. This product is constructed by combining four different global ocean reanalysis systems: GLORYS2V4 (Mercator Ocean, France), ORAS5 (ECMWF), GloSea5 (UK Met Office), and C-GLORSv7 (CMCC, Italy). Each of these models assimilates satellite altimetry and in situ observations to reconstruct the ocean's physical state. The ensemble approach helps to quantify uncertainty and provide more robust estimates than individual reanalysis products. The dataset is accessible via CMEMS at:https://data.marine.copernicus.eu/product/GLOBAL_MULTIYEAR_PHY_ENS_001_031/description.

### 2.1.5 Mesoscale eddy data

The mesoscale Eulerian eddy dataset used in this study is the META3.1exp product developed by Pegliasco et al. (2022). META3.1exp provides a comprehensive catalog of mesoscale eddies, including their geographic locations, lifetime, propagation paths, amplitude, radius, polarity (cyclonic or anticyclonic), and other physical characteristics. The eddies are identified and tracked from gridded ADT fields derived from satellite altimetry maps distributed by AVISO (Archiving, Validation and Interpretation of Satellite Oceanographic data). The dataset covers the global ocean between 1993 and 2020

and is based on daily ADT products with a horizontal resolution of 0.25° × 0.25°. Eddy detection and tracking are performed using an objective algorithm based on the closed-contour method combined with criteria such as amplitude and lifespan thresholds to ensure robust eddy identification. The dataset can be accessed from the AVISO+ website at: https://data.aviso.altimetry.fr/aviso-gateway/data/META3.1exp_DT/.

For a comparative analysis, this study utilizes the global RCLVs dataset from Tian et al. (2022). We specifically selected their data product generated with a 30-day integration time (T=30) and a 7-day time step, resulting in a weekly dataset that is directly comparable to our BHE analysis. Unlike traditional Eulerian eddy detection methods, this Lagrangian approach ensures material coherence by identifying vortices that maintain their structure and boundary over time, based on objective criteria derived from finite-time rotation. The RCLVs dataset is constructed from satellite-derived daily SLA fields with a 0.25°×0.25° resolution, covering the global ocean from 1993-2020. The algorithm efficiently tracks eddies over time, providing information on eddy boundaries, lifetime, polarity, and coherent transport properties. The dataset can be accessed from website at: https://data.casearth.cn/dataset/63369940819aec34df2674d7.

In this study, the Eulerian eddy dataset and the RCLVs dataset are used to compare the coherence with BHE through the advection of Lagrangian particles in them and their co-occurrence.

## 2.2 Methods

### 2.2.1 Identification of Black Hole Eddy

Black Hole Eddies are characterized by boundaries that undergo minimal deformation during advection, maintaining a compact shape without generating filamentous structures (Haller and Beron-Vera, 2013). This minimal deformation reflects their material coherence and serves as a key criterion for their identification. However, the resolution process is complex, resource-intensive, and inefficient, rendering it unsuitable for identifying eddies on a larger scale or over extended periods.

Serra and Haller (2017) presented a simplified and efficient algorithm for computing closed null geodesics that define coherent vortex boundaries in two-dimensional flows. Unlike earlier methods (Karrasch et al., 2015) that relies on direction vector field integration, singularity detection, and user-specified Poincaré sections, the proposed approach formulates a unified initial value problem three-dimensional state space (two special coordinates and an angle variable)that eliminates sensitivity to tensor field singularities and removes the need for manual input, enabling fully automated and robust vortex boundary detection (Serra and Haller, 2017). However, the algorithm's high computational demands and time-consuming process have hindered the production of a dataset of BHE. Our research presents an efficient automated extraction algorithm for Lagrangian eddies based on null geodesics for the North Pacific. This method enhances and accelerates the existing null geodesics technique, thereby enabling the creation of a dataset of BHE in the North Pacific. In our experiment, the grid of initial positions ranges from 130° East longitude to 90° West longitude and from 0° to 50° North latitude.

This research harnessed the powerful parallel processing capabilities of Graphics Processing Unit (GPU) to expedite and optimize the identification of BHE. We consider eddy detection to be a demanding data processing challenge. To accelerate

recurring, large-scale computations, we partitioned them into numerous smaller GPU computational tasks. We generated Compute Unified Device Architecture (CUDA) code suitable for execution on NVIDIA GPU, thereby transferring the computational tasks to the GPU. This approach leverages the GPU's parallel processing capabilities for large-scale task execution. In this paper, CUDA was used as a multithreaded toolkit for GPU parallel computing. Figure 1 illustrates the workflow of the algorithm.

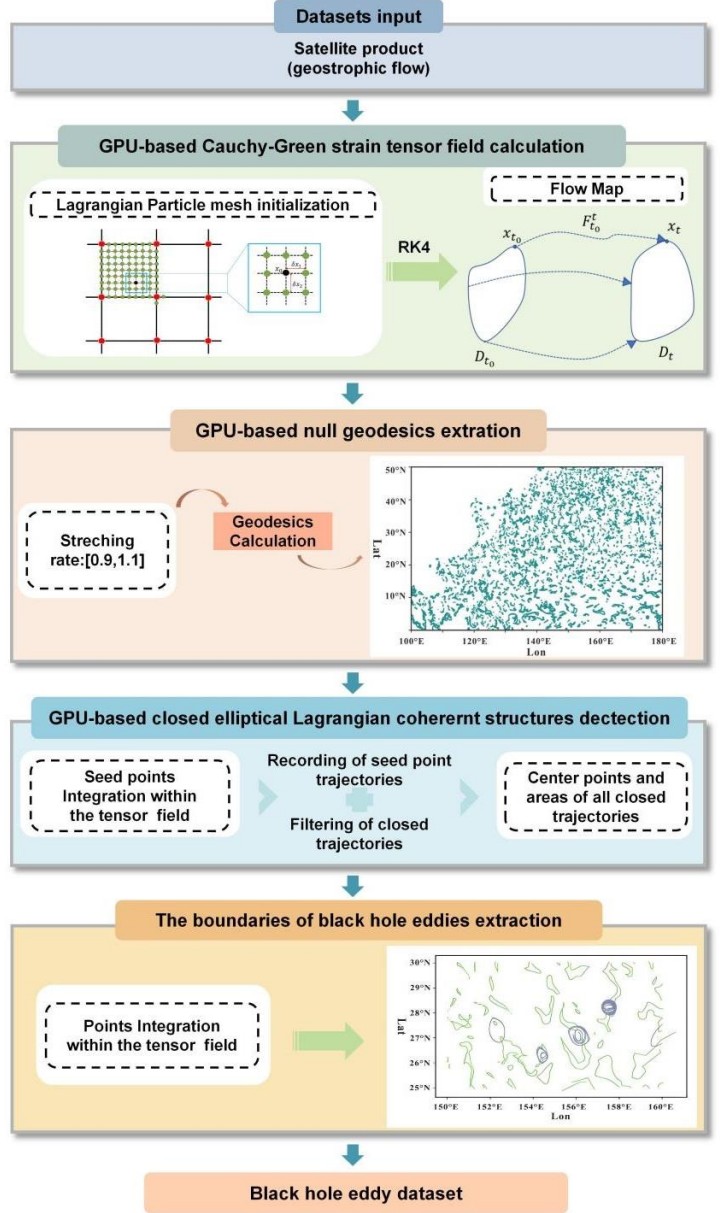

**Figure 1. Flowchart of Black Hole Eddy (BHE) dataset generation based on satellite altimetry.**

1. GPU-based Cauchy-Green strain tensor field calculation

Within the Lagrangian framework, we can express a two-dimensional unsteady flow field as follows:

$$\dot{x} = v(x,t), x \in U, t \in [t_0, t_1] \tag{1}$$

Here, $v(x,t)$ denotes the velocity field at position x and time $t$, $U$ signifies the boundary range of the flow field area, and the time interval $[t_0, t_1]$ represents the variation range of time $t$.

The flow map (Haller and Beron-Vera, 2013) can be elucidated as the trajectory of a fluid particle moving from its initial position $x_0$ at the initial time $t_0$ to its position $x$ at time $t$:

$$F_{t_0}^t(x_0) := x(t; t_0, x_0) \tag{2}$$

We frequently utilize the Cauchy-Green strain tensor field to depict the forces exerted on Lagrangian particles in motion. The tensor field is expressed as follows:

$$C_{t_0}^{t_1}(x_0) = \left[\nabla F_{t_0}^{t_1}(x_0)\right]^T \nabla F_{t_0}^{t_1}(x_0) \tag{3}$$

In this context, $\nabla$ symbolizes the Jacobian matrix of partial derivatives. The Cauchy-Green strain tensor, derived from the preceding computation, is positively definite. $T$ denotes a forward integration time to qualify the coherence of BHE. In this study A BHE defined with $T = 30 \ (or \ T = 90)$ is a material boundary identified at a start time $t_0$ that traps all fluid particles within it for subsequent 30 (90) days. At any given point $x_0$, there will be two eigenvalues $\lambda_1$ and $\lambda_2$, along with two corresponding orthogonal eigenvectors $\xi_1$ and $\xi_2$. These are expressed as follows:

$$C_{t_0}^{t_0+T}(x_0)\xi_i(x_0) = \lambda_i(x_0)\xi_i(x_0), |\xi_i(x_0)| = 1, i = 1,2, 0 < \lambda_1(x_0) \le \lambda_2(x_0) \tag{4}$$

In executing numerical simulations of BHE, our standard approach deploys a homogeneous grid structure throughout the flow field area, with the Cauchy-Green strain tensor computed at these grid nodes. As illustrated in Fig. 2, we enhanced the accuracy of the simulation by densifying the original velocity field grid with a resolution of 1/4° by a factor of 8. The figure shows red solid nodes in the center, representing the main grid points. These are the input velocity field data points and serve as the primary computation locations for the Cauchy-Green strain tensor field. The green points surrounding them are the auxiliary nodes post-densification, and the black points correspond to the auxiliary grid point $x_0$. Following densification, we calculate the Cauchy-Green strain tensor for both the main grid points and auxiliary nodes. We set the distance between the grid points $\delta_{x_1}$ and $\delta_{x_2}$ to be 1/8 of the main grid point distance in this study, which is roughly equivalent to 1/32° when converted into degrees. This detailed grid configuration facilitates a more profound understanding and enhanced accuracy in simulating the dynamics of BHE. The spatial resolution stands at 1/32°, yielding a grid of $(140 \times 4 \times 8) \times (50 \times 4 \times 8)$, equating to dimensions of $(4480 \times 1600)$, and resulting in a total of 7,168,000 particles.

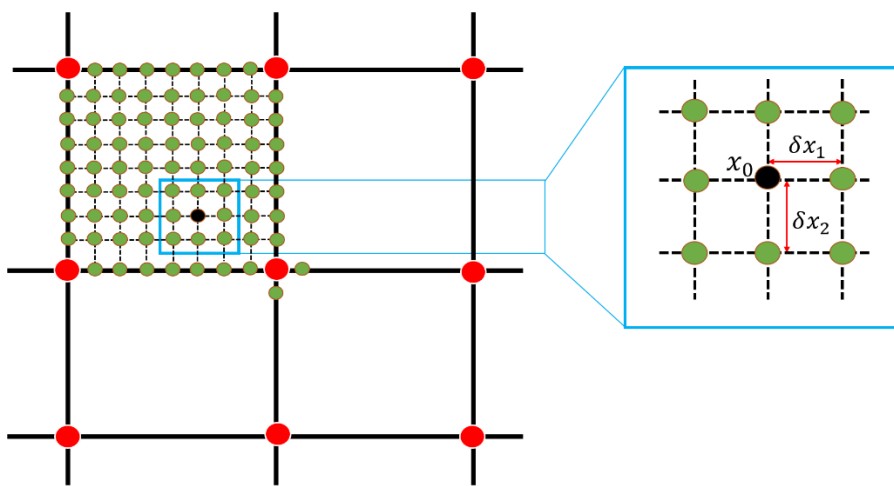

**Figure 2. Schematic diagram of the spatial relationship between a main grid point and its surrounding auxiliary points in the Cauchy-Green strain tensor field. The black dot represents an auxiliary grid point $x_0$, $\delta x_1$ and $\delta x_2$ denotes the distance between the auxiliary grid point (green) and the main grid point (red).**

As depicted inFigure 2, the coordinates of $x_0$ are denoted as $(x_i, x_j)$. Subsequently, the coordinates of the four auxiliary points around it can be represented as:

$$x_j^{up} = \left(x_j, y_j + \delta x_2\right), \; x_j^{down} = \left(x_j, y_j - \delta x_2\right) \tag{5}$$

$$x_j^{right} = \left(x_j + \delta x_1, y_j\right), \; x_j^{left} = \left(x_j - \delta x_1, y_j\right) \tag{6}$$

Once the flow field has advanced over time, we can ascertain the gradient value of the corresponding $x_0$ grid node map by determining the finite difference at the four densified auxiliary grid points around $x_0$:

$$\nabla F_{t_0}^{t_1}\left(x_j\right) \approx \left(\frac{F_{t_0}^{t_1}\left(x_j^{right}\right) - F_{t_0}^{t_1}\left(x_j^{left}\right)}{2\delta x_1} \quad \frac{F_{t_0}^{t_1}\left(x_j^{up}\right) - F_{t_0}^{t_1}\left(x_j^{down}\right)}{2\delta x_2}\right) \tag{7}$$

The incorporation of auxiliary points induces eight times of the particle motion trajectories that must be calculated, hence heightening the computational complexity of the algorithm. In terms of computational speed, the workload at this stage is akin to directly doubling the resolution in both horizontal and vertical directions on the main grid points. However, the utilization of auxiliary points allows for control over the calculation precision of $\nabla F_{t_0}^{t_1}(x_j)$. Specifically, by adjusting the perturbation sizes of $\delta_{x_1}$ and $\delta_{x_2}$, we can refine the numerical approximation of gradients. This local densification reduces discretization error in the finite-difference calculation of the Cauchy-Green strain tensor, thereby smoothing the eigenvector field and alleviating spurious fluctuations caused by numerical sensitivity to eigenvector orientation (Serra and Haller, 2017). As particles move through the unsteady, two-dimensional flow field (defined in longitude, latitude, and time), interpolation is required to estimate their trajectories. The cubic interpolation and B-splines are applied for specifically in the computation of the Cauchy-Green strain tensor field, which underlies the variational eddy boundary detection. This high-order interpolation ensures smooth and accurate estimation of the deformation gradient $\nabla F$, which is essential for evaluating the tensor. The use of cubic interpolation is justified by the global orientability of the eigenvector field derived from the Cauchy-

Green strain tensor, as shown in Serra and Haller (2017). This property enables the use of high-order schemes such as B-splines, which offer superior accuracy and continuity compared to linear interpolation typically used in non-orientable
direction fields. This theoretical advantage contributes to the precise and fully automated detection of coherent Lagrangian vortex boundaries.

When calculating the tensor field and setting up the initial particle grid, we ensured a unique correlation between particle points and threads to maintain computational accuracy. Based on the GPU's computational capacity, once the size of the initial grid$((m \times k) \times (n \times k))$ and the number of threads per Block in the GPU $(m \times n)$ are inputted, the size of the
290 Block to be allocated for each Grid $(k \times p)$ can be determined (Fig. 3). After thread allocation, the auxiliary points moving in the flow field use a GPU-accelerated fourth-order Runge-Kutta integration (RK4) to obtain the advection of Lagrangian particles. To secure sufficient computational precision, it is imperative to curtail the time step to the greatest extent feasible. Bearing computational capability in mind, we settled on a time step of 0.1 days for this study. Upon acquiring the gradient matrix, we derive the Cauchy-Green strain tensor by multiplying the transpose of the gradient matrix by itself. We then
compute the partial derivatives of the Cauchy-Green strain tensor. This procedure constitutes a pivotal step in our investigation of BHE, and it lays a crucial theoretical groundwork for subsequent numerical simulation. The RK4 procedure and tensor field computation are integrated within a CUDA-recognizable function and are recursively called in each thread. Once all threads have completed their tasks, results are compiled and transferred to the host CPU after each step. Logically, this multi-threaded, GPU-based parallel algorithm diminishes the computational complexity from $O_{(o)}$ to $O_{(o/p)}$, where p
signifies the count of threads running on the GPU.

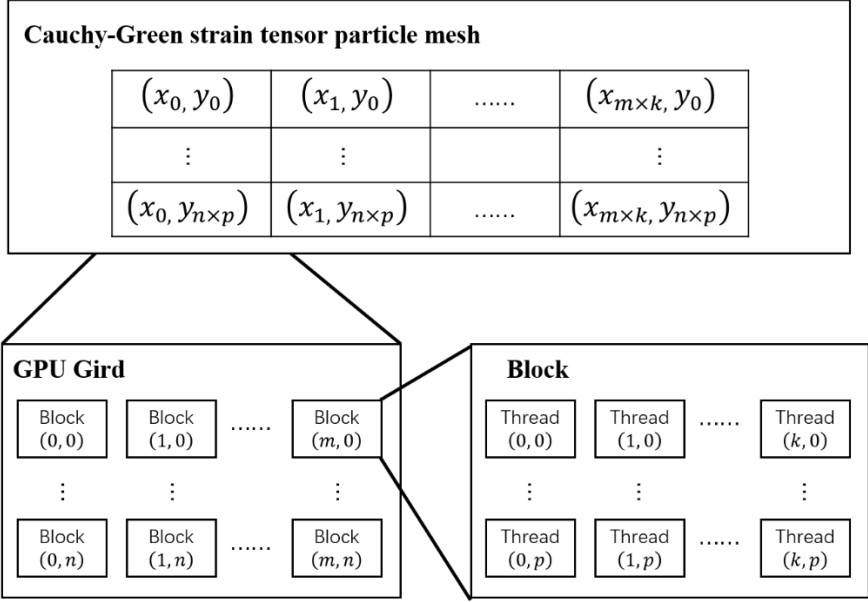

**Figure 3. Thread allocation diagram.**

2. GPU-based null geodesics extraction

For the variational problem:

$$Q[\gamma(s)] = \int_{\gamma} L(x(s), x'(s)) ds \tag{8}$$

There exists a quadratic Lagrangian function:

$$L(x, x') = \frac{1}{2} \langle x', A(x)x' \rangle \tag{9}$$

where $A(x)$ is a tensor within the study area, $\langle \cdot, \cdot \rangle$ denotes the Euclidean inner product, and the parameterized expression of the geodesic $\lambda$ is:

$$gx(x', x') = \frac{1}{2} \langle x', A(x)x' \rangle \tag{10}$$

A geodesic is existent when the Euler-Lagrange equation is satisfied:

$$\frac{1}{2} \nabla_x \langle x', A(x)x' \rangle - \frac{d}{ds}[A(x)x'] = 0 \tag{11}$$

In the computational procedure, Equation 11 can be incorporated into the earlier mentioned expression to derive the geodesics. We define the stretch rate $\lambda$ within the range of [0.9, 1.1], employing a sampling interval of 0.025, with each discrete $\lambda$ yielding a corresponding geodesic calculation. The traditional approach of calculating these geodesics and extracting null geodesic contours individually is computationally inefficient. To overcome this problem, we adopt the key innovation from Serra and Haller (2017) by introducing a polar coordinate system. This transforms the challenge from solving a complex variational problem into a more straightforward functional problem. The primary advantage is that a vortex boundary-a closed loop by definition-can be rigorously identified by testing if the rotation angle, $\phi$, completes a full $2\pi$ rotation. This mathematical simplification is particularly well-suited for parallelization, as it reframes the problem as a large set of independent calculations. We leverage this structure by employing a GPU-based parallel algorithm. In this approach, each thread is assigned the task of computing a single geodesic line, allowing for thousands of simultaneous calculations and thus reducing the overall processing time. Following this parallelized approach within the polar framework, the partial derivatives of the tensor field are used as inputs, and the computation steps for the null geodesic field adhere to the following equation:

$$ZeroSet = C^{11}(x)cos^2\phi + C^{12}(x)sin2\phi + C^{22}(x)sin^2\phi - \lambda^2 \tag{12}$$

In the equation, $\phi$ is the rotation angle. Importantly, given that the Lagrangian vortex forms a closed curve in polar coordinates and its boundary completes a full rotation, we can assign $\phi = 2\pi$. The curve meeting the condition $ZeroSet = 0$ earns the definition of a null geodesic. The application of this computational methodology significantly enriches our comprehension and simulation of BHE.

## 3. GPU-based closed elliptic Lagrangian coherent structure detection

The assemblage of null geodesics derived from the aforementioned step 2 is denoted as ZeroSet and is segmented into various seed points. Subsequently, these seed points undergo integration within the mixed tensor field, which is defined as follows:

$$x' = e_\phi \tag{13}$$

$$\phi' = -\frac{cos^2\phi\langle\nabla_x C^{11}(x), e_\phi\rangle + sin2\phi\langle\nabla_x C^{12}(x), e_\phi\rangle + cos^2\phi\langle\nabla_x C^{22}(x), e_\phi\rangle}{sin2\phi[C^{22}(x) - C^{11}(x)] + 2cos2\phi C^{12}(x)} \tag{14}$$

Within this context, $\phi$ continues to represent the rotation angle. Similar to the tensor field computation in step 1 and the geodesic calculation in step 2, we use the GPU parallel acceleration algorithm for seed point integration. To achieve closed Lagrangian curves as anticipated, the value of $\phi$ is designated as $2\pi$. Ultimately, this will result in a sequence of nested closed curves, jointly constituting an elliptical Lagrangian coherent structure.

It is essential to recognize that not all points on the null geodesics can be integrated within the tensor field to produce closed curves. Only a subset of the seed points on the null geodesics undergo integration within the tensor field, leading to nested closed vortex curves. Additionally, the quantity of layers in these nested curves is variable, ranging from a single layer to multiple ones.

Furthermore, establishing certain thresholds is necessary to discern whether the integrated curves create closed, nested boundaries. This process involves more than just calculating the distance between the curve's start and end points. It also necessitates the computation of the corresponding rotation angles, denoted as $\phi$, for these points. Should the differential equal $2\pi$, we can then classify the boundary as a fully enclosed, nested vortex boundary.

4. The boundaries of Black Hole Eddy extraction

We employed an algorithm aimed at discerning the outermost boundary from an assembly of nested, closed, elliptical Lagrangian coherent structures, deemed as the boundary of the BHE (He et al., 2022). Figure 4 illustrates a suite of closed, elliptical Lagrangian coherent structures where the central points P1 to P6 are not perfectly aligned, precluding determination of the outermost boundary through simple area comparison of the polygons. This process involves the subsequent steps:

(1) Calculate the centroid of all the elliptical Lagrangian coherent structures acquired in the prior step.

(2) Execute clustering predicated on the distances between centroids to procure the collection of centroids belonging to the same nested set of Lagrangian coherent structures. This aligns with P1 to P6 in Fig. 4.

(3) Compute the areas S1 to S6 of the Lagrangian coherent structures corresponding to P1 to P6, and identify the elliptical Lagrangian coherent structure with the maximum area as the Lagrangian Eddy boundary.

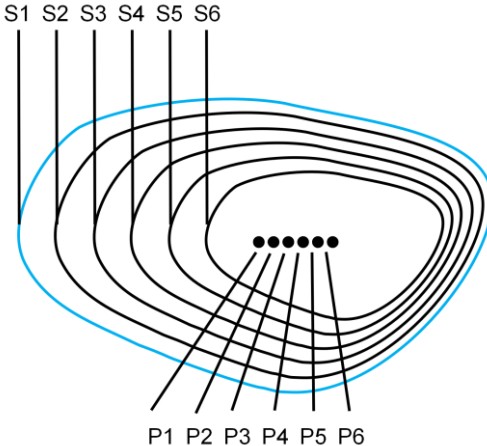

S1 S2 S3 S4 S5 S6

P1 P2 P3 P4 P5 P6

**Figure 4. The outermost boundary of the elliptic Lagrangian continuum. p1-p6 are the centers of mass of the nested elliptic Lagrangian continuum, and s1-s6 are the areas of the nested elliptic Lagrangian continuum, respectively.**

### 2.2.2 Tracking of Black Hole Eddy

On account of the strong coherence of BHE boundaries, following the advection pathway of a single particle was sufficient to track coherent water masses between the 7 d timesteps (Jones-Kellett and Follows, 2024). In this study, we transform the vortex tracking problem into a vortex-centered advection problem and automates the tracking algorithm (Fig. 6), the algorithm proceeds as follows:

(1) Input data: The BHE identification process described in Section 2.2.1 was applied repeatedly to generate the input data

for our tracking algorithm. Specifically, a new set of BHEs was identified every 7 days, creating a time series of BHE 'snapshots' covering the period from January 1, 1993, to May 5, 2023. This weekly dataset of identified BHEs is the primary input for the tracking steps detailed below.

(2) Eddy core advection: Using a GPU-parallelized RK4 algorithm (as detailed in Section 2.2.1), the coordinates of eddy cores are computed for each time step (Jones-Kellett and Follows, 2024). These coordinates are then linked to form the

advection trajectories of BHE within each day (Fig. 5).

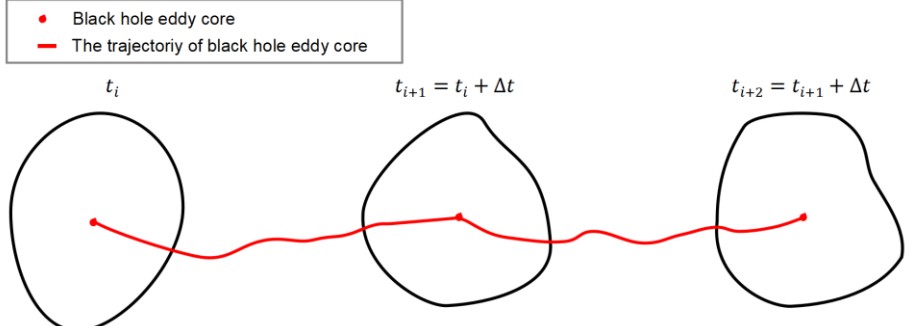

**Figure 5. Schematic of the BHEs tracking algorithm.**

(3) Candidate matching using K-D tree: To facilitate efficient spatial matching, a K-D tree partitions the space. For each time step, candidate BHE is identified within a standard circular region (radius is 1.5°, exceeding the maximum daily displacement) centered on the eddy core.

(4) Segment tracking: For consecutive time windows, each candidate BHE is evaluated. If the advected eddy core from Day $t_i$ falls within the boundary of any candidate BHE on Day $t_{i+1}$, the BHE from Day $t_i$ is marked as "live" and further checks for that core are bypassed to expedite matching. If no matches are found, the BHE is marked as "death" indicating the end of the tracking segment.

(5) Time window progression: The tracking process iterates through time, sliding the window forward until all segments between January 1, 1993 and May 5, 2023 are processed.

(6) Branch construction: Segments marked as "live" are sequentially linked into a branch until a "death" status is encountered. Each branch represents the complete life cycle of BHE, with the final output comprising the full temporal branches of tracked eddies.

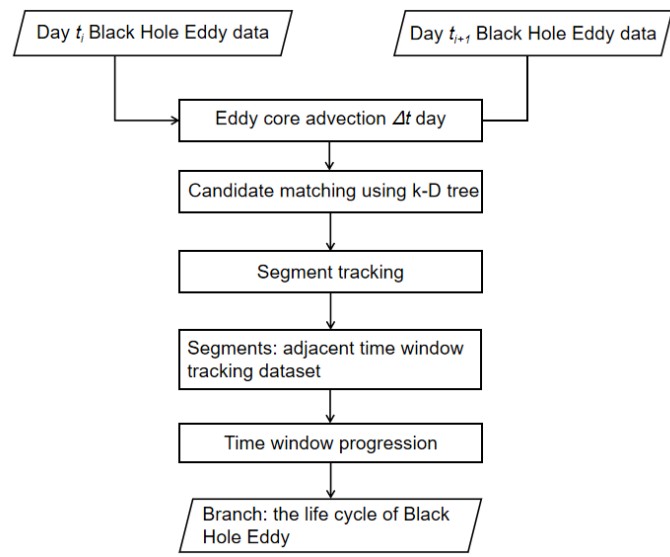

**Figure 6. Flowchart of the eddy core advection tracking algorithm.**

**2.2.3 Algorithm efficiency analysis**

Using the efficient BHE identification algorithm based on GPU parallel acceleration in Section 2.2, this study is planned to generate the North Pacific Black Hole Eddy dataset for a total of 30 years from 1993-2023. The computer configurations used during the BHE dataset computation were Intel(R) Core (TM) i7-10700 CPU @ 2.90GHz and NVIDIA GeForce RTX 3060 12GB. The algorithms were compiled and implemented in Python 3.8 (Anaconda 3), and CUDA in NVIDIA graphics card was used v11.0) for GPU acceleration. All the experiments in this paper as well as the dataset generation were done using this computer configuration.

Take the example of calculating the BHE on January 1, 2021 with a T = 30-day, Figure 7 shows the time used and the percentage of time in each part for the original method and the GPU-accelerated algorithm of this paper in calculating the BHE. It can be seen that before the optimization and acceleration of the algorithm, the null geodesics computation and the nested closed eddy boundary set computation occupy most of the time consumed in the original method, with a time share of 50.79% and 27.09%, respectively, and the tensor field computation part is also higher, with a time share of 18.14%. These three parts take up the majority of the time for identification. However, after GPU acceleration, it can be seen that for the eddy identification on the same day, the identification method in this paper significantly reduces the time used in each part of the identification of BHE. The time used in the tensor field computation part and the geodesic line computation part are about 1/15 of the original time, and the total time used is about 7 min, which greatly shortens the computation time. However, the acceleration of the Black Hole vortex boundaries extraction is not as significant, because this is the smallest part in terms of time and percentage in the original method, which suggests that this is not a key step in the algorithm's efficiency, and there is less room for speedup than the other parts. Moreover, screening each boundary in the outermost closed boundary of the BHE and performs a large number of I/O operations to write the results to files, so that the bottleneck of the storage device can constrain the acceleration results. However, after GPU acceleration, the duration of this part has been as low as 68s, which is well within the acceptable range.

With the optimization of the GPU acceleration process, the overall time consumption of the whole set of identification algorithms proposed in this paper is only about 1/13 of that of the original null geodesic algorithm. At this computational speed, generating 30 years of BHE data on a daily basis takes about 51 days, compared to 662 days using the original algorithm, thereby enabling the construction of a large-scale, long-term BHE dataset.

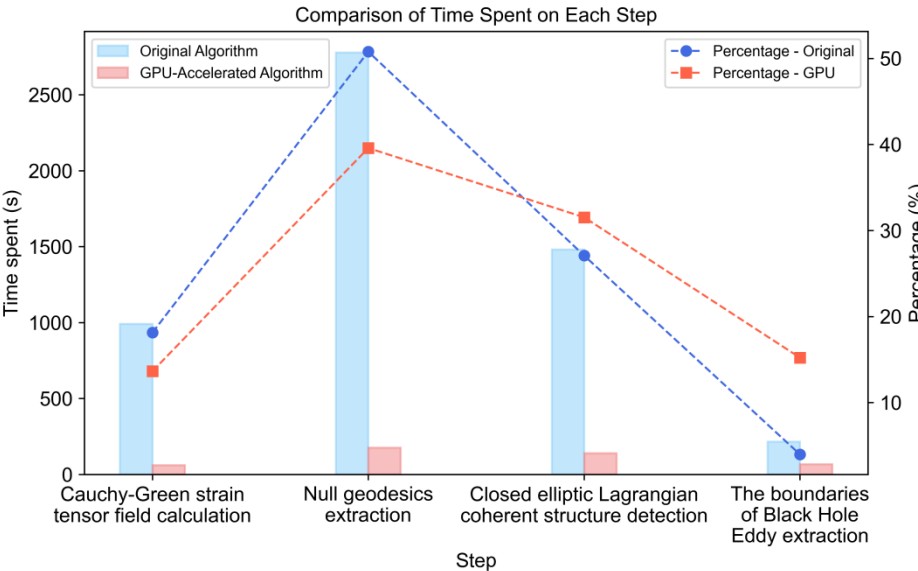

**Figure 7. Comparing the efficiency of GPU-accelerated algorithms and original methods**

# 3 Results

## 3.1 Description of eddy dataset

BHE v1.0 (Tian et al., 2025a)is composed of two main components. The first component, located in the "Eddyidentification" directory, provides the general features of the BHEs. This information is organized by integration duration, with data for 30-day and 90-day BHEs stored in two distinct JSON files. (Table 1).

| Field Name | Description |
| --- | --- |
| key | The name of the BHE. |
| Eddy ID | Index assigned to each eddy on a specific date, ordered ascendingly from 0. |
| Eddy Centroid | Longitude (°E) and latitude (°N) of the eddy center, recorded every 7 days. |
| lon, lat | Longitude and latitude coordinates outlining the boundary of the BHE. |
| Eddy Area | The area of the BHE, typically in square kilometers. |
| Eddy Types | Polarity of the BHE, classified as either *Cyclonic* or *Anticyclonic*. |
| Eddy Radius | The radius of the BHE, typically in kilometers. |
| Lam | The stretch rate associated with each BHE. |

**Table 1. Description of variables in the BHE dataset in the North Pacific Ocean.**

To validate the identified BHEs, we conducted a spatial comparison with the RCLVs dataset published by Liu and Abernathey (2022). As shown in Fig. 8a, the spatial distribution of BHEs (outlined in red for anticyclonic and blue for cyclonic) is overlaid on the FTLE (Finite-Time Lyapunov Exponent) field, with RCLVs marked by black contours. The FTLE background highlights regions of strong Lagrangian stretching, which often correspond to the material boundaries of coherent structures. Two representative regions are enlarged in  Fig. 8b and Fig. 8c, in both regions, BHE boundaries closely align with high FTLE ridges and frequently overlap with or are enclosed within RCLV, indicating that the identified BHEs represent materially coherent eddy structures.

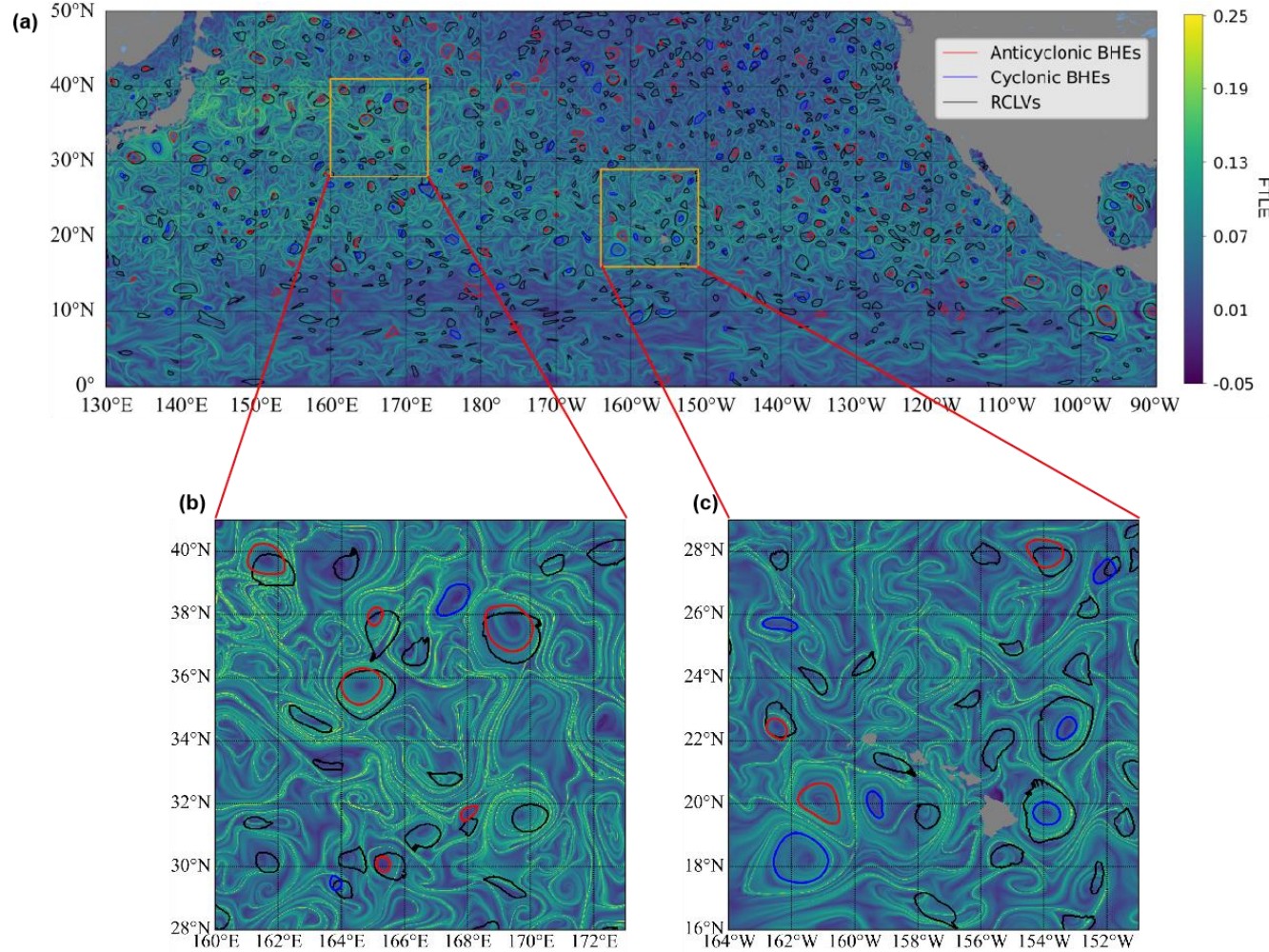

**Figure 8. Comparison between identified Black Hole Eddies (BHEs) and Rotationally Coherent Lagrangian Vortices (RCLVs) from Liu et al. (2022) over the North Pacific on 1 March 2019, over a detection time interval T = 30 days with the Finite-Time Lyapunov Exponent (FTLE) field superimposed. (a) Anticyclonic BHEs are outlined in red, cyclonic BHEs in blue, and RCLVs in black. (b, c) Zoomed-in views of two representative regions.**

The second component, the trajectories of the BHE boundaries are provided in the directory named "Eddytrajectory". This represents the first attempt at creating an open-source BHE dataset. We utilize a Pickle file containing a three-dimensional array to store the core and the boundary every 7 days for each eddy. This dataset provides 18387 with radius larger than 20 km and lifetimes longer than 4 weeks. The minimum radius threshold of 20 km (approximately 4 pixels at the 0.25° resolution of the ADT data) is applied to ensure the physical relevance and adequate resolution of detected eddies, consistent with the filtering approach of Faghmous et al. (2015) to exclude unresolved or noisy features. Additionally, the minimum lifetime of 4 weeks (28 days) is imposed to retain only coherent and dynamically meaningful structures, following common practices in mesoscale eddy studies (Chelton et al., 2011b; Faghmous et al. 2015; Chen and Han, 2019; Liu and Abernathey,

2023). While these threshold values are commonly adopted and physically justified, regional eddy statistics may be sensitive to variations in these parameters.

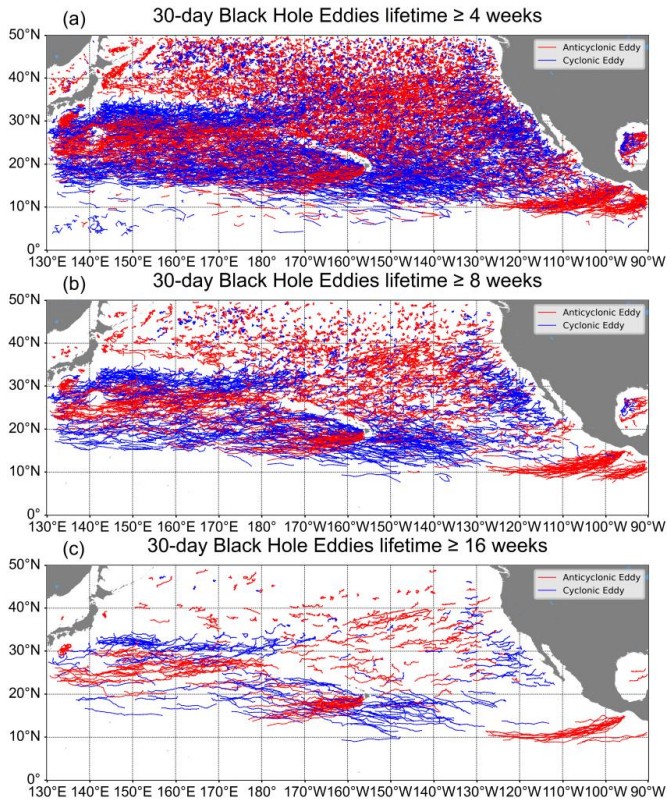

**Figure 9. The trajectories of cyclonic (blue lines) and anticyclonic (red lines) BHEs over the 30-year period Januaray 1993-May 2023 for (a) lifetimes ≥ 4 weeks and (b) lifetimes ≥ 8 weeks and (c) lifetimes ≥ 16 weeks.**

### 3.2 Analysis of the strong coherence of Black Hole Eddy

Figure 10 shows the evolution of a cyclonic eddy in the North Pacific. A set of passive tracers is initialized within the boundaries of the Eulerian eddy, the RCLVs, and the BHE on 07 January 2022 (Fig. 10a). The subsequent evolution clearly demonstrates a hierarchy of coherence. After only 14 days (Fig. 10b), the Eulerian eddy already shows significant filamentation, indicating that material inside the Eulerian boundary begins interacting with the surrounding environment shortly after initialization, while the RCLVs appear slightly filamentary structures and BHE remains highly coherent. By day 28 (Fig. 10c), the original Eulerian eddy structure has largely disintegrated, and the RCLVs also begins to exhibit distinct material leakage through filaments. Although the RCLVs demonstrate stronger material retention than Eulerian eddies, they too undergo gradual mixing, and after 56 days (Fig. 10e), the filaments become prominent, indicating a loss of coherent transport. This trend continues, and by the end of the two-month period, the Eulerian tracers are widely dispersed and the RCLV shows significant material loss. In contrast, the BHE boundary acts as a robust material barrier, remaining perfectly coherent and tightly trapping all its initial tracers throughout the entire period (Fig. 10f).

A similar analysis for a cyclonic eddy, shown in Fig. 11, the Eulerian eddy is highly dispersive, the RCLVs is moderately coherent but still leaky, while the BHE perfectly retains its initial water mass. This comparison highlights the strong material coherence of BHEs over Eulerian eddies and RCLVs. The region between the RCLVs and Eulerian eddy boundaries can be considered a transition zone, where eddy-like motions occur but without coherent material retention. Recognizing such spatial distinctions helps refine our understanding of mesoscale eddy structure and provides a pathway for exploring

submesoscale dynamics embedded within these systems.

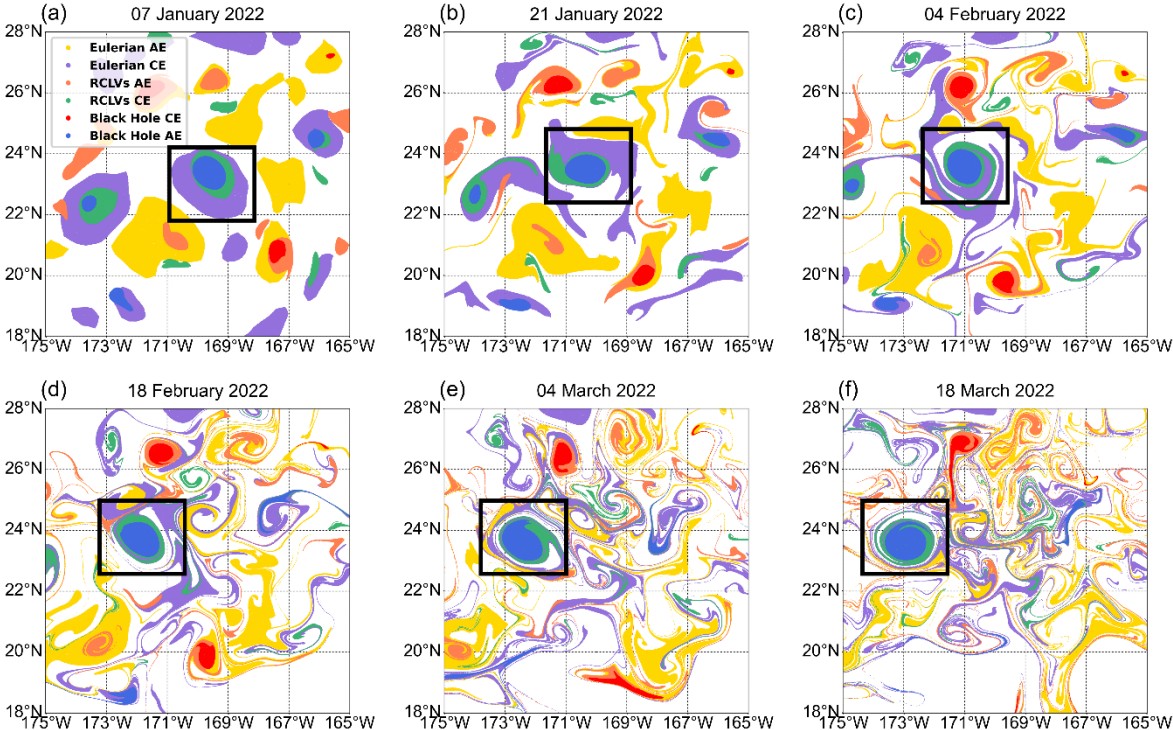

**Figure 10. Evolution of an cyclonic eddy in the open, mid-latitude North Pacific, comparing the coherence of different eddy definitions. Particles are initialized within the boundaries of a Eulerian eddy, an RCLVs, and a BHE on 07 January 2022. Panels**
**(a)-(f) show the subsequent advection of these particle sets at approximately 14-day intervals.**

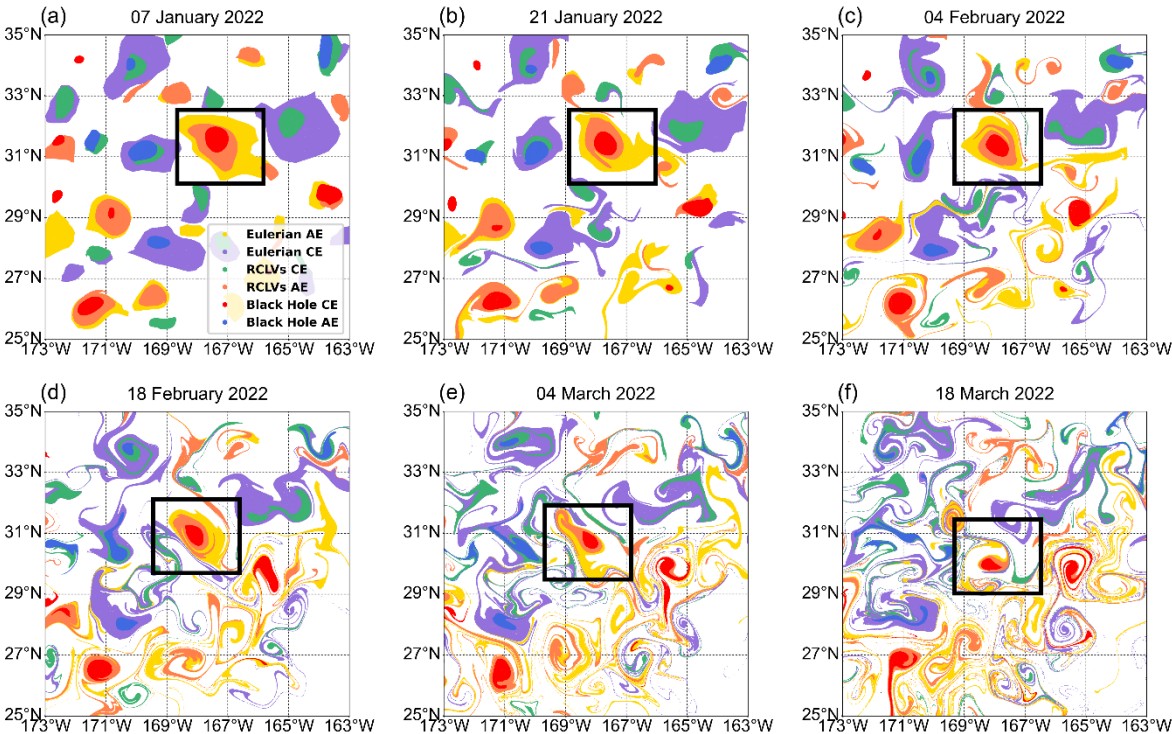

**Figure 11. Evolution of an anticyclonic eddy in the open, mid-latitude North Pacific, comparing the coherence of different eddy definitions. Particles are initialized within the boundaries of a Eulerian eddy, an RCLVs, and a BHE on 07 January 2022. Panels (a)-(f) show the subsequent advection of these particle sets at approximately 14-day intervals.3.3 Relationship of Black Hole Eddies, RCLVs and Eulerian eddies**

To investigate the relationships between the different eddy types, we adopt the classification framework proposed by Jones-Kellett and Follows (2024). This approach categorizes BHEs based on their spatial overlap with eddies identified by the Eulerian and RCLVs methods. Figure 12 illustrates the frequency of instances of different types of eddies and their distribution proportions within both cyclonic and anticyclonic categories, as well as their overlapping and non-overlapping occurrences. Instances where BHEs overlap with both Eulerian eddies and RCLVs are predominantly cyclonic (71.28%) and anticyclonic (71.02%). In contrast, the Naked Black Hole Eddy are eddies that neither overlap with SLA eddies nor with RCLVs, comprising 6.12% of cyclonic and 6.74% of anticyclonic instances. We randomly load two data records to show the particle positions during the eddy lifetime (Fig. 13 and Fig. 15). An interesting phenomenon is observed in Fig. 13a and Fig. 15a: the Naked Black Hole Eddy is not initially located around a closed SLA or LAVD contour. Nevertheless, a coherent structure is still present, suggesting that traditional Eulerian and RCLVs boundaries may not always capture the full extent of coherent transport. This type of coherent eddies are all neglected when using the Eulerian method and LAVD method. The results show that 14 days after the advection of the vortex (June 18), the Eulerian eddy and RCLVs have been severely deformed, with filamentary structures appearing and starting to mix with the background ocean current field; 28 days after

the advection (July 2), the filamentary structure of the Eulerian eddy and RCLVs further develop into an elongated ribbon, which are completely integrated with the ocean water, and the eddy morphology has ceased to exist. Unlike the Eulerian eddy and RCLVs, the Naked Black Hole Eddy remains strongly coherent throughout its life cycles. Although the Naked Black Hole Eddy rotates and moves horizontally during the motion of the flow field, no filamentary structure appears at the eddy boundary, and the particles inside the eddy remain firmly wrapped within the eddy boundary without dispersing or escaping to the surrounding ocean, which proves that the strong coherence of Naked Black Hole Eddy.

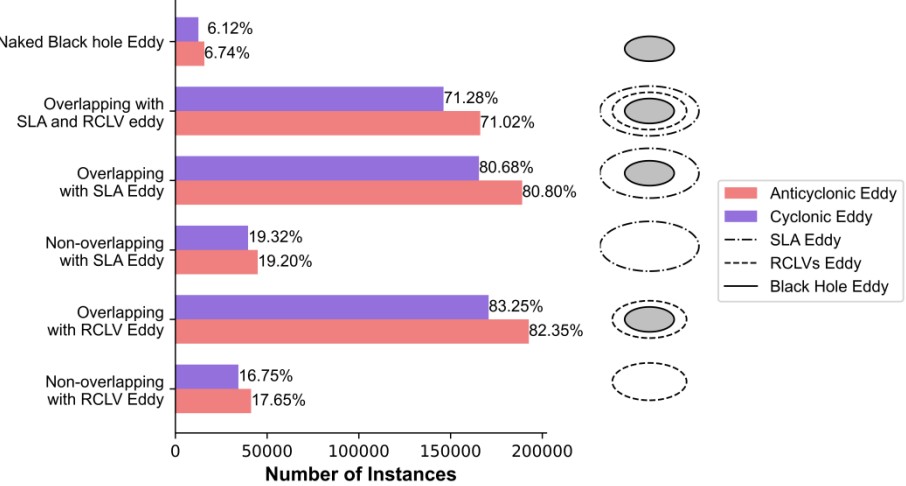

**Figure 12. The number of BHE that are concurrent (overlapping) and not concurrent (non-overlapping) with the other eddy. The percentages of eddies exhibiting the given characteristic compared with the total number of eddies of each polarity. The Naked Black Hole Eddy are eddies that neither overlap with SLA eddies nor with RCLVs.**

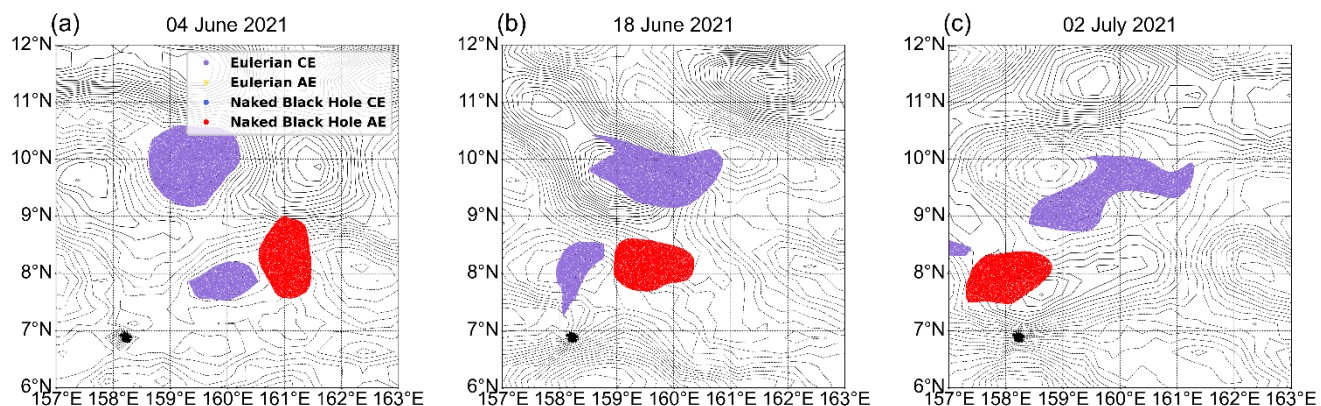

**Figure 13. Evolution of an anticyclonic Naked Black Hole Eddy (red particles). The background contours show the Absolute Dynamic Topography (ADT) field.**

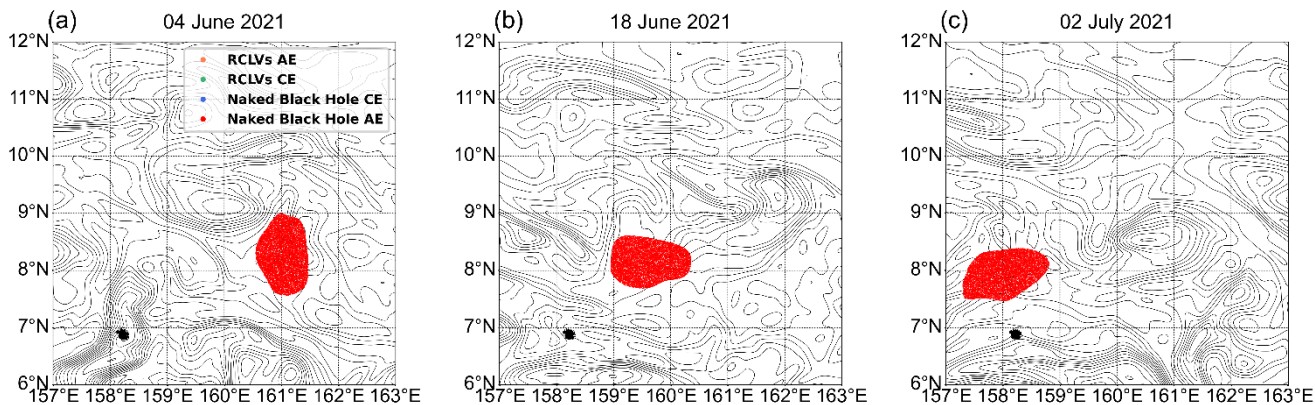

**Figure 14. The same anticyclonic Naked Black Hole Eddy from Figure 13, here overlaid on Lagrangian Averaged Vorticity Deviation (LAVD) contours.**

510

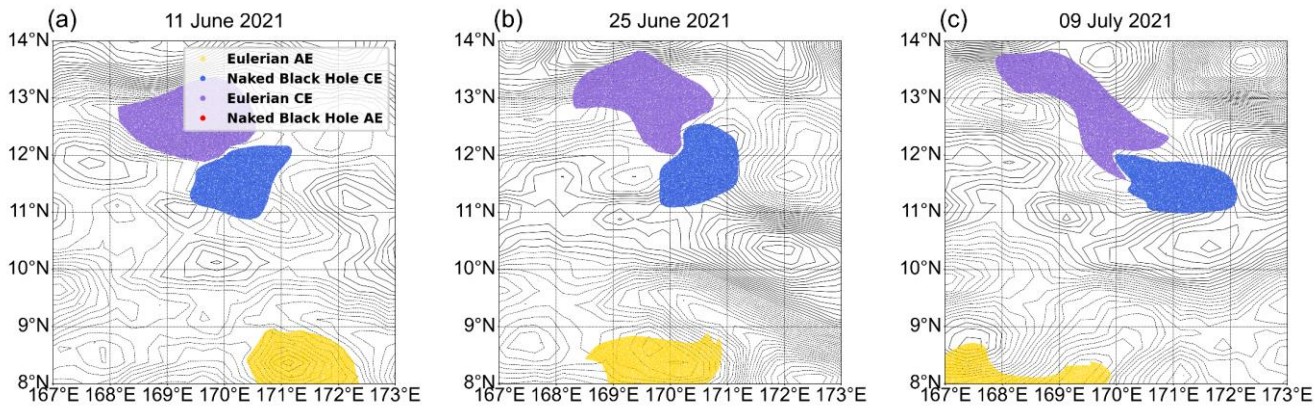

**Figure 15. Evolution of a cyclonic Naked Black Hole Eddy (blue particles), overlaid on ADT contours.**

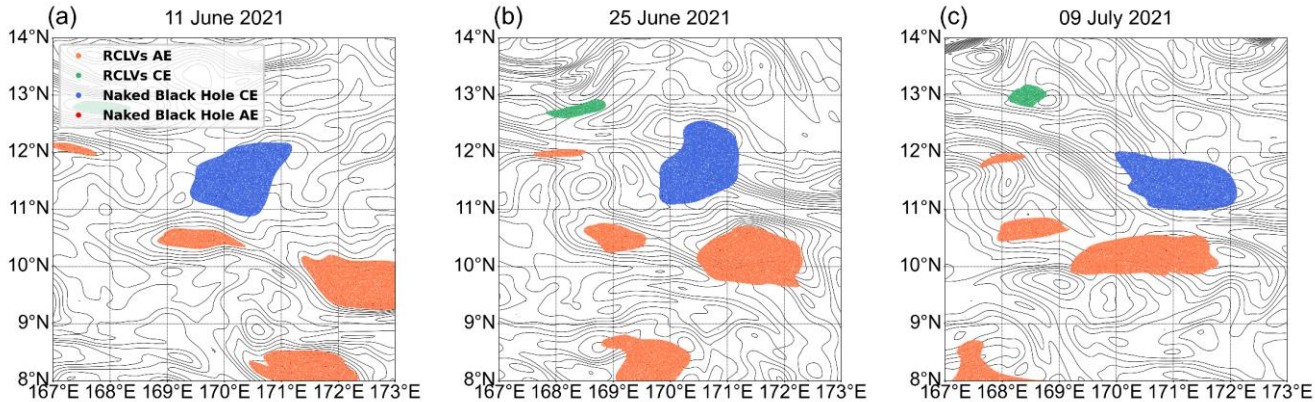

Figure 16. **The same cyclonic** Naked Black Hole Eddy **from Figure 15, here overlaid on LAVD contours.**

## 3.4 Geographic and temporal distribution of Black Hole Eddy

### 3.4.1 Geographic distribution

Figure 17 displays the geographic distribution of the BHEs and RCLVs frequency across the North Pacific Ocean. We computed this frequency by binning the positions of all identified eddy cores from 30 years dataset into $1° \times 1°$ grid cells and counting the number of occurrences in each cell. Compared with RCLVs, the number of BHE is lower, mainly in the Kuroshio Extension (30°N-40°N) and the subtropical region of the northeastern Pacific Ocean (30°N-50°N), and the eddies generated at low latitudes (below 10°N) are relatively rare Fig. 17a). The number of eddies identified by RCLVs is much higher than that of BHE, especially in the subtropical countercurrent (15°N-25°N) and the California coastal current (Fig. 17b). This is because the RCLVs method favors eddies with low coherence, while the BHE method is more rigorous and mainly identifies strongly coherent eddies (Haller and Beron-Vera, 2013; Haller et al., 2016).

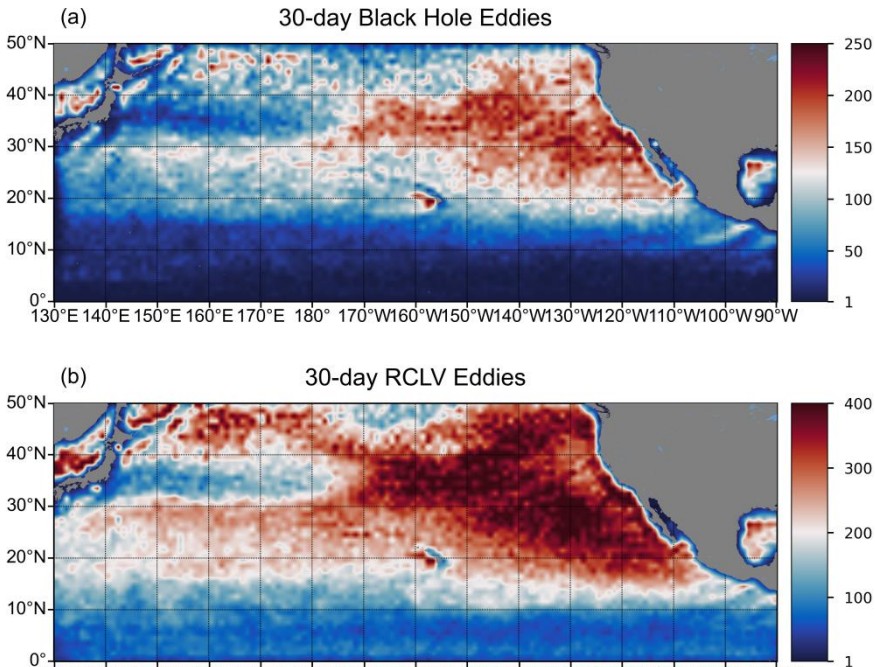

**Figure 17. The geographic distribution of eddy numbers in $1° \times 1°$ grids from 1993 to 2023 for (a) BHE and (b) RCLVs.**

To quantify the local dominance of either cyclonic or anticyclonic eddies, we compute the polarity probability index $P$, following Chaigneau et al. (2009). This index is calculated as $P = \frac{(F_A - F_C)}{(F_A + F_C)}$, where $F_A$ and $F_C$ are the respective frequencies of anticyclonic and cyclonic eddies in a given grid cell. Consequently, positive values of P indicate a prevalence of anticyclones, while negative values denote a prevalence of cyclones. The geographical distribution of this index for BHEs and RCLVs is shown Fig. 18. Compared to the relatively uniform anticyclonic polarity of RCLVs, BHE exhibits greater spatial variability

in polarity probability across latitudes. Especially, the polarity distribution of the BHE (Fig.18a) is strongly anticyclonic (red) in the region north of 30°N and more cyclonic (blue) between 10°N and 25°N. This distribution pattern may be influenced by large-scale circulation features and regional dynamics such as baroclinic instability (Chelton et al., 2011b). Previous studies (Qiu et al., 2014; Wu and Gan, 2023; Jia et al., 2011) have shown clearly that the Hawaiian Lee Eddies generate the strongest signals of two distinct groups of eddy activity in this domain. BHE also outlines two distinct pathways for Lee Eddies, clearly segregated by their polarity. In contrast, RCLVs indentifies Lee Eddies with a more scattered or mixed polarity signal in the same region. This is due to the fact that wake eddies exhibit dipole structure (cyclone-anticyclone pairs) in response to current shear and wind stress (Travis and Qiu, 2017).

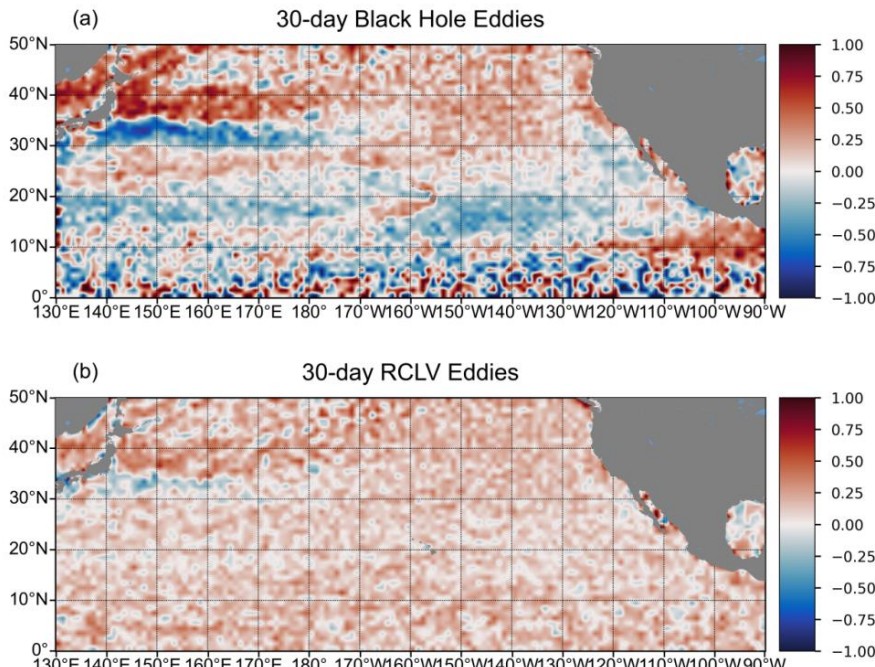

**Figure 18. Spatial distribution of the eddy polarity for (a) BHEs and (b) RCLVs from 1993 to 2023. The polarity probability (*P*) is calculated on a 1° × 1° grid. Blue shading indicates a dominance of cyclonic activity (*P* < 0), while red shading indicates a dominance of anticyclonic activity (*P* > 0).**

### 3.4.2 Temporal distribution

Figure 19a illustrates the interannual variability in the number of detected eddies. While both BHEs and RCLVs follow similar long-term trends, the total count of RCLVs is consistently higher than that of BHEs. This suggests the criteria for identifying BHEs are more rigorous, resulting in a smaller population of the most coherent structures. A notable difference in polarity preference is also observed: more anticyclonic BHEs are found than cyclonic ones, which aligns with previous statistics (Chelton et al., 2011b), whereas the opposite is true for RCLVs. The mean seasonal cycle for both eddy types is shown in b. Both methods show a clear peak in eddy occurrences in late summer (July-August) and a minimum in late winter

(January-February). Our observation of a summer peak is in agreement with the seasonal cycle of eddy kinetic energy in subtropical gyres identified by Zhai et al. (2008).

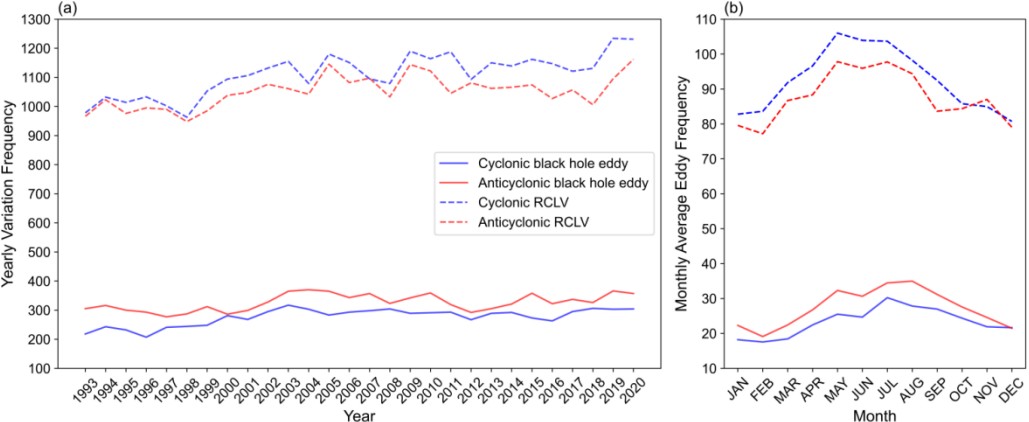

**Figure 19. Time series of eddy counts for BHEs and RCLVs from 1993 to 2020. (a) Interannual variability, showing the number of eddies detected in each 7-day time step. (b) The mean seasonal cycle, showing the monthly average number of eddies. This**
**presentation of results is adapted from Jones-Kellett and Follows (2024).3.5 Black Hole Eddy characteristic statistics**

### 3.5.1 Eddy size

Figure 20 presents a statistical comparison of the radius of BHEs and RCLVs across different latitude zones using a uniform 25 km minimum radius threshold. The median radius of RCLVs (green dashed line) exhibits a clear decreasing trend with increasing latitude, falling from approximately 100 km at 10°N to under 60 km at 50°N. This pattern is consistent with the
theoretical decrease in the first baroclinic Rossby radius of deformation at higher latitudes (Chelton et al., 2011b).

In contrast, the median radius of BHEs (blue dashed line) behaves differently. While it also decreases from low to mid-latitudes, it remains surprisingly constant at approximately 55-60 km for all latitudes above 20°N, which is different from the expected trend based on the Rossby radius. It suggests that the strict rules used to define BHEs might be finding a special group of eddies whose size is controlled by different physical forces.


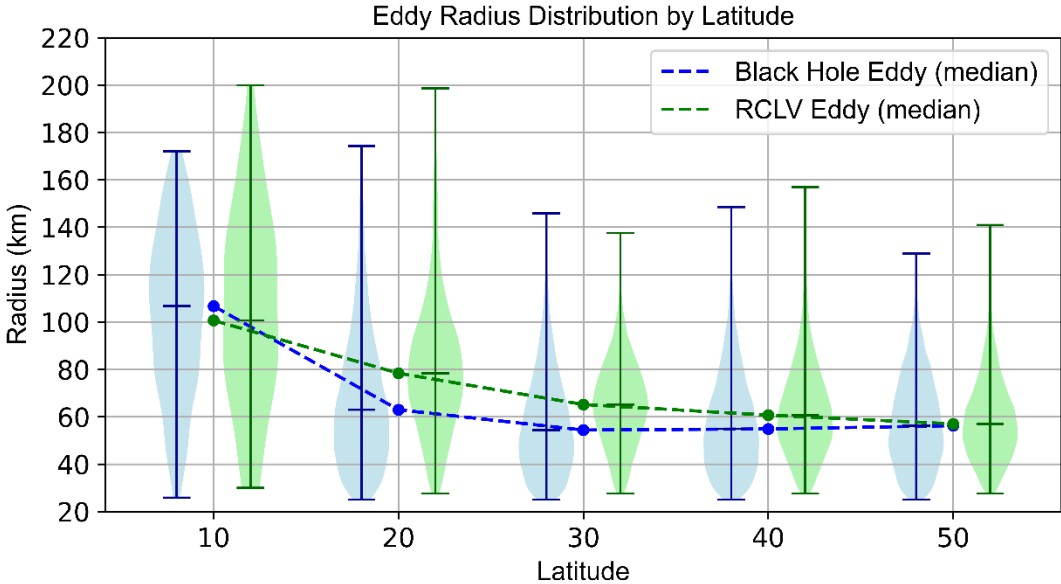

**Figure 20. Latitudinal distribution of eddy radius for BHEs and RCLVs, presented in 10° bins. The violin plots show the probability density of the radius, with the vertical lines indicating the min-max range. The dashed lines connect the median values for bins each bin.**

### 3.5.2 Eddy lifetime

We present a comparative analysis of the BHEs and RCLVs in terms of their joint probability density distributions of lifetime and radius. The results indicate that both eddy types exhibit similar lifetime distributions, predominantly concentrated within the range of 4-6 weeks, with RCLVs displaying slightly higher probability densities, suggesting a greater tendency toward shorter lifetimes. In terms of eddy radius, RCLVs have a slightly larger peak radius (51.39 km) compared to BHEs (44.52 km), yet their overall distribution patterns remain similar, with the majority of eddies falling within the 30-70 km range. Further joint probability density analysis reveals that BHEs more focus on coherent small eddies, while RCLVs are more concentrated in the long-lifetime region, potentially select eddies with longer lifetimes and larger radius.

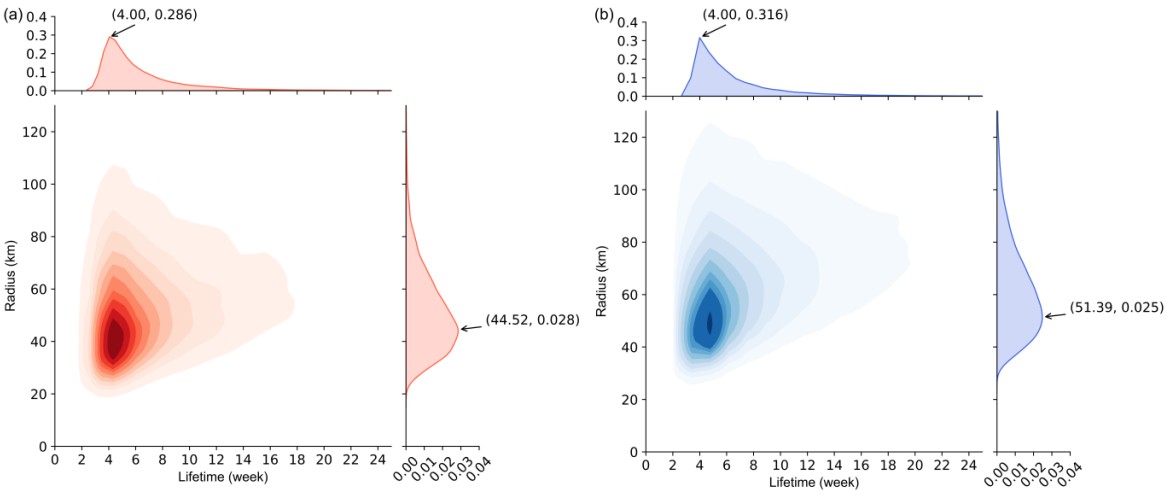

**Figure 21. Joint probability density distributions of lifetime and radius for BHEs (a, red) and RCLVs (b, blue). The contour plots represent the density distributions, while the marginal density plots on the top and right illustrate the probability distributions of lifetime and radius, respectively.**

### 3.6 Transport water by Black Hole Eddy

Following the methods used by and Zhang et al. (2014), we calculate the averaged zonal and meridional transport across the
section for each 1° × 1° grid using the following equations:

$$Q_x = \frac{\sum V \cdot C_x \cdot L}{N \cdot D_x}, Q_y = \frac{\sum V \cdot C_y \cdot L}{N \cdot D_y}$$

Where:

$Q_x$ and $Q_y$ are the average zonal and meridional transports (in m³/s) across the 1° × 1°, respectively;

$V$ and $L$ are the volume of each coherent eddy (in m³) and the lifetime of each coherent eddy (in days), respectively;

$C_x$ and $C_y$ are the zonal and meridional propagation speeds (in m/s) of the eddy core;

$N$ is the number of time points (in days) during the detection period;

$D_x$ and $D_y$ is the length of one degree of longitude and latitude (in meters), respectively, at each grid location.

The eddy volume is estimated by the simplified formulation: $V = s\pi R^2 h$, where $R$ is the eddy radius (in meters), $s = 0.5$ is a shape correction factor accounting for the bowl-like vertical structure of eddies, following Dong et al. (2014), and $h =$
$500m$ is the fixed eddy depth based on the observational analysis by Zhang et al. (2014). We note that our volume calculation is a simplified parameterization. Recent study (Deogharia et al., 2024) has revealed the complex 3D structure of Lagrangian eddies, which our surface altimetry-based analysis cannot resolve. Therefore, we use this established method to provide a first-order volume estimate, sufficient for the climatological scope of our study. For the analysis period of 1993-2020, we selected a minimum lifetime threshold of 28 days for both BHEs and RCLVs. This duration was chosen as it
corresponds to four consecutive 7-day time steps in weekly dataset, ensuring that the analyzed eddies exhibit sustained coherence over a significant number of observations.

Within the North Pacific, as shown in Fig. 22, both eddy types contribute predominantly westward (negative) zonal transport across the North Pacific basin. However, BHEs exhibits notably weaker westward transport intensity compared to RCLVs, particularly between 10° N and 30°N (Fig. 22a and Fig. 22c). In contrast, the meridional transport patterns (Fig. 22b and Fig. 22d) reveal a more diffusive and spatially scattered signal for RCLVs, with alternating northward and southward fluxes. Meanwhile, BHEs show relatively coherent southward transport signals, especially in the eastern tropical Pacific. The meridionally-integrated zonal transport profiles (Fig. 23a) further confirm that Black Hole Eddies induce a consistent but weaker westward transport compared to RCLVs. The peak value of meridionally-integrated zonal transport by BHE is only about 1.5 Sv, which is approximately one-third of the estimate of RCLVs. In the meridional direction (Fig. 23b), BHEs and RCLVs exhibit similar latitudinal trends, with prominent peaks around 10°N and 30°N, yet RCLVs also show enhanced variability and higher transport intensity in several latitude bands. However, the meridional transport of the BHEs is slightly stronger than that of the RCLVs near 40°N. This is because at near 40°N, eddies are often strongly sheared by Kuroshio Extension and tend to be stretched into non-circular, north-south eccentered structures (Early et al., 2011). RCLVs emphasizes the consistency with the center of rotation, and tends to identify "compact" rotating eddies, thus easily omitting those structures that are elongated by shear but still have transport functions (Haller, 2015; Beron-Vera et al., 2013).

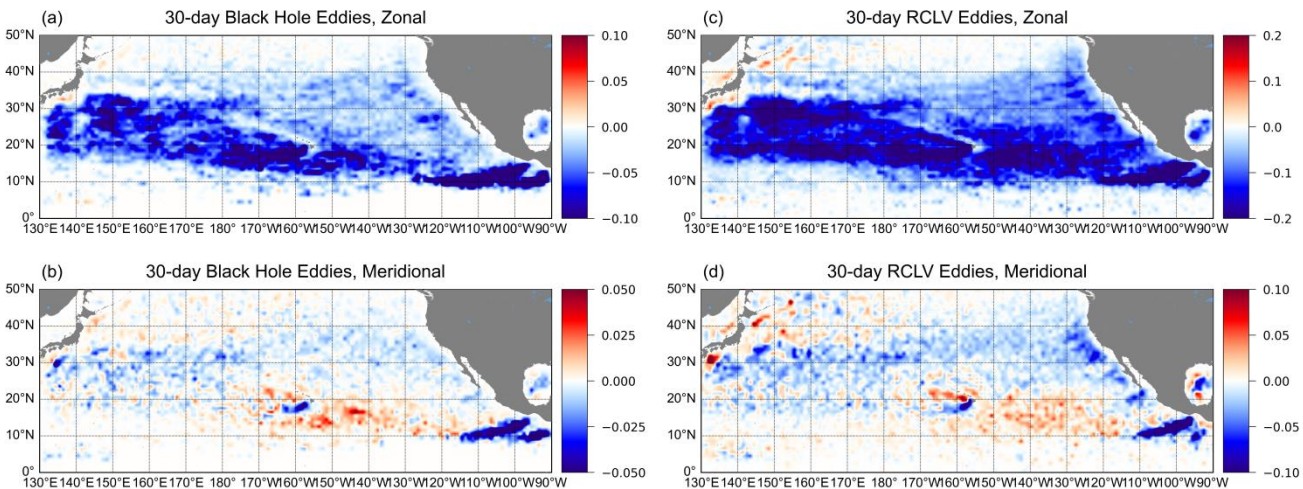

**Figure 22. The Northern Pacific distribution of the (a, c) zonal and (b, d) meridional transport by (a, c) BHEs and (b, d) RCLVs in $1° \times 1°$ bins. Note that different colorbar ranges are used here and the transport unit is the sverdrup [Sv].**

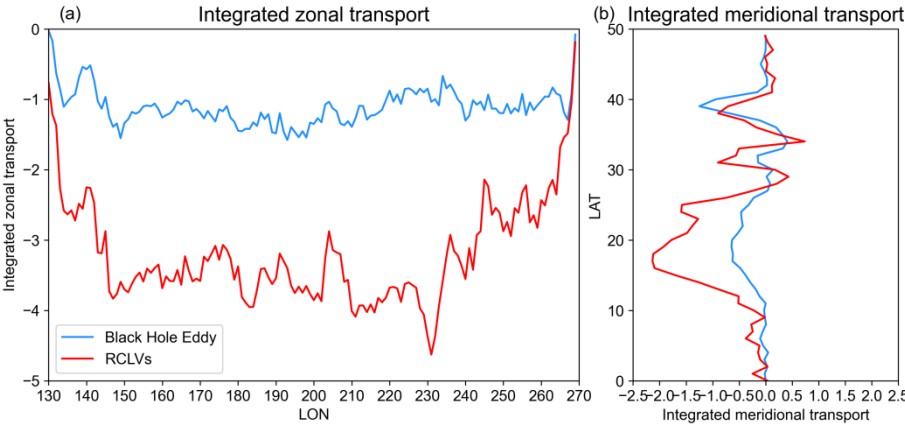

Figure 23. (a) Meridionally integrated zonal transport (in Sv) and (b) zonally integrated meridional transport by BHEs (blue lines) and RCLVs (red lines).

### 3.7 Normalization and composite analysis with Chla, SST and SSS

Composite analysis in a rotated and normalized eddy coordinate system enables researchers to study the spatial patterns of eddy-related physical fields (Mcgillicuddy et al., 2007; Chelton et al., 2011a; Gaube et al., 2013; Delcroix et al., 2019; Trott et al., 2019; Yang et al., 2023; Wang et al., 2023). As suggested in Chen et al. (2021), oceanic eddies have a significant mean egg-like shape rather than a circle or ellipse considering the geophysical anisotropy in eddy properties. Consequently, following the methods used by and Chen et al. (2021), we computed normalized composite anomalies of Chla, SST, and SSS for BHEs in the North Pacific. The resulting composites are shown in normalized coordinates: Chla in Fig. 24a and Fig. 24b, SSTA in Fig. 24c and Fig. 24d, and SSS in Fig. 24e and Fig. 24f. The chla composites (a and Fig. 24b) reveal a clear monopole-like structure, with maximum anomalies concentrated at the eddy centers for both anticyclonic and cyclonic eddies. This centralized enhancement likely reflects the vertical transport of nutrients driven by Ekman pumping-upwelling in cyclonic eddies and downwelling in anticyclonic eddies-modulated by the eddies' asymmetric, egg-like geometry. The shape-imposed constraint not only affects physical transport but also strongly influences biological responses within the eddy cores (Fig. 24). Our findings are consistent with those of Chen et al. (2021), underscoring the role of eddy geometry in shaping ecosystem structure. Notably, SST and SSS anomalies exhibit spatial distributions similar to those of Chla anomalies, with centrally peaked patterns aligned with the eddy cores. This coherence across biophysical variables highlights the tight coupling between physical dynamics and biological responses within mesoscale eddies.

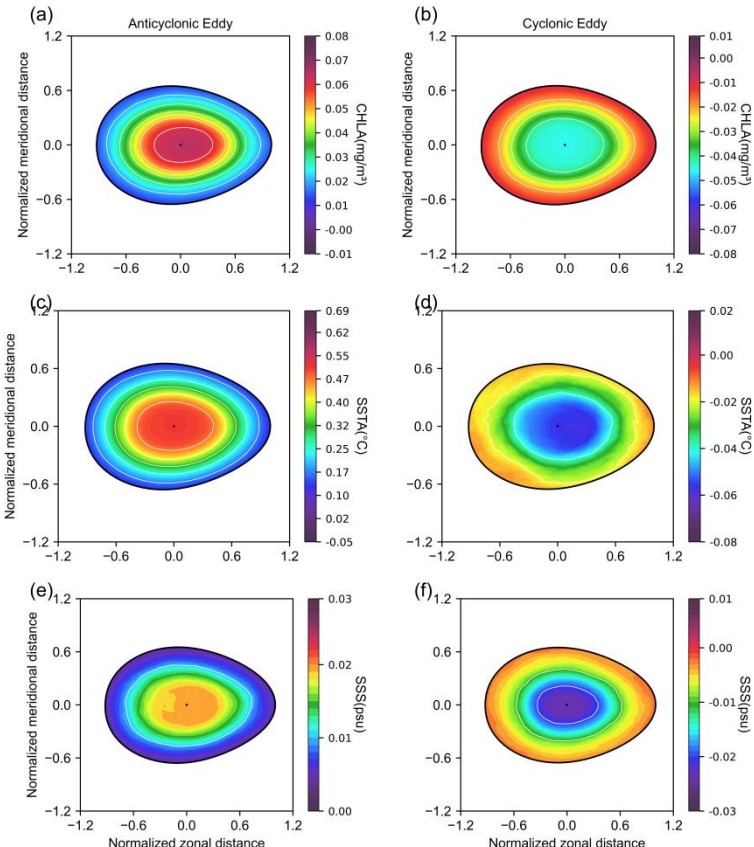

**Figure 24. Normalized composite anomalies of chlorophyll *a* concentration (Chla), sea surface temperature (SST), and sea surface salinity (SSS) for anticyclonic (left column) and cyclonic (right column) BHEs in the North Pacific.**

## 4 Dataset availability

The dataset BHE v1.0 can be accessed at https://doi.org/10.5281/zenodo.15597447 (Tian et al., 2025a). The identification dataset is saved in JSON format and the tracking dataset is saved in Pickle format, ensuring straightforward compatibility with common scientific programming languages like Python, MATLAB, and R. Detailed examples for using Python can be found in the data user manual, in which we also provide the related code to load and analyze BHE v1.0. Users can apply these algorithms to regional identification of BHE with different lifespans based on velocity fields from observations or numerical simulations.

## 5 Conclusions

In order to solve the problem of computational inefficiency caused by the high complexity and time-consuming Black Hole Eddy null-geodesic identification algorithm, we propose an efficient Black Hole Eddy identification algorithm based on

parallel acceleration of GPU, for the efficient extraction of Black Hole Eddy. Through the comparative analysis, it is found that the efficiency of the GPU parallel acceleration algorithm is about 13 times higher than the original algorithm when calculating the BHE at a given day, enabling large-scale and long-time-span ocean data applications. Moreover, the tracking algorithm based on eddy-core advection is used to realize the construction of full-life trajectories of BHE.

Based on the efficient Black Hole Eddy identification algorithm and eddy-centered advection tracking algorithm, we constructed a Black Hole Eddy dataset in the North Pacific Ocean for the years January 1, 1993 to May 5, 2023. The dataset includes the information of BHE with T=30 and 90 day respectively, and provides attributes including eddy boundaries, eddy types, eddy centers, radius, and stretching rates. Also, BHE v1.0 contains 18387 eddies with radius larger than 20 km and lifetimes longer than 4 weeks and captures the complete trajectories of BHE throughout their lifetimes. This is the first

realization of the construction of a large-scale and long-time-span Black Hole Eddy identification and tracking dataset, which effectively fills a gap in the Black Hole Eddy dataset in oceanography.

Through comparisons of Lagrangian particle advection with Eulerian eddies and RCLVs, we demonstrate that Black Hole Eddy maintains strong material coherence and is able to maintain concentration during their life cycle. Unlike traditional Eulerian eddies, whose boundaries are not material and allow for significant exchange with surrounding waters, Lagrangian

vortices such as RCLVs and BHEs are defined by materially coherent boundaries that effectively trap the water mass within. Among these Lagrangian structures, BHEs are characterized by exceptionally stable boundaries that preserve their structure without significant filamentation or mixing during their motion. This remarkable material coherence underscores the importance of BHE in accurately estimating the transport capacity of oceanic features over extended periods. Furthermore, we found that almost 6% of BHEs do not overlap with any RCLVs or Eulerian eddies, which we term the Naked Black Hole

Eddy that may be missed by Eulerian identification based on ADT and RCLVs identification entirely. Through the virtual Lagrangian particle advection validation and combined with ADT contour and LAVD contour, it is verified that the Naked Black Hole Eddy is characterized by strong coherence, which proved that the Naked Black Hole Eddy has a strong capability of wrapping the water mass. However, the correspondence between NBHEs and physically meaningful oceanic structures remains unclear and further investigation is needed to understand their underlying dynamics.

We compare the temporal and spatial distribution of BHEs with RCLVs. The spatial distribution of BHEs tends to be more in the mid-latitudes and less in the low-latitudes, and the number of RCLVs is much higher than that of BHEs, especially in the subtropical counter current and the California coastal current. In terms of temporal distribution, this study identifies similar interannual variations in BHEs occurrences with RCLVs. But RCLVs outnumber BHEs by about 5 times, which may imply that the RCLVs identification method is more likely to capture a larger number of long-lived eddies, whereas the BHE

method may have a higher coherence requirement, resulting in a smaller number of eddies being identified. For BHEs, there are more anticyclonic eddies than cyclonic eddies, while the opposite is observed for RCLVs. Seasonal variations in the presentation of peaks in eddy frequency are observed in summer and lowest frequency occurring in January and February, with cyclonic BHEs being more prevalent than anticyclonic ones. These results suggest differing underlying dynamics governing the formation and persistence of these eddies, providing valuable insights into oceanic circulation patterns.

The statistical analysis highlights key differences between Black Hole Eddies and RCLVs. Black Hole Eddies exhibit a clear latitudinal dependence in size, with the mean radius decreasing beyond 40° latitude, reflecting the influence of geostrophic balance and Coriolis effects. At lower latitudes, Black Hole Eddies tend to have a larger and more dispersed radius, while at higher latitudes, their size becomes more constrained. In contrast, RCLVs show a relatively stable radius distribution without a clear monotonic trend. In terms of lifetime, both eddy types are predominantly concentrated within 4-6 weeks, with

RCLVs showing a slightly higher probability density for shorter lifetimes. The peak radius for RCLVs is 51.39 km, slightly larger than the 44.52 km of Black Hole Eddies, but their overall distributions remain similar, with most eddies falling within the 30-70 km range. The joint probability density analysis further indicates that Black Hole Eddies are more concentrated in the small-coherent eddy category, while RCLVs tend to favor longer-lived and larger-radius eddies.

Regarding material transport, this study presents the zonal and meridional transport patterns of BHE, highlighting their

consistent westward zonal transport in subtropical regions. It was found that the transport by BHEs is only about 1.5 Sv, which is lower than that of RCLVs about three times. BHEs exhibit more uniform transport patterns over time. While the transport by BHEs is relatively small (~1.5 Sv), it does not imply insignificance. Instead, it suggests that strongly coherent eddy cores may contribute only a minor fraction to the total mesoscale transport, which contrasts with the potentially overestimated values associated with RCLVs. Our findings align with the view that the dominant contribution to mesoscale

transport may arise not from coherent eddy cores, but from incoherent processes such as stirring and filamentation at the periphery (Hausmann and Czaja, 2012; Abernathey and Haller, 2018). However, the estimates are limited to zonal and meridional surface-integrated transports. Vertical fluxes, submesoscale exchanges, and isopycnal pathways-which are particularly relevant for understanding biogeochemical cycles-are not considered in this study. More attention is required to incorporate these additional processes to provide a more complete picture of eddy-driven material transport.

Additionally, we link BHE to biogeochemical processes, particularly in terms of Chla, SSS and SST anomalies. Contrary to the dipolar structure observed in other eddy types, BHE exhibits a monopole-like structure, concentrating Chla, SSS and SST within the eddy core. This finding emphasizes the significant role of BHE in influencing marine ecosystems and biogeochemical cycling.

It is important to note that this study focuses exclusively on the North Pacific (0–50°N, 130°E–270°E), and all algorithm

validation and dataset construction are limited to this region, due to its high eddy activity (Chelton et al., 2011b), extensive altimetric coverage favorable for coherent eddy detection (Chaigneau et al., 2008). With the continued advancement of satellite altimetry technology, future research can further expand the spatial and temporal scope of BHE datasets by incorporating more diverse and high-resolution data sources. In particular, integrating depth-resolved observations-such as those from Argo floats and ocean reanalysis data-such as the GLORYS12v1 dataset from CMEMS will enable a more

detailed exploration of the three-dimensional structure and vertical dynamics of BHEs. And further investigations involving systematic sensitivity tests would help assess the robustness of the dataset under alternative detection criteria. Moreover, future studies may focus on the dynamic processes governing BHEs-such as their generation mechanisms and interactions

with surrounding eddies or mesoscale features-providing new insights into their role in ocean circulation and mesoscale turbulence.

**Appendix A: GPU-Accelerated Pseudo-Code for BHE Detection Using Null Geodesics**

Algorithm 1 provides a brief summary of the main steps performed by the PYTHON code for BHE detection based the GPU-accelerated null geodesics algorithm.

---

**Algorithm 1. BHE detection based the GPU-accelerated null geodesics algorithm.**

---

**Input:**

- u(x, y, t), v(x, y, t): 2D velocity field in time window $[t_0, t_1]$

- Range of stretch rate $\lambda \in [0.9, 1.1]$ with step size $\Delta\lambda$

- Grid of initial polar angle $\theta \in [0, 2\pi]$

- dt: integration time step

- grid resolution: Resolution of the grid

- domain bounds: Domain boundaries for the particle seeding

**#Step 1: Load preprocessed velocity field (u, v)**

VelocityField,velocity_field = load_velocity_data("u_field.nc", "v_field.nc");

**#Step 2: Initialize particle grid and allocate on GPU(see equations (5) and (6))**

Grid grid = initialize_uniform_grid(resolution=1/32.0);

Particle* d_particles = allocate_particles_on_GPU(grid);

**#Step 3: Integrate particle trajectories using RK4 (GPU parallelism)**

RK4_integrate<<<gridDim, blockDim>>>(velocity_field, d_particles, time_step=0.1, total_steps);

**#Step 4: Compute Cauchy-Green strain tensor on GPU (see equation (3))**

compute_CG_tensor<<<gridDim, blockDim>>>(d_particles, d_C_tensor, grid);

**#Step 5: Perform eigendecomposition for each C tensor (on GPU) (see equation (4))**

eigendecompose_C_tensor<<<gridDim, blockDim>>>(d_C_tensor, d_lambda1, d_lambda2, d_xi1, d_xi2);

**#Step 6: Extract null geodesics using ZeroSet condition (each thread for φ at fixed λ) (see equation (12))**

for λ in linspace (0.9, 1.1, 9): #λ from 0.9 to 1.1 with step of 0.025

```
extract_null_geodesics<<<gridDim_lambda,blockDim_phi>>>(d_C_tensor,λ,d_phi_array,d_null_flags);
```

**#Step 7: Check which geodesics are closed based on φ = 2π**

```
check_geodesic_closure<<<gridDim_geo, blockDim>>>(d_null_flags, d_phi_diff_array, d_closed_flags);
```

**#Step 8: Integrate valid seed points inside tensor field to generate nested LCS(see equations (13) and (14))**

```
integrate_seeds_in_tensor<<<gridDim_seed,blockDim>>>(d_C_tensor,d_seeds,d_nested_curves,target_phi=2π);
```

**#Step 9: Transfer nested curves to CPU for clustering and BHE boundary extraction**

```
download_curves_to_host(d_nested_curves, nested_curves);

clusters = cluster_curves_by_centroids(nested_curves);

for cluster in clusters:

    areas = compute_area_of_each_curve(cluster);

    max_curve = find_curve_with_max_area(areas);

    save_BHE_boundary(max_curve);
```

**Output:**

  - A closed contour representing the boundary of Black Hole Eddy

---

**Video supplement.** Video S1 (https://doi.org/10.5446/70890, Tian et al., 2025b): Comparison of the coherence between Eulerian Eddy, RCLVs and Black Hole Eddy, highlighting the coherence of Black Hole Eddy through the advection of Lagrangian particles. Video S2 (https://doi.org/10.5446/70891, Tian et al., 2025c): Analysis of the coherence of Naked Black Hole Eddy, highlighting the coherence of Naked Black Hole Eddy.


**Author contributions.** GC acquired funding and resources for the execution of the project. FT proposed the idea. FT, YZ and QL developed the related algorithm. FT and YZ organized the eddy dataset and conducted the data analysis. YZ, FT, SL wrote and edited the manuscript.

**Declaration of Generative AI in scientific writing.** During the preparation of this work, the authors used WORDVICEAI in order to improve readability and language, not to replace key researcher tasks such as interpreting data or drawing scientific conclusions. After using this tool, the authors reviewed and edited the content as needed and took full responsibility for the content of the publication.


**Competing interests.** The contact author has declared that none of the authors has any competing interests.

**Disclaimer.** Publisher's note: Copernicus Publications remains neutral with regard to jurisdictional claims made in the text, published maps, institutional affiliations, or any other geographical representation in this paper. While Copernicus Publications makes every effort to include appropriate place names, the final responsibility lies with the authors.

**Acknowledgment.** We are very grateful to the anonymous Reviewers for the constructive comments and helpful suggestions which significantly help us to improve the quality of this paper. The altimeter data and the SSS data were freely downloaded
from the CMEMS website (https://marine.copernicus.eu/). The high-resolution SST product was freely downloaded from the NOAA PSL website (https://psl.noaa.gov/data/gridded/data.noaa.oisst.v2.html). The mesoscale Eulerian eddy dataset was freely downloaded from the AVISO+ website at: https://data.aviso.altimetry.fr/aviso-gateway/data/META3.1exp_DT/.

**Financial support.** This work was jointly supported by National Natural Science Foundation of China (Grant No. 42530404) and National Natural Science Foundation of China (Grant No. 42030406).

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
