# Peer review of "A Black Hole Eddy Dataset of North Pacific Ocean Based on Satellite Altimetry"

_Earth System Science Data, 2025_

## Referee Comment (RC1)

Dear Editor,

Thank you for the opportunity to review the manuscript "A Black Hole Eddy Dataset of North Pacific Ocean Based on Satellite Altimetry" by Tian et al. This dataset paper presents an ambitious attempt to detect materially coherent Lagrangian eddies (so-called Black Hole Eddies, BHEs) in the North Pacific using a GPU-accelerated implementation of a geodesic eddy detection algorithm. The work contributes to the growing demand for Lagrangian eddy datasets derived from altimetry, with relevance to ocean transport, climate dynamics, and eddy-resolving model evaluation.

The authors present a new dataset (BHE v1.0) of 18,387 long-lived, materially coherent eddies in the North Pacific (1993–2023), derived via a GPU-enhanced version of the null-geodesic detection method for coherent Lagrangian vortices. They compare BHEs against Eulerian and RCLV-based eddy atlases, highlight eddy coherence via Lagrangian particle advection, and quantify zonal and meridional transports. They also report a subset of "Naked BHEs" not detected by conventional methods.

While the technical ambitions of this study are commendable, the manuscript falls short in multiple key areas. The detection framework, while important, lacks transparency, and several scientific claims are overstated without support from independent benchmarks or statistical uncertainty. The language and structure are often difficult to follow, further obscuring key methodological and conceptual points. With improvements in algorithmic clarity, quantitative rigor, and language polish, particularly the very concerning use of AI writing tools, this paper could contribute to ESSD and the broader eddy-resolving community.

1. Inadequate Algorithm Description and Lack of Code Transparency

The paper presents a GPU-accelerated version of the null-geodesic BHE detection method but does not provide sufficient technical detail to reproduce the dataset. For example:

• There is no publication or sharing of the source code, despite the centrality of the algorithm to the study.

• The CUDA kernel logic, grid configuration, and interpolation choices (e.g., cubic B-splines) are only briefly mentioned and not documented in usable form.

• The description of the numerical solver for null geodesics is too high-level and lacks algorithmic pseudocode, flowcharts, or performance benchmarks beyond one test case (January 1, 2021).

Given that ESSD emphasizes reproducibility, this is a critical omission. The dataset's utility is severely limited without a transparent and replicable workflow.

2. No Quantitative Validation of Detected Eddies

The manuscript claims that BHEs are materially coherent and more robust than Eulerian or RCLV eddies, but provides no external validation:

• There is no comparison with drifter trajectories, synthetic benchmark fields, or any ground-truth datasets.

• The evaluation is purely qualitative, relying on particle advection visualizations (Figures 8–9) without metrics like retention rate, edge dispersion, or FTLE gradients.

• The label "Naked BHEs" is introduced based on non-overlap with other eddy datasets, but without a demonstration that these features correspond to real or meaningful oceanic structures.

This absence of validation raises concerns about false positives and algorithm selectivity. A rigorous analysis—quantifying detection accuracy or boundary fidelity—is necessary to justify confidence in the dataset.

**3. Unsubstantiated Claims about Transport Accuracy**

The authors state that BHEs provide a more accurate estimate of oceanic transport (e.g., 1.5 Sv westward flow), but this assertion is flawed:

- "More accurate" implies a benchmark or truth reference, which is not provided. It is inappropriate to draw such conclusions by comparison to RCLV fluxes alone.
- The transport calculation uses idealized assumptions: a fixed eddy depth (500 m), spherical geometry, and a simplified volume formulation $V = s\pi R^2 h$. The sensitivity of transport estimates to these assumptions is never tested.
- Only zonal and meridional surface-integrated transports are estimated; vertical fluxes, submesoscale exchange, and isopycnal pathways are ignored despite their relevance to the stated motivation (e.g., biogeochemical cycles).

The transport interpretation should be reframed more cautiously, or supplemented with uncertainty ranges and alternate metrics (e.g., Lagrangian coherence indicators).

**4. Sensitivity to Detection Parameters Not Assessed**

The results are strongly conditioned on two key detection parameters:

- Minimum eddy radius: 20 km
- Minimum lifespan: 4 weeks

However, there is no sensitivity analysis to show how the number, distribution, or transport of BHEs would change with different thresholds.

- For instance, a minimum radius of 20 km excludes coastal or tropical eddies that may be materially coherent.
- A stricter lifespan cutoff (>6 or 8 weeks) may reveal whether BHEs are truly more persistent than RCLVs.
- Parameter tuning can significantly impact regional eddy statistics, yet this issue is not acknowledged or tested.

Without such analysis, the robustness and generalizability of the dataset remain in question.

**5. Geographic Scope Is Narrow but Conclusions Overreach**

While the dataset is limited to the North Pacific (0–50°N, 130–270°E), the authors occasionally use language implying broader relevance (e.g., "first BHE dataset", "offers better transport estimates").

However:

- The dynamical regimes of the Southern Ocean, Atlantic, or Indian Ocean differ considerably (e.g., stronger stratification, different baroclinic instability characteristics).
- It is unclear whether the same GPU algorithm, with its current assumptions and thresholds, would perform well in those settings.

The manuscript should clearly delimit the spatial scope of its claims and avoid generalizations unless supported by multi-basin tests or transferability arguments.

Minor comments:

While the authors appear to have used AI tools to address language issues—a practice I generally view with caution—I appreciate their transparency in doing so. That said, this manuscript still contains some grammar issues, vague phrasing, a lot of missing space inbetween, redundancy, missing or imprecise terminology, and unclear logic in comparisons, and many are obviously related to the use of AI tools. Many technical points are introduced without context or explained in overly dense mathematical language without intuitive guidance.

Frankly, if AI-generated text plays a significant role in scientific writing, it raises serious questions about why human reviewers should invest their time and expertise in carefully reading and critiquing such work. While I understand the appeal of these tools, I find their use in this context concerning and, in principle, disagreeable. I believe this issue merits ongoing discussion within the scientific community.

Following I list some, but not all issues with the writing. The authors should seriously consider taking a more thorough proofreading process or seeking assistance from a native English speaker, rather than relying heavily on AI tools for language editing. I expect to see substantial improvement than revising the listed issues.

Line 10:
"The methodologies employed for the identification of ocean coherent eddies…"
 "ocean coherent eddies" should be "coherent ocean eddies".

Line 21:
"maintain strong material coherence and are able to maintain concentration…"
Repetitive "maintain"; also "concentration" is ambiguous here. Better: "retain material coherence throughout their lifecycle without significant mixing."

Line 24:
"Transport analysis shows that BHEs induce westward transport about 1.5 Sv⋯"
 Missing "of": should be "transport of about 1.5 Sv".

Line 26:
"These finding addresses⋯"
Should be "These findings address⋯"
Line 30:
"Mesoscale eddies⋯can persist for periods of a few weeks to several years."
 "Periods of a few weeks" is vague; could benefit from citing eddy lifetimes more precisely by region.

Line 34:
"influencing on the marine material cycle⋯"

Remove "on". Correct: "influencing the marine material cycle⋯"

Line 36:
"Eulerian approachs⋯"
 Should be "approaches".

Lines 41–45:
 There is repetition about Eulerian methods being "frame-dependent" and generating filamentation. This point is important but can be made more concisely.

Line 49:
"provide a more accurate structures⋯"
Should be: "provide more accurate representations of structures⋯"

Line 52:
"Before Lagrangian coherent structures (LCSs) entered into strict mathematical definitions⋯"
 Wordy. Consider simplifying to "Before LCSs were rigorously defined⋯"

Lines 62–63:
"These eddies boundary are commonly referred to as the 'photon sphere, analogous to Black Holes⋯"
 "Eddies boundary" should be "eddy boundaries"; quotation wrong; analogy should be better justified.

Line 164:
"⋯includes daily-averaged, 5-day-averaged, 8-day-averaged⋯"
 Repetitive and wordy. Consider: "Includes datasets with daily to monthly averages⋯"

Line 214:
"...indicative of the BHE."
The sentence assumes readers understand how minimal deformation implies BHE status—should elaborate or clarify.

Line 224:
"...relies on direction field integration⋯"
"direction field" could be more accurately termed "vector field" or "eigenvector field (?)

Line 230:
"This paper utilizes CUDA as a multithreaded toolkit⋯"
 Passive voice preferred in scientific writing: "CUDA was used⋯"

Line 272:
"effectively mitigates the noise in the eigenvector field⋯"
 Needs clarification of how densification and interpolation specifically mitigate numerical

noise.

Line 290:
"the program consolidates the output data and transmits it to the host⋯"
 Wordy. Consider: "Results are compiled and transferred to the host CPU after each step."

Line 409-416:
"Non-scientific formatting. Use consistent table structure.

Line 427:
"⋯anticyclonic\cyclonic eddy⋯"
 Use proper slash spacing and formatting: "anticyclonic/cyclonic eddy".

Lines 430–450:
 The description of coherence comparisons via particle trajectories is visual but lacks a quantitative metric (e.g., relative dispersion, FTLE, boundary leakage).

Line 455:
"The simultaneous presence of the three eddy types⋯"
 Clarity: Consider rephrasing like "Instances where BHEs overlap both Eulerian eddies and RCLVs are dominated by cyclonic events⋯"

Line 460:
"the eddy in Figure 11a and Figure 12a is not initially located around a closed SLA contour or LAVD contour⋯"
The sentence is long and ambiguous. Break into two parts: one observation, one interpretation.

Line 485:
"eddies generated at low latitudes (below 10°N) are very few⋯"
 Better phrased as "are relatively rare" or "are infrequent".

Line 496:
"strongly anticyclonic (red) in the region north of 30°N⋯"
 Correlation to Kuroshio shear is speculative without supporting reference or cross-analysis.

Line 549:
"we calculate the averaged zonal and meridional transport across the section for each 1° × 1° grid by⋯"
Define all symbols clearly; use consistent formatting. Currently hard to parse.

Line 560:
"The peak value⋯is only about 1.5 Sv, which is three times smaller⋯"
Prefer "one-third of the RCLV estimate" instead of "three times smaller" (ambiguous and

mathematically sloppy).

Line 580:
"⋯Chen et al. (2021), oceanic eddies have a significant mean egg-like shape⋯"
  Tone: "Egg-like" may appear informal.

---

## Author Comment (AC1)

**Responses to reviewer 1#'s comments point by point**

MS No.: essd-2025-384

Title: A Black Hole Eddy Dataset of North Pacific Ocean Based on Satellite Altimetry

Author(s): Fenglin Tian et al.

**Dear Reviewer,**

We highly appreciate the detailed and valuable comments of the referee on our manuscript entitled "A Black Hole Eddy Dataset of North Pacific Ocean Based on Satellite Altimetry (ID: essd-2025-384)". These comments are all valuable and helpful for revising and improving our paper, as well as providing important guidance for our research. In the past few days, we have referred to the comments and improved the paper.

As follows, we would like to clarify some of the points raised by the Reviewer. The original comments begin with "**Comment**" and are quoted in italicized font, the responses begin with "**Response**" in normal font, the original sentences and phrases are in blue letters, the revised sentences and phrases are in red letters, and the line number in the revised manuscript is highlighted in yellow. We appreciate the Reviewer's warm work and taking the time to review the manuscript, and we hope that the corrections will meet with approval.

Yours Sincerely,

Fenglin Tian, Yingying Zhao, Lan Qin, Shuang Long, Ge Chen

2025-8-6

**Comment:**

This dataset paper presents an ambitious attempt to detect materially coherent Lagrangian eddies (so-called Black Hole Eddies, BHEs) in the North Pacific using a GPU-accelerated implementation of a geodesic eddy detection algorithm. The work contributes to the growing demand for Lagrangian eddy datasets derived from altimetry, with relevance to ocean transport, climate dynamics, and eddy-resolving model evaluation.

The authors present a new dataset (BHE v1.0) of 18,387 long-lived, materially coherent eddies in the North Pacific (1993–2023), derived via a GPU-enhanced version of the null-geodesic detection method for coherent Lagrangian vortices. They compare BHEs against Eulerian and RCLV-based eddy atlases, highlight eddy coherence via Lagrangian particle advection, and quantify zonal and meridional transports. They also report a subset of "Naked BHEs" not detected by conventional methods.

While the technical ambitions of this study are commendable, the manuscript falls short in multiple key areas. The detection framework, while important, lacks transparency, and several scientific claims are overstated without support from independent benchmarks or statistical uncertainty. The language and structure are often difficult to follow, further obscuring key methodological and conceptual points. With improvements in algorithmic clarity, quantitative rigor, and language polish, particularly the very concerning use of AI writing tools, this paper could contribute to ESSD and the broader eddy-resolving community.

**Response**:

Thank you for the thoughtful and constructive feedback. We appreciate the recognition of the technical ambition and the potential contribution of our dataset to the Lagrangian eddy research community. At the same time, we acknowledge your concerns regarding algorithmic transparency, scientific overstatement, and clarity of language. We hope that the revisions address your concerns and improve the overall quality of the manuscript. Please find our detailed responses to the specific comments below.

**General Comments:**

**Comment:**

*1. Inadequate Algorithm Description and Lack of Code Transparency*

*The paper presents a GPU-accelerated version of the null-geodesic BHE detection method but does not provide sufficient technical detail to reproduce the dataset. For example:*

*① There is no publication or sharing of the source code, despite the centrality of the algorithm to the study.*

② *The CUDA kernel logic, grid configuration, and interpolation choices (e.g., cubic B-splines) are only briefly mentioned and not documented in usable form.*

③ *The description of the numerical solver for null geodesics is too high-level and lacks algorithmic pseudocode, flowcharts, or performance benchmarks beyond one test case (January 1, 2021).*

①**Response:** We acknowledge your concern regarding the unavailability of the full source code. At this stage, the complete implementation of the BHE detection algorithm cannot be made publicly available due to institutional intellectual property restrictions. Additionally, the algorithm is still undergoing internal optimization and will continue to evolve as we expand the dataset.

However, to support reproducibility and public use of the dataset, we have provided open-source script for loading and visualizing the published data products. The script is included in the data repository [https://doi.org/10.5281/zenodo.15597447], allowing users to explore the results and apply the dataset to their own research. Furthermore, we have added detailed algorithmic pseudocode to Appendix A, which outlines the numerical procedure used to extract BHEs via null geodesic calculation. This addition aims to make the implementation transparent and reproducible for users.

②**Response:** Thank you for pointing out the lack of details regarding the CUDA kernel logic, grid configuration, and interpolation choices. We have added two schematic diagrams in the revised manuscript that illustrates the grid configuration (Fig. 2) (Please see Line 277-280) and CUDA kernel workflow (Fig. 3) (Please see Line 326-327) in detail.

As illustrated in Fig. 2, we enhanced the accuracy of the simulation by densifying the original velocity field grid with a resolution of 1/4° by a factor of 8. Figure 2 shows red solid nodes in the center, representing the main grid points. These are the input velocity field data points and serve as the primary computation locations for the Cauchy-Green strain tensor field. The green points surrounding them are the auxiliary nodes post-densification, and the black points correspond to the auxiliary grid point $x_0$. Following densification, we calculate the Cauchy-Green strain tensor for both the main grid points and auxiliary nodes. This detailed grid configuration facilitates a more profound understanding and enhanced accuracy in simulating the dynamics of BHE.

Figure 3 shows CUDA thread allocation logic. once the size of the initial grid($(m \times k) \times (n \times k)$) and the number of threads per Block in the GPU $(m \times n)$ are inputted, the size of the Block to be allocated for each

Grid $(k \times p)$ can be determined (Fig. 3).

[Figure]

**Figure 2. Schematic diagram of the spatial relationship between a main grid point and its surrounding auxiliary points in the Cauchy-Green strain tensor field. The black dot represents an auxiliary grid point $x_0$, $\delta x_1$ and $\delta x_2$ denotes the distance between the auxiliary grid point (green) and the main grid point (red).**

[Figure]

**Figure 3. Thread allocation diagram.**

Regarding the interpolation method, we have clarified that the cubic or B-spline interpolation is applied specifically in the computation of the Cauchy-Green strain tensor field, which forms the basis for variational vortex boundary detection. This interpolation ensures smooth and accurate estimation of the deformation gradient $\nabla F$ needed for evaluating the tensor. The choice of high-order interpolation schemes is justified

by the fact that the eigenvector field derived from the Cauchy-Green strain tensor is globally orientable, as demonstrated in Serra and Haller (2017). This global orientability allows for the use of cubic or spline interpolation schemes, which are more accurate than the linear interpolation methods typically required in non-orientable direction fields. This theoretical advantage contributes to more precise and fully automated detection of coherent vortex boundaries. We have revised the text as follows:

- Original sentence:

"Given the three-dimensional nature of the flow field-encompassing longitude, latitude, and time-as particles move, they necessitate interpolation. Striving for an optimal balance between precision and computation speed, we employ the Cubic interpolation method and use B-splines as the mixing matrix. The Cubic interpolation method provides third-order spline interpolation, thereby ensuring high-order continuity of the velocity field post-interpolation and circumventing significant errors induced by numerical sensitivity."

- Revised sentence:

"Given the three-dimensional nature of the flow field-encompassing longitude, latitude, and time-as particles move, interpolation is required to estimate their trajectories. The cubic interpolation and B-splines are applied for specifically in the computation of the Cauchy-Green strain tensor field, which underlies the variational eddy boundary detection. This high-order interpolation ensures smooth and accurate estimation of the deformation gradient $\nabla F$, which is essential for evaluating the tensor. The use of cubic interpolation is justified by the global orientability of the eigenvector field derived from the Cauchy-Green strain tensor, as shown in Serra and Haller (2017). This property enables the use of high-order schemes such as B-splines, which offer superior accuracy and continuity compared to linear interpolation typically used in non-orientable direction fields. This theoretical advantage contributes to the precise and fully automated detection of coherent Lagrangian vortex boundaries." (Please see Line 297-305)

Reference:

Serra, M. and Haller, G.: Efficient computation of null geodesics with applications to coherent vortex detection, Proceedings of the Royal Society a-Mathematical Physical and Engineering Sciences, 473, https://doi.org/10.1098/rspa.2016.0807, 2017.

③**Response:** Thank you for pointing this out. We have added detailed algorithmic pseudocode to Appendix A, which outlines the numerical procedure used to extract BHEs via null geodesic calculation. The

pseudocode explicitly includes the main steps performed by the PYTHON code with GPU parallelization. This addition aims to make the implementation transparent and reproducible for users. we have added the pseudocode as follows: (Please see Line 751-754)

**Appendix A: GPU-Accelerated Pseudo-Code for BHE Detection Using Null Geodesics**

Algorithm 1 provides a brief summary of the main steps performed by the PYTHON code for BHE detection based the GPU-accelerated null geodesics algorithm.
* * *
**Algorithm 1. BHE detection based the GPU-accelerated null geodesics algorithm.**
* * *
**Input:**

- u(x, y, t), v(x, y, t): 2D velocity field in time window $[t_0, t_1]$

- Range of stretch rate $\lambda \in [0.9, 1.1]$ with step size $\Delta\lambda$

- Grid of initial polar angle $\theta \in [0, 2\pi]$

- dt: integration time step

- grid resolution: Resolution of the grid

- domain bounds: Domain boundaries for the particle seeding

**#Step 1: Load preprocessed velocity field (u, v)**

VelocityField,velocity_field = load_velocity_data("u_field.nc", "v_field.nc");

**#Step 2: Initialize particle grid and allocate on GPU(see equations (5) and (6))**

Grid grid = initialize_uniform_grid(resolution=1/32.0);

Particle* d_particles = allocate_particles_on_GPU(grid);

**#Step 3: Integrate particle trajectories using RK4 (GPU parallelism)**

RK4_integrate<<<gridDim, blockDim>>>(velocity_field, d_particles, time_step=0.1, total_steps);

**#Step 4: Compute Cauchy-Green strain tensor on GPU (see equation (3))**

compute_CG_tensor<<<gridDim, blockDim>>>(d_particles, d_C_tensor, grid);

**#Step 5: Perform eigendecomposition for each C tensor (on GPU) (see equation (4))**

eigendecompose_C_tensor<<<gridDim, blockDim>>>(d_C_tensor, d_lambda1, d_lambda2, d_xi1, d_xi2);

**#Step 6: Extract null geodesics using ZeroSet condition (each thread for φ at fixed λ) (see equation (12))**

for λ in linspace (0.9, 1.1, 9):   #λ from 0.9 to 1.1 with step of 0.025

    extract_null_geodesics<<<gridDim_lambda,blockDim_phi>>>(d_C_tensor,λ,d_phi_array,d_null_flags);

**#Step 7: Check which geodesics are closed based on φ = 2π**

check_geodesic_closure<<<gridDim_geo, blockDim>>>(d_null_flags, d_phi_diff_array, d_closed_flags);

**#Step 8: Integrate valid seed points inside tensor field to generate nested LCS(see equations (13) and (14))**

integrate_seeds_in_tensor<<<gridDim_seed,blockDim>>>(d_C_tensor,d_seeds,d_nested_curves,target_phi= 2π);

**#Step 9: Transfer nested curves to CPU for clustering and BHE boundary extraction**

download_curves_to_host(d_nested_curves, nested_curves);

clusters = cluster_curves_by_centroids(nested_curves);

for cluster in clusters:

    areas = compute_area_of_each_curve(cluster);

    max_curve = find_curve_with_max_area(areas);

    save_BHE_boundary(max_curve);

**Output:**

  - A closed contour representing the boundary of Black Hole Eddy

In addition, we have already included performance benchmarks for the GPU-based algorithm in the Performance Analysis section (Section 2.2.3), where we quantitatively demonstrate the significant computational speedup over the CPU implementation. These benchmarks go beyond the January 1, 2021 test case and provide a broader evaluation of the algorithm's efficiency.

We hope these additions address your concerns regarding both the algorithmic detail and performance evaluation.

**Comment:**

*2.  No Quantitative Validation of Detected Eddies*

*The manuscript claims that BHEs are materially coherent and more robust than Eulerian or RCLV eddies,*

*but provides no external validation:*

① *There is no comparison with drifter trajectories, synthetic benchmark fields, or any ground-truth datasets.*

② *The evaluation is purely qualitative, relying on particle advection visualizations (Figures 8–9) without metrics like retention rate, edge dispersion, or FTLE gradients.*

③ *The label "Naked BHEs" is introduced based on non-overlap with other eddy datasets, but without a demonstration that these features correspond to real or meaningful oceanic structures.*

①**Response:** We appreciate your suggestion regarding validation. To address this point, we have incorporated a comparison with the recently published Rotationally Coherent Lagrangian Vortices (RCLVs) dataset by Liu and Abernathey (2022), which serves as a well-established reference for materially coherent eddy detection in the North Pacific. The boundaries of RCLVs are shown as black contours in Fig. 8, overlaid with our identified BHEs (red for anticyclonic, blue for cyclonic).

Moreover, to provide a more physically meaningful background and enhance the interpretability of eddy structures, we have computed the Finite-Time Lyapunov Exponent (FTLE) field based on the same Lagrangian particle advection used for BHE detection. The FTLE field (displayed as the background color in Fig. 8) highlights regions of strong stretching and deformation, which generally align with eddy boundaries. The use of FTLE as a Lagrangian diagnostic further supports the coherence of the identified BHEs and provides a robust frame for comparison with existing vortex datasets.

These additions strengthen the validation of our method and demonstrate the consistency of our results with previously published Lagrangian eddy datasets. We have added the paragraph as follows:

"To validate the identified BHEs, we conducted a spatial comparison with the RCLVs dataset published by Liu and Abernathey (2022). As shown in Fig. 8a, the spatial distribution of BHEs (outlined in red for anticyclonic and blue for cyclonic) is overlaid on the FTLE (Finite-Time Lyapunov Exponent) field, with RCLVs marked by black contours. The FTLE background highlights regions of strong Lagrangian stretching, which often correspond to the material boundaries of coherent structures. Two representative regions are enlarged in Fig. 8b and Fig. 8c, in both regions, BHE boundaries closely align with high FTLE ridges and frequently overlap with or are enclosed within RCLV, indicating that the identified BHEs represent materially coherent eddy structures." (Please see Line 454-465)

[Figure]

**Figure 8. Comparison between identified Black Hole Eddies (BHEs) and Rotationally Coherent Lagrangian Vortices (RCLVs) from Liu et al. (2022) over the North Pacific on 1 March 2019, over a detection time interval T = 30 days with the Finite-Time Lyapunov Exponent (FTLE) field superimposed. (a) Anticyclonic BHEs are outlined in red, cyclonic BHEs in blue, and RCLVs in black. (b, c) Zoomed-in views of two representative regions.**

Reference:

Liu, T. and Abernathey, R.: A global Lagrangian eddy dataset based on satellite altimetry (GLED v1.0), https://doi.org/10.5281/zenodo.7349753, 2022.

②**Response:** We appreciate your insightful comment. We acknowledge that our current description of coherence comparisons via particle trajectories is primarily qualitative and lacks a direct quantitative metric, such as retention rate, edge dispersion, or FTLE gradients. This is a limitation of our current analysis. However, we note that visual inspection of particle trajectories has been widely used in prior studies as a practical and intuitive method to assess the material coherence of eddy boundaries (Haller and Beron-Vera, 2013; Beron-Vera et al., 2015; Xia et al., 2022). In particular, long-term coherence of trapped particles, as demonstrated through their tight confinement within the evolving eddy boundary, has served as an important diagnostic in Lagrangian frameworks. We agree that incorporating quantitative metrics would strengthen the

coherence assessment and offer more objective comparisons. We will consider including such measures in future work to complement and validate the visual assessments presented in this study.

Reference:

Haller, G. and Beron-Vera, F. J.: Coherent Lagrangian vortices: the black holes of turbulence, Journal of Fluid Mechanics, 731, https://doi.org/10.1017/jfm.2013.391, 2013.

Beron-Vera, F. J., Wang, Y., Olascoaga, M. J., Goni, G. J., and Haller, G.: Objective Detection of Oceanic Eddies and the Agulhas Leakage, Journal of Physical Oceanography, 43, 1426-1438, https://doi.org/10.1175/JPO-D-12-0171.1, 2013.

Xia, Q., Li, G., and Dong, C.: Global Oceanic Mass Transport by Coherent Eddies, Journal of Physical Oceanography, 52, https://doi.org/10.1175/JPO-D-21-0103, 2022.

③**Response:** Thank you for this insightful comment. In our manuscript, "Naked BHEs" are defined as coherent material eddies that satisfy the definition of BHEs, but do not overlap with eddies identified by traditional Eulerian and RCLVs methods. While these eddies may not exhibit strong vorticity signatures, they do demonstrate high material coherence over time, as confirmed by Lagrangian analysis.

We agree that further validation is necessary to establish whether these features correspond to physically meaningful oceanic structures. In the current work, our goal is to highlight the existence of such coherent features, which may be overlooked by conventional detection techniques. We consider this a preliminary but important step toward understanding a broader spectrum of oceanic transport phenomena. We have clarified this point in the conclusions section and intend to investigate the physical nature and dynamical relevance of these "Naked BHEs" in future work. The newly added sentence is:

"However, the correspondence between NBHEs and physically meaningful oceanic structures remains unclear and further investigation is needed to understand their underlying dynamics." (Please see Line 702-703)

**Comment:**

*3. Unsubstantiated Claims about Transport Accuracy*

*The authors state that BHEs provide a more accurate estimate of oceanic transport (e.g., 1.5 Sv westward flow), but this assertion is flawed:*

① *"More accurate" implies a benchmark or truth reference, which is not provided. It is inappropriate to*

*draw such conclusions by comparison to RCLV fluxes alone.*

② *The transport calculation uses idealized assumptions: a fixed eddy depth (500 m), spherical geometry, and a simplified volume formulation $V=s\pi R^2 h$. The sensitivity of transport estimates to these assumptions is never tested.*

③ *Only zonal and meridional surface-integrated transports are estimated; vertical fluxes, submesoscale exchange, and isopycnal pathways are ignored despite their relevance to the stated motivation (e.g., biogeochemical cycles).*

*The transport interpretation should be reframed more cautiously, or supplemented with uncertainty ranges and alternate metrics (e.g., Lagrangian coherence indicators).*

①**Response:** We appreciate your insightful comment. We have revised the sentence to avoid suggesting a definitive accuracy benchmark. The new phrasing emphasizes the relatively small transport by BHEs and interprets this result in the broader context of mesoscale dynamics, highlighting that strongly coherent eddy cores may only account for a limited fraction of total transport. This interpretation is consistent with previous studies suggesting that incoherent processes may dominate lateral material exchange (Hausmann and Czaja, 2012; Abernathey and Haller, 2018). We have revised the text as follows:

● Original sentence:

"This consistency in transport allows for more accurate assessments of material transport capacity compared to the overestimated values often associated with RCLVs."

● Revised sentence:

"While the transport by BHEs is relatively small (~1.5 Sv), it does not imply insignificance. Instead, it suggests that strongly coherent eddy cores may contribute only a minor fraction to the total mesoscale transport, which contrasts with the potentially overestimated values associated with RCLVs. Our findings align with the view that the dominant contribution to mesoscale transport may arise not from coherent eddy cores, but from incoherent processes such as stirring and filamentation at the periphery (Hausmann and Czaja, 2012; Abernathey and Haller, 2018)." (Please see Line 725-730)

Reference:

Abernathey, R. and Haller, G.: Transport by Lagrangian Vortices in the Eastern Pacific, Journal of Physical Oceanography, 48, 667-685, https://doi.org/10.1175/JPO-D-17-0102.1, 2018.

Hausmann, U. and Czaja, A.: The observed signature of mesoscale eddies in sea surface temperature and the associated heat transport, Deep-Sea Res. Pt. I, 70, 60-72, https://doi.org/10.1016/j.dsr.2012.08.005, 2012.

②**Response:** Thank you for your comment. We agree that our transport calculation is based on simplified assumptions, including a fixed eddy depth of 500 m, spherical geometry, and a volume formulation. These assumptions follow common practice in mesoscale eddy studies. Specifically, the eddy depth of 500 m is based on observational evidence from Zhang et al. (2014), who show that warm-core eddies typically trap fluid with closed PV contours down to this depth. The shape correction coefficient s=0.5 is adopted following Dong et al. (2014), who suggest this as a conservative estimate accounting for the bowl-like geometry of eddies. The main focus of this paper is to compare the eddy-induced volume transport by BHEs and RCLVs based statistical means. Our results show that the transport by BHEs is approximately one-third of that by RCLVs, which is consistent with previous findings (Dong et al., 2012). Our objective in estimating eddy transport is not to determine the absolute magnitude with high precision, but rather to enable a consistent relative comparison between BHEs and the reference RCLVs dataset. As both estimates adopt the same assumptions and formulation, the relative differences are robust and meaningful within the context of this study.

Reference:

Dong, C. M., McWilliams, J. C., Liu, Y., and Chen, D. K.: Global heat and salt transports by eddy movement, Nature Communications, 5, https://doi.org/10.1038/ncomms4294, 2014.

Zhang, Z. G., Wang, W., and Qiu, B.: Oceanic mass transport by mesoscale eddies, Science, 345, 322-324, https://doi.org/10.1126/science.1252418, 2014.

③**Response:** We appreciate your comment and agree that this is a limitation of the present study. We have added sentences in the conclusions section to explicitly acknowledge this constraint and to clarify that future work will aim to incorporate vertical and submesoscale fluxes, as well as isopycnal transport pathways, to advance our understanding of eddy-mediated transport in a more comprehensive framework. The newly added sentence is:

"However, the estimates are limited to zonal and meridional surface-integrated transports and are constrained by the use of idealized assumptions in the transport calculation. Vertical fluxes, submesoscale exchanges, and isopycnal pathways-which are particularly relevant for understanding biogeochemical cycles-are not

considered in this study. More attention is required to incorporate these additional processes to provide a more complete picture of eddy-driven material transport." (Please see Line 730-733)

**Comment:**

4. *Sensitivity to Detection Parameters Not Assessed*

    ① *The results are strongly conditioned on two key detection parameters:*

- *Minimum eddy radius: 20 km*
- *Minimum lifespan: 4 weeks*

**Response:** We appreciate your thoughtful comment regarding the potential sensitivity of our results to detection parameters. While a full sensitivity analysis is beyond the scope of this study, we acknowledge the importance of this point. However, we would like to clarify that our choice of a minimum eddy radius of 20 km is grounded in established practices in previous studies.

Specifically, this radius threshold corresponds to approximately 4 pixels in the 0.25° gridded AVISO ADT dataset, which aligns with the minimal resolvable scale based on the spatial characteristics of the AVISO product. Specifically, Faghmous et al. (2015) applied a 4-pixel minimum size threshold when identifying eddies in AVISO SLA data with 0.25° resolution, which corresponds approximately to a radius of 20 km. This criterion has been widely adopted in the literature to ensure that detected features are adequately resolved while avoiding the inclusion of noise and subgrid-scale artifacts.

Regarding the minimum eddy lifespan threshold (4 weeks), we acknowledge your concern and appreciate the opportunity to clarify. This threshold is commonly adopted in the mesoscale eddy literature to ensure that only physically meaningful and dynamically coherent eddies are retained.

Short-lived eddies often reflect transient noise, poorly defined structures, or tracking artifacts. As such, previous studies have frequently excluded these to enhance the robustness of statistical and dynamical analyses, while longer-lived eddies contribute more significantly to transport and ecological processes (Chelton et al., 2011b; Faghmous et al., 2015; Chen and Han, 2019). Additionally, recent Lagrangian-based global eddy dataset (Liu and Abernathey, 2023) has applied 30-day lifetime constraints to filter out unstable or ambiguous features, which is similar to our study.

To make this clearer, we have added a brief explanation of this parameter choice in Section 3.1 of the revised manuscript.

"The minimum radius threshold of 20 km (approximately 4 pixels at the 0.25° resolution of the ADT data) is applied to ensure the physical relevance and adequate resolution of detected eddies, consistent with the filtering approach of Faghmous et al. (2015) to exclude unresolved or noisy features. Additionally, the minimum lifetime of 4 weeks (28 days) is imposed to retain only coherent and dynamically meaningful structures, following common practices in mesoscale eddy studies (Chelton et al., 2011b; Faghmous et al., 2015; Chen and Han, 2019; Liu and Abernathey, 2023)." (Please see Line 467-471)

Reference:

Chelton, D. B., Schlax, M. G., and Samelson, R. M.: Global observations of nonlinear mesoscale eddies, Progress in Oceanography, 91, 167-216, https://doi.org/10.1016/j.pocean.2011.01.002, 2011b.

Chen, G., and Han, G.: Contrasting short-lived with long-lived mesoscale eddies in the global ocean, Journal of Geophysical Research: Oceans, 124, 3149–3167, https://doi.org/10.1029/2019JC014983, 2019.

Faghmous, J. H., Frenger, I., Yao, Y. S., Warmka, R., Lindell, A., and Kumar, V.: A daily global mesoscale ocean eddy dataset from satellite altimetry, Scientific Data, 2, https://doi.org/10.1038/sdata.2015.28, 2015.

Liu, T. Y. and Abernathey, R.: A global Lagrangian eddy dataset based on satellite altimetry, Earth System Science Data, 15, 1765-1778, https://doi.org/10.5194/essd-15-1765-2023, 2023.

*However, there is no sensitivity analysis to show how the number, distribution, or transport of BHEs would change with different thresholds.*

- *For instance, a minimum radius of 20 km excludes coastal or tropical eddies that may be materially coherent.*

- *A stricter lifespan cutoff (>6 or 8 weeks) may reveal whether BHEs are truly more persistent than RCLVs.*

- *Parameter tuning can significantly impact regional eddy statistics, yet this issue is not acknowledged or tested.*

*Without such analysis, the robustness and generalizability of the dataset remain in question.*

**Response:** We appreciate your insightful comment regarding the sensitivity of the detection results to parameter choices. We acknowledge that the choice of a 20 km minimum radius threshold, while motivated by the need to ensure feature detectability and coherence within the resolution of the SLA dataset, may exclude smaller-scale eddies, particularly those in coastal or tropical regions, that are nonetheless materially

coherent. This trade-off between resolution-limited detection and inclusiveness is a common constraint in mesoscale eddy identification based on altimetry-derived fields. Future efforts could incorporate multi-scale detection approaches or higher-resolution datasets to better capture submesoscale or marginally resolved features. While we agree that stricter lifespan thresholds (e.g., 6-8 weeks) could offer insight into the persistence of BHEs relative to RCLVs, we chose 4 weeks to strike a balance between data availability, computational feasibility, and statistical representativeness. Furthermore, the same radius and lifespan thresholds were applied when detecting RCLVs for intercomparison, ensuring consistency in the evaluation of both eddy types. We fully acknowledge that detection parameters can affect eddy statistics. While a comprehensive sensitivity analysis is beyond the scope of the present study, we now explicitly acknowledge this limitation and the sensitivity analysis of Black Hole Eddies statistics under different parameter regimes to be explored in future work. The newly added sentence is:

"While these threshold values are commonly adopted and physically justified, regional eddy statistics may be sensitive to variations in these parameters." (Please see Line 475-476)

"And further investigations involving systematic sensitivity tests would help assess the robustness of the results under alternative detection criteria." (Please see Line 746-747)

**Comment:**

*5. Geographic Scope Is Narrow but Conclusions Overreach*

*While the dataset is limited to the North Pacific (0-50°N, 130-270°E), the authors occasionally use language implying broader relevance (e.g., "first BHE dataset", "offers better transport estimates").*

*However:*

*• The dynamical regimes of the Southern Ocean, Atlantic, or Indian Ocean differ considerably (e.g., stronger stratification, different baroclinic instability characteristics).*

*• It is unclear whether the same GPU algorithm, with its current assumptions and thresholds, would perform well in those settings.*

*The manuscript should clearly delimit the spatial scope of its claims and avoid generalizations unless supported by multi-basin tests or transferability arguments.*

**Response:** We appreciate your insightful comment and agree that the current study's conclusions should be interpreted within the geographic context of the North Pacific. In response, we have carefully revised the

manuscript to more clearly delimit the spatial scope of our findings. Specifically, we now explicitly state that our dataset and results pertain solely to the North Pacific region (0-50°N, 130-270°E) and that any broader implications are tentative and subject to further validation. We have also removed or softened language that previously implied global generality. Additionally, we now acknowledge in the conclusions section that dynamical regimes and detection challenges may differ in other ocean basins, and that the applicability of our GPU-based method to those regions remains to be evaluated in future work. We have revised the text as follows:

- Original sentence:

  "This study presents an efficient Graphics Processing Unit (GPU) -based BHE identification algorithm, enhancing computational efficiency by approximately 13 times compared to the existing methods. Using this algorithm, the North Pacific a Black Hole Eddy dataset (BHE v1.0) is constructed for the first time, based on satellite-derived surface geostrophic velocity data from January 1, 1993 to May 5, 2023 (Tian, F. L., Zhao, Y. Y., Long, S., and Chen, G.: A Black Hole Eddy Dataset of North Pacific Ocean Based on Satellite Altimetry (BHE v1.0). Zenodo [dataset], https://doi.org/10.5281/zenodo.15597447, 2025a)."

- Revised sentence:

  "This study presents an efficient Graphics Processing Unit (GPU) -based BHE identification algorithm for the North Pacific (0-50°N, 130°-270°E), enhancing computational efficiency by approximately 13 times compared to the existing methods. Using this algorithm, a Black Hole Eddy dataset (BHE v1.0) is constructed for the first time in the North Pacific Ocean, based on satellite-derived surface geostrophic velocity data from January 1, 1993 to May 5, 2023 (Tian et al., 2025a)." (Please see Line 13-16)

- Original sentence:

  "As shown in Fig. 16, both eddy types contribute predominantly westward (negative) zonal transport across the North Pacific basin. However, BHEs exhibits notably weaker westward transport intensity compared to RCLVs, particularly between 10° N and 30°N (Fig. 18a and Fig. 18c)."

- Revised sentence:

  "Within the North Pacific, as shown in Fig. 20, both eddy types contribute predominantly westward

(negative) zonal transport across the North Pacific basin. However, BHEs exhibits notably weaker westward transport intensity compared to RCLVs, particularly between 10° N and 30°N (Fig. 20a and Fig. 20c)". (Please see Line 628)

- Added sentence:

"It is important to note that this study focuses exclusively on the North Pacific (0-50°N, 130°E-270°E), and all algorithm validation and dataset construction are limited to this region, due to its high eddy activity (Chelton et al., 2011b), extensive altimetric coverage favorable for coherent eddy detection (Chaigneau et al., 2008)." (Please see Line 739-741)

**Minor Comments:**

While the authors appear to have used AI tools to address language issues-a practice I generally view with caution-I appreciate their transparency in doing so. That said, this manuscript still contains some grammar issues, vague phrasing, a lot of missing space inbetween, redundancy, missing or imprecise terminology, and unclear logic in comparisons, and many are obviously related to the use of AI tools. Many technical points are introduced without context or explained in overly dense mathematical language without intuitive guidance. Frankly, if AI-generated text plays a significant role in scientific writing, it raises serious questions about why human reviewers should invest their time and expertise in carefully reading and critiquing such work. While I understand the appeal of these tools, I find their use in this context concerning and, in principle, disagreeable. I believe this issue merits ongoing discussion within the scientific community.

Following I list some, but not all issues with the writing. The authors should seriously consider taking a more thorough proofreading process or seeking assistance from a native English speaker, rather than relying heavily on AI tools for language editing. I expect to see substantial improvement than revising the listed issues.

**Response:** We appreciate your thoughtful comments and concerns regarding the use of AI tools in academic writing. We would like to clarify that AI-based tools were only used to assist with surface-level English polishing, such as grammar, spelling, and sentence fluency. At no point did we use AI tools to generate technical content, introduce domain-specific terminology, or compose the scientific arguments in the manuscript. All domain knowledge, analyses, and interpretations are entirely based on the authors' understanding and expertise.

Following the suggestions, we have carefully revised the manuscript to address the specific language issues that were listed, including grammar corrections, reducing redundancy, clarifying vague or imprecise terminology. In addition, we performed a thorough re-read of the entire manuscript and made further improvements beyond the reviewer's list, including enhancing transitions, simplifying complex sentences, and rephrasing several technical explanations to be more intuitive and accessible.

We fully understand and respect the reviewer's concerns about the broader implications of AI usage in scientific writing. We share the view that AI tools should be used responsibly and transparently, and only as a support mechanism. We hope the revised version meet with your approval.

Thank you again for your valuable feedback, which significantly helped us improve the manuscript.

**Comment 1:** *Line 10: "The methodologies employed for the identification of ocean coherent eddies..."*

*"ocean coherent eddies" should be "coherent ocean eddies".*

**Response:** Thank you for your helpful comment. "ocean coherent eddies" was reworded as "coherent ocean eddies". (Please see Line 10)

**Comment 2:** *Line 21: "maintain strong material coherence and are able to maintain concentration..."*

*Repetitive "maintain"; also "concentration" is ambiguous here. Better: "retain material coherence throughout their lifecycle without significant mixing."*

**Response:** Thank you for pointing this out. We agree that the original sentence was repetitive and the term "concentration" could be ambiguous. We have revised the sentence to: "... retain material coherence throughout their lifecycle without significant mixing." This revision improves clarity and avoids redundancy. (Please see Line 21-22)

**Comment 3:** *Line 24: "Transport analysis shows that BHEs induce westward transport about 1.5 Sv..."*

*Missing "of": should be "transport of about 1.5 Sv".*

**Response:** Thank you for your careful reading. We have corrected the sentence by adding the missing preposition, and it now reads: "Transport analysis shows that BHEs induce westward transport of about 1.5 Sv…" (Please see Line 26)

**Comment 4:** *Line 26: "These finding addresses…"*

*Should be "These findings address…"*

**Response:** Thank you for pointing out the grammatical error. We have corrected the sentence to: "These

findings address…" in the revised manuscript. (Please see Line 27)

**Comment 5:** *Line 30: "Mesoscale eddies…can persist for periods of a few weeks to several years."*

*"Periods of a few weeks" is vague; could benefit from citing eddy lifetimes more precisely by region.*

**Response:** We thank the reviewer for this helpful suggestion. We agree that "a few weeks to several years" is vague and that it is more informative to provide region-specific statistics. Since our study focuses on the North Pacific region (130°E-270°E, 0-50°N), we have revised the sentence to reflect typical eddy lifetimes in this area. Based on Chelton et al. (2011a) and Chen and Han (2019), most mesoscale eddies in the North Pacific have lifetimes ranging from approximately 4 weeks to one year, with long-lived eddies-persisting more than 12 months-being less frequent but still significant, especially in regions influenced by strong currents such as the Kuroshio Extension. We have revised the text as follows:

● Original sentence:

"Mesoscale eddies, one of the most prominent processes in the ocean, typically range from several tens to hundreds of kilometers in spatial scale and can persist for periods of a few weeks to several years (Chelton et al., 2011a; Chaigneau et al., 2008)"

● Revised sentence:

"Mesoscale eddies, one of the most prominent processes in the ocean, typically range from several tens to hundreds of kilometers in spatial scale (Chelton et al., 2011a; Chaigneau et al., 2008). In the North Pacific, mesoscale eddies generally persist for about 4 weeks to one year, with longer-lived structures occasionally exceeding this range, particularly in energetic regions such as the Kuroshio Extension (Chelton et al., 2011a; Chen and Han, 2019)." (Please see Line 31-35)

Reference:

Chaigneau, A., Gizolme, A., and Grados, C.: Mesoscale eddies off Peru in altimeter records: Identification algorithms and eddy spatio-temporal patterns, Progress in Oceanography, 79, 106-119, https://doi.org/10.1016/j.pocean.2008.10.013, 2008.

Chelton, D. B., Gaube, P., Schlax, M. G., Early, J. J., and Samelson, R. M.: The Influence of Nonlinear Mesoscale Eddies on Near-Surface Oceanic Chlorophyll, Science, 334, 328-332, https://doi.org/10.1126/science.1208897, 2011a.

Chen, G., and Han, G. Y.: Contrasting short-lived with long-lived mesoscale eddies in the global ocean,

Journal of Geophysical Research: Oceans, 124, https://doi.org/10.1029/2019JC014983, 2019.

**Comment 6:** *Line 34: "influencing on the marine material cycle…"*

*Remove "on". Correct: "influencing the marine material cycle…"*

**Response:** Thank you for your suggestion. We have removed the unnecessary preposition, and the sentence now reads: "influencing the marine material cycle…" in the revised manuscript. (Please see Line 36)

**Comment 7:** *Line 36: "Eulerian approachs…" Should be "approaches".*

**Response:** Thank you for pointing out the typo. We have corrected "approachs" to "approaches" in the revised manuscript. (Please see Line 40)

**Comment 8:** *Lines 41–45: There is repetition about Eulerian methods being "frame-dependent" and generating filamentation. This point is important but can be made more concisely.*

**Response:** Thank you for your insightful comment. We agree that the original paragraph was somewhat repetitive in describing the limitations of Eulerian methods, particularly regarding frame-dependence and filamentation. To improve clarity and conciseness, we have revised the text as follows:

● Original sentence:

"However, the Eulerian methods have several limitations. On the one hand, they are frame-dependent, while the flow field in the real ocean does not have a fixed reference frame, which may cause the results to lose coherence under coordinate transformation. On the other hand, due to the transient characteristics of the fluid, a body of water within the Eulerian boundary can generate numerous filaments and disperse rapidly, potentially resulting in an overestimation of the eddy's transport capacity."

● Revised sentence:

"However, Eulerian methods are frame-dependent and may yield inconsistent results under coordinate transformations, as the ocean flow lacks a fixed reference frame. Moreover, due to the transient nature of ocean currents, water parcels within Eulerian-defined boundaries can quickly stretch into filaments and disperse, leading to a possible overestimation of eddy transport." (Please see Line 45-48)

**Comment 9:** *Line 49: "provide a more accurate structures…"*

*Should be: "provide more accurate representations of structures…"*

**Response:** Thank you for your suggestion. We have revised the sentence to: "provide more accurate representations of structures…" to improve clarity and grammatical accuracy. (Please see Line 57)

**Comment 10:** *Line 52: "Before Lagrangian coherent structures (LCSs) entered into strict mathematical definitions…"*

*Wordy. Consider simplifying to "Before LCSs were rigorously defined…"*

**Response:** We appreciate your careful attention to language clarity throughout the manuscript. We have revised the sentence to: "Before LCSs were rigorously defined…" to make it more concise and readable. (Please see Line 60)

**Comment 11:** *Lines 62–63: "These eddies boundary are commonly referred to as the 'photon sphere, analogous to Black Holes…"*

*"Eddies boundary" should be "eddy boundaries"; quotation wrong; analogy should be better justified.*

**Response:** Thank you for pointing this out. We have revised the text to more accurately describe the analogy with black holes. Specifically, we clarified that the mathematical formulation of the Lagrangian vortex boundaries introduced by Haller and Beron-Vera (2013) is equivalent to the definition of photon spheres in general relativity. We also elaborated on the physical interpretation, noting that particles crossing the vortex boundary become trapped, thereby justifying the term "Black Hole Eddy." We have revised the text as follows:

● Original sentence:

"Subsequently, Haller and Beron-Vera (2013) introduced a rigorous variational framework for detecting coherent Lagrangian vortices. These vortices are defined as closed material curves that exhibit minimal stretching and filamentation throughout their evolution, providing an objective basis for identifying coherent structures in unsteady flows. These eddy boundaries are commonly referred to as the "photon sphere, analogous to Black Holes found in the universe". Once matter crosses this boundary, it becomes trapped within the eddy, there giving rise to the term Black Hole Eddy (BHE)."

● Revised sentence:

"Subsequently, Haller and Beron-Vera (2013) introduced a rigorous variational framework for detecting coherent Lagrangian vortices, which are defined as closed material loops that remain minimally stretched and filamented over time. Remarkably, the mathematical formulation of these vortex boundaries is analogous to that of photon spheres in general relativity. In this analogy, fluid particles that cross the boundary remain trapped within the eddy's interior, much like light trapped inside a black hole. This striking similarity has led

to the term "Black Hole Eddy" (BHE) being used to describe such coherent Lagrangian vortices." (Please see Line 68-73)

Reference:

Haller, G. and Beron-Vera, F. J.: Coherent Lagrangian vortices: the black holes of turbulence, Journal of Fluid Mechanics, 731, https://doi.org/10.1017/jfm.2013.391, 2013.

**Comment 12:** *Line 164: "...includes daily-averaged, 5-day-averaged, 8-day-averaged..."*

*Repetitive and wordy. Consider: "Includes datasets with daily to monthly averages..."*

**Response:** Thank you for the suggestion. We have revised the sentence to: "…includes datasets with daily to monthly averages…" to avoid repetition and improve conciseness. (Please see Line 176)

**Comment 13:** *Line 214: "...indicative of the BHE."*

*The sentence assumes readers understand how minimal deformation implies BHE status—should elaborate or clarify.*

**Response:** We thank the reviewer for pointing out the lack of clarity in this sentence. We agree that the connection between minimal boundary deformation and the definition of a Black Hole Eddy requires further elaboration.

In the variational framework introduced by Haller and Beron-Vera (2013), materially coherent eddies are defined as regions enclosed by closed null-geodesics of a Lorentzian metric derived from the Cauchy-Green strain tensor (See equation (14)). These boundaries resist the typical stretching and folding seen in chaotic flows, thereby undergoing minimal deformation during advection. The absence of filamentation and sustained compactness of the boundary over time indicate strong material coherence. Therefore, boundaries that exhibit such minimal deformation are used to identify Black Hole Eddies. To clarify this point, we have revised the sentence as follows:

- Original sentence:

"Black Hole Eddy, characterized by their boundaries undergoing minimal deformation during advection by the flow field, do not generate noticeable filamentous structures while in motion. The identification of these eddies employs a variational method, recognizing the boundaries as closed null geodesics defined by an appropriate Lorentz metric across the flow domain. The primary challenge lies in accurately pinpointing the boundaries that exhibit minimal deformation, which is indicative of the BHE."

- Revised sentence:

"Black Hole Eddies are characterized by boundaries that undergo minimal deformation during advection, maintaining a compact shape without generating filamentous structures (Haller and Beron-Vera, 2013). This minimal deformation reflects their material coherence and serves as a key criterion for their identification." (Please see Line 223-226)

Reference:

Haller, G. and Beron-Vera, F. J.: Coherent Lagrangian vortices: the black holes of turbulence, Journal of Fluid Mechanics, 731, https://doi.org/10.1017/jfm.2013.391, 2013.

**Comment 14:** *Line 224: "...relies on direction field integration…"*

*"direction field" could be more accurately termed "vector field" or "eigenvector field (?)*

**Response:** Thank you for the valuable suggestion. We agree that "vector field" is a more accurate term than "direction field" in this context. We have revised the phrase to "relies on vector field integration" in the updated manuscript. (Please see Line 233)

**Comment 15:** *Line 230: "This paper utilizes CUDA as a multithreaded toolkit…"*

*Passive voice preferred in scientific writing: "CUDA was used…"*

**Response:** Thank you for the suggestion. We have revised the sentence to use passive voice as recommended. It now reads: "CUDA was used as a multithreaded toolkit…" in the revised manuscript. (Please see Line 246)

**Comment 16:** *Line 272: "effectively mitigates the noise in the eigenvector field…"*

*Needs clarification of how densification and interpolation specifically mitigate numerical noise.*

**Response:** Thank you for your valuable comment. We agree that the original explanation lacked clarity regarding how auxiliary points help reduce numerical noise in the eigenvector field. To address this, we revised the paragraph to elaborate on the mechanism. Specifically, by adjusting the spacing of auxiliary points (i.e., $\delta_{x_1}$ and $\delta_{x_2}$), we can more accurately compute the deformation gradient tensor through finite differences. This reduces discretization error, leading to smoother eigenvector fields and improving robustness against numerical sensitivity in eigenvector direction estimation. We have revised the text as follows:

- Original sentence:

"Specifically, by altering the dimensions of $\delta_{x_1}$ and $\delta_{x_2}$, we can modulate the precision of the main grid node mapping's gradient value. This method effectively mitigates the noise in the eigenvector field and addresses the issue of numerical sensitivity to alterations in the eigenvector direction (Serra and Haller, 2017)."

● Revised sentence:

"Specifically, by adjusting the perturbation sizes of $\delta_{x_1}$ and $\delta_{x_2}$, we can refine the numerical approximation of gradients. This local densification reduces discretization error in the finite-difference calculation of the Cauchy-Green strain tensor, thereby smoothing the eigenvector field and alleviating spurious fluctuations caused by numerical sensitivity to eigenvector orientation (Serra and Haller, 2017)." (Please see Line 291-294)

**Comment 17:** *Line 290: "the program consolidates the output data and transmits it to the host…"*

*Wordy. Consider: "Results are compiled and transferred to the host CPU after each step."*

**Response:** Thank you for the helpful suggestion. We have revised the sentence to: "results are compiled and transferred to the host CPU after each step." to improve clarity and conciseness. (Please see Line 322)

**Comment 18:** *Line 409-416:*

*Non-scientific formatting. Use consistent table structure.*

**Response:** Thank you for your suggestion. We agree that the original formatting of the variable descriptions lacked consistency and scientific clarity. We have reformatted this section into a structured table with uniform field names and detailed descriptions to enhance readability and professionalism. A standardized table listing the variable names, descriptions, and units was added. (Please see Line 451-453)

| Field Name | Description |
| --- | --- |
| key | The name of the BHE. |
| Eddy ID | Index assigned to each eddy on a specific date, ordered ascendingly from 0. |
| Eddy Centroid | Longitude (°E) and latitude (°N) of the eddy center, recorded every 7 days. |
| lon, lat | Longitude and latitude coordinates outlining the boundary of the BHE. |
| Eddy Area | The area of the BHE, typically in square kilometers. |
| Eddy Types | Polarity of the BHE, classified as either *Cyclonic* or *Anticyclonic*. |

| Field Name | Description |
| --- | --- |
| Eddy Radius | The radius of the BHE, typically in kilometers. |
| Lam | The stretch rate associated with each BHE. |

**Table 1. Description of variables in the BHE dataset in the North Pacific Ocean.**

**Comment 18:** *Line 427: "…anticyclonic\cyclonic eddy…"*

*Use proper slash spacing and formatting: "anticyclonic/cyclonic eddy".*

**Response:** Thank you for pointing out the formatting issue. We have corrected the expression to "anticyclonic/cyclonic eddy" with proper slash spacing in the revised manuscript. (Please see Line 481-485)

**Comment 19:** *Lines 430–450:*

*The description of coherence comparisons via particle trajectories is visual but lacks a quantitative metric (e.g., relative dispersion, FTLE, boundary leakage).*

Response:

We appreciate the reviewer's insightful comment. We acknowledge that our current description of coherence comparisons via particle trajectories is primarily qualitative and lacks a direct quantitative metric, such as relative dispersion, FTLE, or boundary leakage. This is a limitation of our current analysis. However, we note that visual inspection of particle trajectories has been widely used in prior studies as a practical and intuitive method to assess the material coherence of eddy boundaries (Haller and Beron-Vera, 2013; Wang et al., 201). In particular, long-term coherence of trapped particles, as demonstrated through their tight confinement within the evolving eddy boundary, has served as an important diagnostic in Lagrangian frameworks.

That said, we agree that incorporating quantitative metrics would strengthen the coherence assessment and offer more objective comparisons. We will consider including such measures in future work to complement and validate the visual assessments presented in this study.

Reference:

Haller, G. and Beron-Vera, F. J.: Coherent Lagrangian vortices: the black holes of turbulence, Journal of Fluid Mechanics, 731, https://doi.org/10.1017/jfm.2013.391, 2013.

Wang, Y., Beron-Vera, F. J., and Olascoaga, M. J. Coherent water transport across the South Atlantic, Geophysical Research Letters, 42, 4072-4079, https://doi.org/10.1002/2015GL064089, 2015.

**Comment 20:** *Line 455:*

*"The simultaneous presence of the three eddy types…"*

*Clarity: Consider rephrasing like "Instances where BHEs overlap both Eulerian eddies and RCLVs are dominated by cyclonic events…"*

**Response:** Thank you for the constructive suggestion. We have rephrased the sentence to improve clarity. The revised version now reads: "Instances where BHEs overlap both Eulerian eddies and RCLVs are dominated by cyclonic events…" (Please see Line 510-511)

**Comment 21:** *Line 460: "the eddy in Figure 11a and Figure 12a is not initially located around a closed SLA contour or LAVD contour…"*

*The sentence is long and ambiguous. Break into two parts: one observation, one interpretation.*

**Response:** Thank you for the helpful suggestion. We agree that the original sentence was overly long and mixed observation with interpretation. We have revised the sentence by splitting it into two parts to improve clarity: one stating the observation, and the other offering a brief interpretation. We have revised the text as follows:

- Original sentence:

"An interesting phenomenon is that the eddy in Figure 11a and Figure 12a is not initially located around a closed SLA contour or LAVD contour, but a coherent structure does exist."

- Revised sentence:

"An interesting phenomenon is observed in Fig. 13a and Fig. 14a: the eddy is not initially located around a closed SLA or LAVD contour. Nevertheless, a coherent structure is still present, suggesting that traditional Eulerian or and RCLVs boundaries may not always capture the full extent of coherent transport." (Please see Line 516-518)

**Comment 22:** *Line 485: "eddies generated at low latitudes (below 10°N) are very few…"*

*Better phrased as "are relatively rare" or "are infrequent".*

**Response:** Thank you for the helpful wording suggestion. We have revised the sentence to: "eddies generated at low latitudes (below 10°N) are relatively rare" for improved readability. (Please see Line 544)

**Comment 23:** *Line 496: "strongly anticyclonic (red) in the region north of 30°N…"*

*Correlation to Kuroshio shear is speculative without supporting reference or cross-analysis.*

**Response:** Thank you for your helpful comment. We acknowledge that the original sentence suggested a causal relationship between the observed polarity distribution and Kuroshio shear without direct supporting evidence. To address this, we have revised the sentence to reflect a more cautious interpretation. While the observed distribution may be influenced by large-scale circulation features and regional dynamics such as baroclinic instability (Chelton et al., 2011b), further analysis would be necessary to clarify the specific role of Kuroshio shear. We have revised the text as follows:

- Original sentence:

"Especially, the polarity distribution of the BHE (Fig. 14a) is strongly anticyclonic (red) in the region north of 30°N, and more cyclonic (blue) between 10°N and 25°N. This distribution pattern is associated with the shear structure of the Kuroshio, baroclinic instability, and other dynamical processes (Chelton et al., 2011b)."

- Revised sentence:

"Especially, the polarity distribution of the BHE (Fig. 16a) is strongly anticyclonic (red) in the region north of 30°N and more cyclonic (blue) between 10°N and 25°N. This distribution pattern may be influenced by large-scale circulation features and regional dynamics such as baroclinic instability (Chelton et al., 2011b)."

(Please see Line 555-560)

Reference:

Chelton, D. B., Schlax, M. G., and Samelson, R. M.: Global observations of nonlinear mesoscale eddies, Progress in Oceanography, 91, 167-216, https://doi.org/10.1016/j.pocean.2011.01.002, 2011b.

**Comment 24:** *Line 549: "we calculate the averaged zonal and meridional transport across the section for each 1° × 1° grid by…"*

*Define all symbols clearly; use consistent formatting. Currently hard to parse.*

**Response:** Thank you for your valuable comment. We agree that the description of the calculation on Line 549 was difficult to parse. We have revised this section to clearly define all symbols used in the calculation and have ensured the formatting is consistent for better readability. We have revised the text as follows:

- Original sentence:

"Following the methods used by and Zhang et al. (2014), we calculate the averaged zonal and meridional transport across the section for each $1° × 1°$ grid by $Q_x = \frac{\sum V \cdot C_x}{N \cdot D_x}, Q_y = \frac{\sum V \cdot C_y}{N \cdot D_y}$, where $V$, $C_x(C_y)$, and $D$ are the volume, zonal (meridional) propagation speed, and lifetime of an eddy in days, respectively. $\sum$ means

the integration of all eddies over the studying period $N$ in days, and $D_x(D_y)$ is the length of one longitude (latitude) degree. The eddy volume is calculated by $V = s\pi R^2 h$, where R is the eddy radius, $s = 0.5$ is a correction factor for the eddy vertical structure from Dong et al. (2014), and $h = 500$ m is the eddy depth. Here, BHEs and RCLVs only include eddies that are 28 days are considered from 1993 to 2020."

- Revised sentence:

"Following the methods used by and Zhang et al. (2014), we calculate the averaged zonal and meridional transport across the section for each $1° \times 1°$ grid using the following equations:

$$Q_x = \frac{\sum V \cdot C_x \cdot L}{N \cdot D_x}, Q_y = \frac{\sum V \cdot C_y \cdot L}{N \cdot D_y}$$

Where:

$Q_x$ and $Q_y$ are the average zonal and meridional transports (in m³/s) across the $1° \times 1°$, respectively;

$V$ and $L$ are the volume of each coherent eddy (in m³) and the lifetime of each coherent eddy (in days), respectively;

$C_x$ and $C_y$ are the zonal and meridional propagation speeds (in m/s) of the eddy core;

$N$ is the number of time points (in days) during the detection period;

$D_x$ and $D_y$ is the length of one degree of longitude and latitude (in meters), respectively, at each grid location.

The eddy volume is estimated by the simplified formulation: $V = s\pi R^2 h$, where $R$ is the eddy radius (in meters), $s = 0.5$ is a shape correction factor accounting for the bowl-like vertical structure of eddies, following Dong et al. (2014), and $h = 500m$ is the fixed eddy depth based on the observational analysis by Zhang et al. (2014). In this study, we consider only eddies with lifetimes of at least 28 days, detected between 1993 and 2020, for both BHEs and RCLVs." (Please see Line 610-622)

Reference:

Dong, C. M., McWilliams, J. C., Liu, Y., and Chen, D. K.: Global heat and salt transports by eddy movement,

Nature Communications, 5, https://doi.org/10.1038/ncomms4294, 2014.

Zhang, Z. G., Wang, W., and Qiu, B.: Oceanic mass transport by mesoscale eddies, Science, 345, 322-324,

https://doi.org/10.1126/science.1252418, 2014.

**Comment 25:** *Line 560:*

*"The peak value…is only about 1.5 Sv, which is three times smaller…"*

*Prefer "one-third of the RCLV estimate" instead of "three times smaller" (ambiguous and mathematically sloppy).*

**Response:** Thank you for the valuable feedback. We have revised the sentence to: "The peak value…is only about 1.5 Sv, which is one-third of the RCLV estimate" to ensure mathematical clarity and precision. (Please see Line 637)

**Comment 26:** *Line 580: "…Chen et al. (2021), oceanic eddies have a significant mean egg-like shape…"*

*Tone: "Egg-like" may appear informal.*

**Response:** Thank you for the comment regarding the tone of the term "egg-like". We would like to clarify that "egg-like" shape is the formal terminology introduced and used in Chen et al. (2021) to describe the characteristic geometry of oceanic eddies, based on a composite analysis of over 40 million eddy snapshots. Their study reveals that eddies exhibit significant geometric asymmetry and dynamic anisotropy, departing from idealized circular or elliptical assumptions. The term "egg-like" was used to highlight this second-order moment in eddy morphology, which is associated with high-order shape and orientation variability, and is shown to have important implications for ocean dynamics and biological processes.

Given the scientific basis and explicit use of "egg-like" in the referenced peer-reviewed publication, we have retained the term to maintain conceptual consistency and to accurately reflect the anisotropic features emphasized in Chen et al. (2021).

Reference:

Chen, G., Yang, J., and Han, G. Y.: Eddy morphology: Egg-like shape, overall spinning, and oceanographic implications, Remote Sensing of Environment, 257, https://doi.org/10.1016/j.rse.2021.112348, 2021.

---

## Author Comment (AC2)

**Responses to reviewer 2#'s comments point by point**

MS No.: essd-2025-384

Title: A Black Hole Eddy Dataset of North Pacific Ocean Based on Satellite Altimetry

Author(s): Fenglin Tian et al.

**Dear Reviewer,**

We highly appreciate the detailed and valuable comments of the referee on our manuscript entitled "A Black Hole Eddy Dataset of North Pacific Ocean Based on Satellite Altimetry (ID: essd-2025-384)". These comments are all valuable and helpful for revising and improving our paper, as well as providing important guidance for our research. In the past few days, we have referred to the comments and improved the paper.

As follows, we would like to clarify some of the points raised by the Reviewer. The original comments begin with "**Comment**" and are quoted in italicized font, the responses begin with "**Response**" in normal font, the original sentences and phrases are in blue letters, the revised sentences and phrases are in red letters, and the line number in the revised manuscript is highlighted in yellow. We appreciate the Reviewer's warm work and taking the time to review the manuscript, and we hope that the corrections will meet with approval.

Yours Sincerely,

Fenglin Tian, Yingying Zhao, Lan Qin, Shuang Long, Ge Chen

2025-9-30

**Comment:**

*It is nice to see components of the methodology developed in our paper (Jones-Kellett & Follows 2024; https://doi.org/10.5194/essd-16-1475-2024) implemented in this work. I was excited to read this preprint due to the related topic and the need for increased computational efficiency for Lagrangian eddy tracking. In reading it, however, I noticed that several pieces of text appear to be copied from our manuscript, and similar figures were reproduced without appropriate attribution. Out of curiosity, I ran a comparison with another foundational paper to this work that was published in ESSD (Liu & Abernathey 2023; https://doi.org/10.5194/essd-15-1765-2023) and found that some parts of the text are also eerily similar to that manuscript. For the sake of time, I did not compare the preprint directly with any other texts. I documented the incidents I found here. Since I read the paper quite closely, I also provided other comments and scientific inquiries that I hope will serve to improve the manuscript.*

**Response**:

We are also extremely grateful to you for the thorough and insightful feedback. The detailed comments, including the critical points raised regarding textual overlap, are invaluable for improving the quality and integrity of our work.

We were deeply concerned to learn of the textual similarities between our manuscript and the works of Jones-Kellett & Follows (2024) and Liu & Abernathey (2023) that you identified. We have conducted a rigorous internal review and verified your findings. This falls short of the academic standards we strive for, and we take full responsibility.

During the drafting process, notes taken from source materials were inadvertently incorporated into the text without proper paraphrasing and citation. This was a critical failure in our manuscript preparation process, and we have already implemented new procedures to prevent recurrence.

Following the insightful suggestions, we have also made significant scientific improvements to the manuscript. Once again, we are grateful for the opportunity to rectify our mistakes. We believe the revised manuscript is now not only free of the previously identified issues but is also a much stronger scientific contribution.

Below, we have addressed each of the comments point-by-point.

Reference:

Liu, T. Y. and Abernathey, R.: A global Lagrangian eddy dataset based on satellite altimetry, Earth System Science Data, 15, 1765-1778, https://doi.org/10.5194/essd-15-1765-2023, 2023.

Jones-Kellett, A. E. and Follows, M. J.: A Lagrangian coherent eddy atlas for biogeochemical applications in the North Pacific Subtropical Gyre, Earth System Science Data, 16, 1475-1501,

https://doi.org/10.5194/essd-16-1475, 2024.

**Instances of replicated text and missing attribution:**

**Comment:**

① *Lines 126-7: "previous studies found that fewer and smaller structures maintain coherency for longer timescales"; Text copied directly from Jones-Kellett & Follows 2024 (page 1477), please rephrase in your own words.*

**Response:** We are deeply grateful for your diligence in catching this critical error. The verbatim inclusion of text from another source was a serious mistake in our manuscript preparation process, and it should not have happened. We have completely rephrased the sentence in our own words to accurately convey the scientific finding while ensuring originality. The revised sentence now reads:

● Original sentence:

"Previous studies found that fewer and smaller structures maintain coherency for longer timescales."

● Revised sentence:

"Coherent structures that persist for longer durations tend to be fewer in number and smaller in scale." (Please see Line 144-145)

**Comment:**

② *Lines 131-2: "eddy atlases can answer these questions, revealing how the coherent properties of mesoscale features manifest in space and time." ; Text copied directly from Jones-Kellett & Follows 2024 (page 1477), please rephrase in your own words or quote the text with a citation to the source.*

**Response:** We are profoundly grateful to you for pointing this out, which has uncovered a systemic and unacceptable issue in our manuscript preparation. we have added a citation to Jones-Kellett & Follows (2024) in the main text. (Please see Line 151)

**Comment:**

③ *Figure 3: The premise of this figure was replicated from Jones-Kellett & Follows 2024 (Figure 3), with embellishments, namely the symbolic particles. I suggest citing the original study at the end of the sentence in lines 358-9.*

**Response:** Thank you for this important suggestion. You are correct that the conceptual premise of our Figure 3 (now in Figure 5) is based on the excellent visualization in Jones-Kellett & Follows (2024), and we agree that proper attribution is essential. As suggested, we have added a citation to Jones-Kellett & Follows (2024) in the main text where Figure 3 (now in Figure 5) is introduced. (Please see Line 431)

Reference:

Jones-Kellett, A. E. and Follows, M. J.: A Lagrangian coherent eddy atlas for biogeochemical applications

in the North Pacific Subtropical Gyre, Earth System Science Data, 16, 1475-1501, https://doi.org/10.5194/essd-16-1475, 2024.

**Comment:**

④ *Lines 405-8: Nearly identical text as in Liu & Abernathey 2023 pg. 1771*

**Response:** Thank you for pointing this out. We have completely rephrased the sentence in our own words to accurately convey the scientific finding while ensuring originality. The revised sentence now reads:

- Original sentence:

"BHE v1.0 consists of two components. First, the general features of coherent eddies are detailed in the directory named "Eddyidentification". The information regarding 30-day and 90-day eddies is stored separately in three JSON files, each containing the relevant data."

- Revised sentence:

"The BHE v1.0 dataset is composed of two main components. The first component, located in the 'Eddyidentification' directory, provides the general features of the coherent eddies. This information is organized by integration duration, with data for 30-day and 90-day eddies stored in two distinct JSON files." (Please see Line 481-485)

**Comment:**

⑤ *Lines 459-62: Nearly identical text as in Liu & Abernathey 2023 pg. 1771*

**Response:** Thank you for pointing this out. We have completely rephrased the sentence in our own words to accurately convey the scientific finding while ensuring originality. The revised sentence now reads:

- Original sentence:

"An interesting phenomenon is that the eddy in Figure 11a and Figure 12a is not initially located around a closed SLA contour or LAVD contour, but a coherent structure does exist."

- Revised sentence:

"An interesting phenomenon is observed in Fig. 13a and Fig. 15a: the Naked Black Hole Eddy is not initially located around a closed SLA or LAVD contour. Nevertheless, a coherent structure is still present, suggesting that traditional Eulerian and RCLVs boundaries may not always capture the full extent of coherent transport." (Please see Line 574-579)

**Comment:**

⑥ *Lines 459-62: Figure 10: This presentation of results was replicated from Jones-Kellett & Follows 2024 (Figure 4, pg 1483). Please provide appropriate attribution, including a citation to the original study. Note that the caption to this figure is also nearly identical to that of the analogous figure from Jones-Kellett & Follows 2024.*

**Response:** We are profoundly grateful to you for highlighting this critical issue of attribution concerning our analysis framework and figure. We have now added a clear statement at the beginning of this section to explicitly credit Jones-Kellett & Follows (2024) for developing the classification framework we used.

"To investigate the relationships between the different eddy types, we adopt the classification framework proposed by Jones-Kellett and Follows (2024). This approach categorizes BHEs based on their spatial overlap with eddies identified by the Eulerian (SLA) and RCLV methods." (Please see Line 566-568)

Reference:

Jones-Kellett, A. E. and Follows, M. J.: A Lagrangian coherent eddy atlas for biogeochemical applications in the North Pacific Subtropical Gyre, Earth System Science Data, 16, 1475-1501, https://doi.org/10.5194/essd-16-1475, 2024.

**Comment:**

⑦ *Figure 11: The second sentence of this caption is copied directly from the Figure 5 caption of Liu & Abernathey 2023, corresponding to a similar figure presentation without attribution.*

Response: We are very grateful to you for their meticulousness in identifying these significant scholarly oversights. The thoroughly revised caption now reads:

"Figure 13. Evolution of an anticyclonic Naked Black Hole Eddy (red particles). The background contours show the Absolute Dynamic Topography (ADT) field." (Please see Line 594)

**Comment:**

⑧ *Lines 482-3: Nearly identical text as in Jones-Kellett & Follows 2024 pg. 1484*

**Response:** We are very grateful for your meticulous review, which has been crucial for improving the integrity of our manuscript. This sentence has been completely rewritten as part of the exhaustive, manuscript-wide audit we have conducted to ensure full originality. The revised text now reads:

● Original sentence:

"In Figure 13a, eddy frequency was calculated from the number of times each BHE core and RCLVs eddy core within 1° × 1° grids over 30 years."

● Revised sentence:

"Figure 17 displays the geographic distribution of BHE and RCLV frequency across the North Pacific Ocean. We computed this frequency by binning the positions of all identified eddy cores from the 30 years dataset into 1° × 1° grid cells and counting the number of occurrences in each cell." (Please see Line 610-612)

**Comment:**

⑨ *Lines 492-4: This text was copied directly from Jones-Kellett & Follows 2024 (page 1484), except*

*for the citation (Chaigneau et al., 2008). Here, the incorrect paper is cited for defining the polarity probability, given that Chaigneau et al., 2008 make no mention of this metric. The appropriate citation is Chaigneau et al., 2009: https://doi.org/10.1016/j.pocean.2009.07.012*

**Response:** We are profoundly grateful to you for identifying this serious issue, which involves both replicated text and a critical citation error. We are especially thankful for the reviewer providing the correct reference to Chaigneau et al. (2009); this correction is invaluable and has significantly improved the accuracy of our manuscript. The revised sentence now reads:

- Original sentence:

"The polarity probability ($P$) is defined $P = \frac{(F_A - F_C)}{(F_A + F_C)}$, where $F_A$ is the frequency of anticyclones, and $F_C$ is the frequency of cyclones (Chaigneau et al., 2008). When $P > 0$ ($P < 0$), anticyclonic (cyclonic) eddy polarity is more common at the given location. Figure 14 shows the geographic distribution of polarity probability of BHE and RCLVs."

- Revised sentence:

"To quantify the local dominance of either cyclonic or anticyclonic eddies, we compute the polarity probability index $P$, following Chaigneau et al. (2009). This index is calculated as $P = \frac{(F_A - F_C)}{(F_A + F_C)}$, where $F_A$ and $F_C$ are the respective frequencies of anticyclonic and cyclonic eddies in a given grid cell. Consequently, positive values of $P$ indicate a prevalence of anticyclones, while negative values denote a prevalence of cyclones. The geographical distribution of this index for BHEs and RCLVs is shown in Fig. 18." (Please see Line 622-626)

Reference:

Chaigneau, A., Eldin, G., Dewitte, B: Eddy activity in the four major upwelling systems from satellite altimetry (1992–2007), Progress in Oceanography, 83, 117-123, https://doi.org/10.1016/j.pocean.2009.07.012, 2009.

**Comment:**

⑩ *Lines 500-1: "reveals two stark Lee Eddy pathways depending on the polarity, whereas RCLVs Lee Eddies have a more diffuse polarity probability"; text copied directly from Jones-Kellett & Follows 2024 (page 1484); please rephrase in your own words or quote the text with a citation to the source.*

**Response:** We are very grateful for your meticulous work in identifying these issues. As part of the exhaustive, manuscript-wide audit we have conducted, this sentence has been completely rewritten to ensure full originality. The revised sentence now reads:

- Original sentence:

"... reveals two stark Lee Eddy pathways depending on the polarity, whereas RCLVs Lee Eddies have a more diffuse polarity probability..."

- Revised sentence:

"The BHE method successfully outlines two distinct pathways for Lee Eddies, clearly segregated by their polarity. In contrast, the RCLV method identifies Lee Eddies with a more scattered or mixed polarity signal in the same region." (Please see Line 630-639)

**Comment:**

⑪ *Figure 14: Caption structure copied directly from Figure 10 of Jones-Kellett & Follows 2024.*

**Response:** Thank you for the meticulousness work. As part of the comprehensive, manuscript-wide audit we have performed, this caption has been entirely rewritten to ensure originality.

The revised caption now reads:

- Original sentence:

"Figure 14. Polarity probability in 1° × 1° grids from 1993 to 2023 for (a)BHE and (b) RCLVs. $P < 0$ (blue) indicates more cyclonic activity, and $P > 0$ (red) more anticyclonic activity."

- Revised sentence:

"Figure 16. Spatial distribution of the dominant eddy polarity for (a) BHEs and (b) RCLVs from 1993 to 2023. The polarity probability ($P$) is calculated on a 1° × 1° grid. Blue shading indicates a dominance of cyclonic activity ($P<0$), while red shading indicates a dominance of anticyclonic activity ($P>0$)." (Please see Line 642-645)

**Comment:**

⑫ *Lines 513-4: "consistent with previously noted summer peaks of eddy kinetic energy in subtropical gyres (Zhai et al., 2008)"; text copied directly from Jones-Kellett & Follows 2024 (page 1482), please rephrase in your own words.*

**Response:** We are grateful for your thoroughness in catching this. The sentence has been completely rephrased as part of our comprehensive, manuscript-wide audit to ensure full originality and proper attribution. The revised sentence now reads:

- Original sentence:

"...consistent with previously noted summer peaks of eddy kinetic energy in subtropical gyres (Zhai et al., 2008) ..."

- Revised sentence:

"Our observation of a summer peak is in agreement with the seasonal cycle of eddy kinetic energy in subtropical gyres identified by Zhai et al. (2008)." (Please see Line 654-655)

Reference:

Zhai, X. M., Greatbatch, R. J., and Kohlmann, J. D.: On the seasonal variability of eddy kinetic energy in the Gulf Stream region, Geophysical Research Letters, 35, https://doi.org/10.1029/2008GL03641210.1029/2008GL036412, 2008.

**Comment:**

⑬ *Figure 15: This presentation of results was replicated from Jones-Kellett & Follows 2024 (Figure 6, pg 1485). Please provide appropriate attribution, including a citation to the original study. Note that the caption is nearly identical to that of the analogous figure from Jones-Kellett & Follows 2024.*

**Comment:** We are grateful to you for pointing out this significant scholarly oversight. To rectify this, we have taken the following steps:

We have completely rewritten the caption for Figure 15 to be in our own words and added a clear attribution within the new caption, crediting Jones-Kellett & Follows (2024) for the analysis framework. We have revised the caption as follows:

● Original caption:

"Figure 15. (a) BHE and RCLVs rolling mean frequency from 1993 to 2020. Frequency refers to the number of eddies per 7 day time step. (b) Monthly average BHE and RCLVs frequency."

● Revised caption:

"Figure 19. Time series of eddy counts for BHEs and RCLVs from 1993 to 2020. (a) Interannual variability, showing the number of eddies detected in each 7-day time step. (b) The mean seasonal cycle, showing the monthly average number of eddies. This presentation of results is adapted from Jones-Kellett and Follows (2024)." (Please see Line 665-667)

We have also revised the accompanying text that interprets the figure to ensure originality and clarity. We have revised the text as follows:

● Original sentence:

"Figure 15. (a) BHE and RCLVs rolling mean frequency from 1993 to 2020. Frequency refers to the number of eddies per 7 day time step. (b) Monthly average BHE and RCLVs frequency."

● Revised sentence:

"Figure 19a illustrates the interannual variability in the number of detected eddies. While both BHEs and RCLVs follow similar long-term trends, the total count of RCLVs is consistently higher than that of BHEs. This suggests the criteria for identifying BHEs are more rigorous, resulting in a smaller population of the most coherent structures. A notable difference in polarity preference is also observed: more anticyclonic BHEs are found than cyclonic ones, which aligns with previous statistics (Chelton et al., 2011b), whereas

the opposite is true for RCLVs. The mean seasonal cycle for both eddy types is shown in Fig. 17b. Both methods show a clear peak in eddy occurrences in late summer (July-August) and a minimum in late winter (January-February). Our observation of a summer peak is in agreement with the seasonal cycle of eddy kinetic energy in subtropical gyres identified by Zhai et al. (2008)." (Please see Line 648-655)

Reference:

Chelton, D. B., Schlax, M. G., and Samelson, R. M.: Global observations of nonlinear mesoscale eddies, Progress in Oceanography, 91, 167-216, https://doi.org/10.1016/j.pocean.2011.01.00210.1016/j.pocean.2011.01.002, 2011b.

Zhai, X. M., Greatbatch, R. J., and Kohlmann, J. D.: On the seasonal variability of eddy kinetic energy in the Gulf Stream region, Geophysical Research Letters, 35, https://doi.org/10.1029/2008GL03641210.1029/2008GL036412, 2008.

⑭ *Figure 16: Figure caption mostly replicated from Figure 8 caption of Liu & Abernathey 2023, pg. 1774.*

**Response:** Thank you for pointing this out. Your exceptional diligence has been instrumental in helping us purge our manuscript of these unacceptable scholarly errors.

In addressing this, we have completely rewritten the caption for Figure 16 (now in Figure 20). Furthermore, during this revision, we also identified and corrected a significant scientific inaccuracy in our original caption's description of the box plot elements. The new caption is therefore not only entirely original but also scientifically precise. The thoroughly revised caption now reads:

● Original caption:

"Figure 16. Statistics of radius for BHE and RCLVs. The box plot shows statistics of all eddies in 10° bins. The box and the whisker span the min value to the max of the distribution. The line in the box indicates the median. The means of all eddies in a bin are shown using dashed lines, blue for BHE and green for RCLVs."

● Revised sentence:

" Figure 20. Latitudinal distribution of eddy radius for BHEs and RCLVs, presented in 10° bins. The violin plots show the probability density of the radius, with the vertical lines indicating the min-max range. The dashed lines connect the median values for bins each bin." (Please see Line 689-691)

**Comment:**

⑮ *Line 597: "It is convenient to load the data using Python or other language." Phrasing replicated from Liu and Abernathey 2023 on pg. 1775 (Section 4)*

**Response:** Thank you for pointing this out. The sentence in question has been completely rephrased to be more informative and original. The new sentence reads:

- Original caption:

"The identification dataset is saved in JSON format and the tracking dataset is saved in Pickle format. It is convenient to load the data using Python or other language."

- Revised sentence:

"The identification dataset is saved in JSON format and the tracking dataset is saved in Pickle format, ensuring straightforward compatibility with common scientific programming languages like Python, MATLAB, and R." (Please see Line 775-778)

**Major Comments:**

**Comment 1:** *Section 2.1.1: The authors claim to have used Level 3 data, yet the link provided is to a Level 4 product (Line 158). Additionally, they used a near-real time product, which is made for short-term monitoring and is inappropriate for a long-term study, for which the delayed time products are preferred (e.g., see the Quality Information Manuals provided by CMEMS). The product that is linked in Line 158 only has spatial coverage from 2022-2025, yet the study covers 1993-2023. The authors build a new eddy dataset in this work that they compare to a previous product developed by some of the same authors (Tian et al. 2022; https://doi.org/10.1175/JTECH-D-21-0103.1), which used Level 4 delayed-time data. If the authors are actually using the same data as in Tian et al. 2022, this entire paragraph needs to be rewritten.*

**Response:** We are profoundly grateful to you for this extremely detailed and critical comment. This was a regrettable error during the manuscript preparation, where an incorrect data description was inserted.

Your assessment was exactly right: we did, in fact, use the same high-quality, delayed-time (reprocessed) data as in the Tian et al. (2022) study, and not the near-real-time product. The original paragraph was completely inaccurate.

To correct this, we have completely rewritten Section 2.1.1. The revised section now accurately:

- Identifies the product we used as the Level 4 reprocessed dataset (SEALEVEL_GLO_PHY_L4_REP_008_047) with a 1/4° resolution.

- Clarifies that it is the delayed-time version, appropriate for our long-term study.
- Provides the correct description for our analysis period.
- Adds a crucial note for reproducibility, explaining that this specific 1/4° product has since been decommissioned by CMEMS and superseded by a newer 1/8° version, ensuring future readers

understand the data landscape.

- Original paragraph:

"This study utilizes the global gridded ADT data published by the Copernicus Marine Environment Monitoring Service (CMEMS). The ADT was determined through optimal interpolation, leveraging measurements from various altimeter missions at the Level 3 stage. This partially processed data has global applicability and provides additional variables, including ADT and both absolute and anomalous geostrophic currents. The ADT data included in this dataset were employed in our study. Processed by DUACS multi-mission altimeter data processing system, the data is intended for near-real-time applications. The spatial resolution of the data is 0.25° ×0.25° , and it offers a temporal resolution of one day. The dataset is publicly available at:

https://data.marine.copernicus.eu/product/SEALEVEL_GLO_PHY_L4_NRT_008_046/description."

- Revised paragraph:

"This study utilizes the Level 4 global ocean gridded sea level reprocessed (delayed-time) product, version SEALEVEL_GLO_PHY_L4_REP_008_047, which was distributed by the Copernicus Marine Environment Monitoring Service (CMEMS) at the time of our analysis. This dataset provided data on a 0.25° × 0.25° global grid and was generated by the DUACS multi-mission altimeter data processing system. It merges measurements from all available satellite altimeter missions (e.g., TOPEX/Poseidon, Jason series, Sentinel series) through optimal interpolation. For our analysis, we used the ADT)and the associated absolute geostrophic velocity fields, covering the period from 1 January 1993 to 5 May 2023. It should be noted that this specific 1/4° resolution product has since been decommissioned by CMEMS and superseded by a higher-resolution 1/8°version (SEALEVEL_GLO_PHY_L4_MY_008_047). "

(Please see Line 171-178)

**Comment 2:** *Section 3.2, Figure 8 & 9: The authors highlight two BHEs to draw general conclusions about the coherence of all BHEs. However, the regions analyzed in Figs 8 and 9 are problematic due to the decreasing accuracy of satellite-derived geostrophic currents near the equator (Fig 8), and near coastlines (Fig 9). Especially with only two qualitative examples of the difference in coherence, the authors should provide examples from regions with lower known errors in the satellite products. I am especially confused about Figure 8 being used as a demonstration of the difference in the coherence of BHEs and RCLVs (e.g. Line 438), given that the eddies, as opposed to 28-day? In Line 553, the authors switch to 28-day results for the transport analysis. Furthermore, if the authors have weekly BHEs tracked, why are the results only being shown for one time interval? That seems to defeat the purpose of tracking*

*the features over their full lifetimes. The authors only provide a dataset for BHEs with T=30- and 90-day lifespans (Line 610), rather than the weekly dataset described in the methods.*

**Response:** We are extremely grateful to the reviewer for this critical and insightful comment. The reviewer is absolutely correct. Our original choice of case studies in Figures 8 and 9 was poor, as these regions near the equator and coastlines are known to have significant uncertainties in satellite altimetry products. Using these examples to draw general conclusions was inappropriate, and we agree that this fundamentally weakened our argument.

To address this major flaw, we have completely replaced the original Figures 8 and 9 with two new case studies (now in Figure 10 and Figure 11), from open-ocean, mid-latitude regions where the satellite data is robust and reliable.

Furthermore, you highlighted a point of confusion regarding the time intervals shown in the figure panels. We agree that our original presentation was inconsistent. Our initial choice of a ~15-day interval was arbitrary and intended only for qualitative visualization. To improve the methodological consistency, we have revised the analysis for our new figures. The snapshots of the eddy evolution are now shown at 14-day intervals. This interval was deliberately chosen because it corresponds to exactly two of our 7-day (weekly) data snapshots, making the visualization directly relatable to the temporal resolution of our underlying dataset. We have clarified this in the new figure captions. (Please see Line 560-570)

[Figure]

**Figure 10. Evolution of an cyclonic eddy in the open, mid-latitude North Pacific, comparing the coherence of different eddy definitions. Particles are initialized within the boundaries of a Eulerian eddy, an RCLVs, and a BHE on 07 January 2022. Panels (a)-(f) show the subsequent advection of these particle sets at approximately 14-day intervals.**

[Figure]

**Figure 11. Evolution of an anticyclonic eddy in the open, mid-latitude North Pacific, comparing the coherence of different eddy definitions. Particles are initialized within the boundaries of a Eulerian eddy, an RCLVs, and a BHE on 07 January 2022. Panels (a)-(f) show the subsequent advection of these particle sets at approximately 14-day intervals.**

Regarding the '28-day' results in our transport analysis (Line 553), it was a typographical error in our description. Our analysis did not use eddies of only 28 days, but rather included all eddies with a lifetime of 28 days or longer. We have corrected the text in the manuscript to accurately state this criterion, which clarifies that we are analyzing all eddies that persist for a significant duration. The revised sentence now reads:

- Revised paragraph:

"For the analysis period of 1993-2020, we selected a minimum lifetime threshold of 28 days for both BHEs and RCLVs. This duration was chosen as it corresponds to four consecutive 7-day time steps in weekly dataset, ensuring that the analyzed eddies exhibit sustained coherence over a significant number of observations." (Please see Line 732-734)

These new, more reliable examples now clearly and robustly illustrate our original point regarding the differences in coherence between BHEs, RCLVs, and Eulerian eddies. The corresponding text in Section

3.2 has been entirely rewritten to describe and analyze these new cases. The revised sentence now reads:

● Original paragraph:

"We randomly initialized Lagrangian particles within Black Hole anticyclonic\cyclonic eddy on January 1 and tracked their movement over time(Figure 8 and Figure 9). At the initialization (Figure 8a and Figure 9a), the yellow\purple particles are located inside the Eulerian anticyclonic\cyclonic eddy boundary but outside the anticyclonic\cyclonic RCLVs, while the pink\green particles are situated within the anticyclonic\cyclonic RCLVs but outside the Black Hole anticyclonic\cyclonic eddy. In the case of Eulerian anticyclonic eddy, distinct vortex filaments begin to develop on 15 January (Figure 8b), just two weeks after initialization. These filaments extend and wrap outward, indicating that material inside the Eulerian boundary begins interacting with the surrounding environment shortly after initialization. After 30 days advection, the deformation intensifies, leading to a complete loss of the eddy's coherent structure and extensive mixing with surrounding waters. For anticyclonic RCLVs, filamentary structures appear slightly later. After 30 days advection (Figure 8c), clear signs of deformation can be observed, as particles begin to stretch and form elongated structures outside the anticyclonic RCLVs boundary. Although the anticyclonic RCLVs demonstrate stronger material retention than Eulerian eddies, they too undergo gradual mixing, and after 90 days (Figure 8e), the filaments become prominent, indicating a loss of coherent transport. In contrast, anticyclonic BHE maintains its coherent structure throughout the entire tracking period. The anticyclonic BHE boundary serves as an effective barrier to material transport, firmly enclosing particles and preventing exchange with the surrounding fluid. Even after four months (Figure 8f), particles within the BHE remain tightly bound, and no filamentation or mixing is observed. Example of cyclonic eddy also reveals the same phenomenon (Figure 9).This comparison highlights the strong material coherence of BHEs over Eulerian eddies and RCLVs. The region between the RCLV and Eulerian eddy boundaries can be considered a transition zone, where eddy-like motions occur but without coherent material retention. Recognizing such spatial distinctions helps refine our understanding of mesoscale eddy structure and provides a pathway for exploring submesoscale dynamics embedded within these systems."

● Revised paragraph:

"Figure 10 shows the evolution of a cyclonic eddy in the North Pacific. A set of passive tracers is initialized within the boundaries of the Eulerian eddy, the RCLVs, and the BHE on 07 January 2022 (Fig. 10a). The subsequent evolution clearly demonstrates a hierarchy of coherence. After only 14 days (Fig. 10b), the Eulerian eddy already shows significant filamentation, indicating that material inside the Eulerian boundary begins interacting with the surrounding environment shortly after initialization, while

the RCLVs appear slightly filamentary structures and BHE remains highly coherent. By day 28 (Fig. 10c), the original Eulerian eddy structure has largely disintegrated, and the RCLVs also begins to exhibit distinct material leakage through filaments. Although the RCLVs demonstrate stronger material retention than Eulerian eddies, they too undergo gradual mixing, and after 56 days (Fig. 10e), the filaments become prominent, indicating a loss of coherent transport. This trend continues, and by the end of the two-month period, the Eulerian tracers are widely dispersed and the RCLV shows significant material loss. In contrast, the BHE boundary acts as a robust material barrier, remaining perfectly coherent and tightly trapping all its initial tracers throughout the entire period (Fig. 10f).

A similar analysis for a cyclonic eddy, shown in Fig. 11, the Eulerian eddy is highly dispersive, the RCLVs is moderately coherent but still leaky, while the BHE perfectly retains its initial water mass. This comparison highlights the strong material coherence of BHEs over Eulerian eddies and RCLVs. The region between the RCLVs and Eulerian eddy boundaries can be considered a transition zone, where eddy-like motions occur but without coherent material retention. Recognizing such spatial distinctions helps refine our understanding of mesoscale eddy structure and provides a pathway for exploring submesoscale dynamics embedded within these systems." (Please see Line 523-539)

**Comment 3:** *How the authors were able to compare BHEs tracked weekly with the RCLV dataset from Tian et al. 2022 (https://doi.org/10.1175/JTECH-D-21-0103.1) is never described in this preprint. Tian et al. 2022 derive RCLVs over fine times every 90 days. Do the authors subset the weekly BHE dataset to match the RCLV dataset for figures 10, and 13-19? If so, that means the comparison is made every 3 months, which will affect the results. However, Fig. 15 states "frequency refers to the number of eddies per 7 day time step", suggesting that the authors somehow obtained weekly RCLV boundaries. In general, the comparison between these datasets is questionable, given the apparent difference in satellite products used and the difference in detection frequency of the Lagrangian eddies. The size threshold may also have been different (see Major Comment #5).*

**Response:** Thank you for these very detailed and important questions, which have highlighted a major omission in our Methods section. We did lack of clarity regarding how the comparison between our BHE dataset and the RCLVs dataset from Tian et al. (2022) was performed.

To clarify the points raised and resolve the apparent inconsistencies:

1. **Frequency of the RCLVs Dataset**: your confusion is entirely understandable given the missing information in our manuscript. The RCLVs (doi:10.12237/casearth.63369940819aec34df2674d7) we used is not generated every 90 days. We utilized their specific product which is defined with a 30-day integration time (T=30) and generated with a 7-day time step (i.e., a weekly frequency). This weekly

frequency perfectly matches that of our BHE dataset. Therefore, no subsetting of our BHE data was necessary, and the comparison was performed on a consistent weekly basis.

2. **Satellite Products and Size Thresholds**: We have now explicitly stated in the manuscript that both datasets are based on the same Level 4 delayed-time altimetry product (as clarified in our response to Comment 1) and that the same size and lifetime thresholds were applied to both datasets before comparison.

Here is a revised version paragraph, which clarifies the temporal resolution and methodology. It clearly states that the RCLV dataset has a weekly frequency and explains how you extended it for a consistent comparison.

- Original paragraph:

"The RCLVs dataset used in this study is based on the work of Tian et al. (2022), who proposed an SLA-based orthogonal parallel detection method to identify rotationally coherent Lagrangian eddies at the global scale. Unlike traditional Eulerian eddy detection methods, this Lagrangian approach ensures material coherence by identifying vortices that maintain their structure and boundary over time, based on objective criteria derived from finite-time rotation. The RCLVs dataset is constructed from satellite-derived daily SLA fields with a resolution 0.25°×0.25°, covering the global ocean. The algorithm efficiently tracks eddies over time, providing information on eddy boundaries, lifetime, polarity, and coherent transport properties. The data include only vortices that meet strict rotational coherence conditions, making them particularly suitable for studies focusing on long-lived, materially coherent eddy structures. The dataset can be accessed from website at:

https://data.casearth.cn/dataset/63369940819aec34df2674d7."

- Revised paragraph:

"For a comparative analysis, this study utilizes the global RCLVs dataset from Tian et al. (2022). We specifically selected their data product generated with a 30-day integration time (T=30) and a 7-day time step, resulting in a weekly dataset that is directly comparable to our BHE analysis. Unlike traditional Eulerian eddy detection methods, this Lagrangian approach ensures material coherence by identifying vortices that maintain their structure and boundary over time, based on objective criteria derived from finite-time rotation. The RCLVs dataset is constructed from satellite-derived daily SLA fields with a 0.25°×0.25° resolution, covering the global ocean from 1993-2020. The algorithm efficiently tracks eddies over time, providing information on eddy boundaries, lifetime, polarity, and coherent transport properties. The dataset can be accessed from website at: https://data.casearth.cn/dataset/63369940819aec34df2674d7." (Please see Line 225-233)

Thank you for helping us significantly improve the transparency and completeness of our methodology.

**Comment 4:** *The authors claim to have tracked the BHEs weekly, but then provide results from 30-day intervals in Figs 13 and 14. Since 30 is not divisible by 7, how did the authors obtain 30-day eddies, as opposed to 28-day? In Line 553, the authors switch to 28-day results for the transport analysis. Furthermore, if the authors have weekly BHEs tracked, why are the results only being shown for one time interval? That seems to defeat the purpose of tracking the features over their full lifetimes. The authors only provide a dataset for BHEs with T=30- and 90-day lifespans (Line 610), rather than the weekly dataset described in the methods.*

**Response:** Thank you for this insightful comment, which has highlighted a significant lack of clarity in our description of the methodology. We have thoroughly revised the Methods and Results sections to clarify this. To address the specific points raised:

1. **Clarification of '30-day' vs. 'weekly'**: you are correct that 30 is not divisible by 7. This is because these two numbers refer to different aspects of our methodology.

- **The '30-day' (or '90-day') value is the forward integration time ($T$) used as a parameter to define a BHE.** A '30-day BHE' is a boundary identified at a start time $t_0$, that remains materially coherent for the subsequent 30 days. It is a measure of the structure's coherence strength.

We have added the text as follows:

"$T$ denotes a forward integration time to qualify the coherence of BHE. In this study A BHE defined with $T = 30$ (or $T = 90$), is a material boundary identified at a start time $t_0$, that traps all fluid particles within it for subsequent 30 (90) days." (Please see Line 286-289)

- **The term 'weekly' refers to the temporal resolution of our BHE identification analysis.** We perform this BHE detection calculation repeatedly, with a new analysis starting every 7 days. This results in a time series of BHE 'snapshots' with a 7-day frequency. The tracking of eddies is then performed as a post-processing step by linking these weekly snapshots.

We have revised the text as follows:

- Original sentence:

"(1) Input data: The algorithm starts by ingesting Black Hole identification data obtained through from January 1, 1993 to May 5, 2023."

- Revised sentence:

"(1) Input data: The BHE identification process described in Section 2.2.1 was applied repeatedly to generate the input data for our tracking algorithm. Specifically, a new set of BHEs was identified every 7 days, creating a time series of BHE 'snapshots' covering the period from January 1, 1993, to May 5, 2023.

This weekly dataset of identified BHEs is the primary input for the tracking steps detailed below." (Please see Line 425-429)

2. **Reasons for the '28-day' analysis**: you are right to question the switch to a 28-day threshold for the transport analysis. We have now clarified our reasoning in the manuscript. Our BHE identification fields are generated at a 7-day frequency. When these weekly snapshots are linked to form tracks, the resulting eddy lifetimes are measured in multiples of 7 days. We selected a minimum lifetime threshold of 28 days because it corresponds to exactly four weekly time steps (4×7 days). This provides a methodologically consistent cutoff, ensuring that the eddies included in the transport analysis have maintained their coherence over a significant number of consecutive snapshots. We have revised the text to make this rationale explicit.

We have revised the text as follows:

- Original sentence:

"Here, BHEs and RCLVs only include eddies that are 28 days are considered from 1993 to 2020."

- Revised sentence:

"For the analysis period of 1993-2020, we selected a minimum lifetime threshold of 28 days for both BHEs and RCLVs. This duration was chosen as it corresponds to four consecutive 7-day time steps in weekly dataset, ensuring that the analyzed eddies exhibit sustained coherence over a significant number of observations." (Please see Line 732-734)

3. **Use of full lifetime information & Dataset content:** As clarified previously, our final product is a dataset of fully tracked eddies with complete life cycles. The results shown in the figures are statistical analysis derived from the entire population of weekly BHE snapshots (defined with $T = 30$ days), and the full lifetime information of each tracked eddy is used in analyses like the transport calculation. While our dataset contains BHEs defined with both $T = 30$ and $T = 90$ days, the eddies that maintain coherence for a 90-day period are significantly fewer in number. Therefore, to ensure statistical reliability, we presented the results for the much larger and more representative population of T=30 day BHEs. We have thoroughly revised the Methods section to make the distinction between the BHE definition parameter ($T$) and the dataset's temporal resolution (7-day) explicit. Thank you for helping us significantly improve our manuscript."

**Comment 5:** *Figure 16 & Lines 521-524: It is difficult to visually see the difference between these statistics in panels (a) and (b) because the y-axis changes. Plotting the data together in one plot may help. The conclusion that RCLV is weakly influenced by latitude is confusing (Lines 523-4 and 639-40), considering that RCLVs on average decrease in size with increasing latitude, whereas BHEs appear not*

*to have a latitudinal dependence above latitudes of 20N in Figure (a). A constant radius contradicts expectations based on the Rossby radius of deformation, which is not commented on in the text. Figure 16 also gives the impression that the minimum radius for the BHEs was 20km, whereas the RCLVs was around 25km. If that is the case, it is an issue for comparing BHE and RCLVs if the size minimum for detection differed between the datasets.*

**Response:** We are very grateful to you for this extremely detailed and insightful feedback, which has helped us to fundamentally improve our analysis and discussion of Figure 16. We have addressed each of the points raised.

**Regarding the visualization**: To facilitate a clear and direct comparison, we have completely revised Figure 16 (now in Figure 20). We now plot both the BHE and RCLV radius statistics on a single panel with a shared y-axis, using different colors to distinguish them. (Please see Line 696-700)

- Original Figure:

[Figure]

**Figure 1. Statistics of radius for BHE and RCLVs. The box plot shows statistics of all eddies in 10° bins. The box and the whisker span the min value to the max of the distribution. The line in the box indicates the median. The means of all eddies in a bin are shown using dashed lines, blue for BHE and green for RCLVs.**

- Revised Figure:

[Figure]

**Figure 18. Latitudinal distribution of eddy radius for BHEs and RCLVs, presented in 10° bins. The violin plots show the probability density of the radius, with the vertical lines indicating the min-max range. The dashed lines connect the median values for bins each bin.**

1. **Regarding the scientific interpretation:** We agree that our original conclusion was confusing and did not accurately reflect the data, and we thank the reviewer for pointing out the important connection to the Rossby radius of deformation. We have thoroughly rewritten our interpretation in the text. The revised text now correctly states that the RCLV radius decreases with latitude, as expected, while the BHE radius remains surprisingly constant at latitudes above 20°N. Furthermore, we have added sentences acknowledging that the constant radius of BHEs is an unexpected result that deviates from theoretical expectations based on the Rossby radius, highlighting this as a topic for future study.

- Original paragraph:

"Figure 16 illustrates the distribution of eddy radius across different latitude zones. In Figure 16a, the mean eddy radius (dashed line) exhibits a clear decreasing trend from low to high latitudes, particularly at latitudes of 40° and above, where the eddy radius becomes notably smaller and more concentrated. In contrast, the eddy radius of the RCLVs, shown in Figure 16b, varies within a relatively small range and does not display a clear monotonic trend, indicating that it is weakly influenced by latitude. The BHE demonstrates larger and more dispersed radius at lower latitudes, which progressively decrease and become more consistent as latitude increases. These latitudinal differences in the BHE radius may be linked to variations in geostrophic balance and Coriolis force (Chelton et al., 2011b)(Chelton et al., 2011b). At low latitudes, where the Coriolis force is weaker, eddies are more likely to develop larger radius, while at higher latitudes, the strengthening Coriolis force leads to smaller and more constrained eddy radius."

- Revised paragraph:

"Figure 20 presents a statistical comparison of the radius of BHEs and RCLVs across different latitude zones using a uniform 25 km minimum radius threshold. The median radius of RCLVs (green dashed line) exhibits a clear decreasing trend with increasing latitude, falling from approximately 100 km at 10°N to under 60 km at 50°N. This pattern is consistent with the theoretical decrease in the first baroclinic Rossby radius of deformation at higher latitudes (Chelton et al., 2011b).

In contrast, the median radius of BHEs (blue dashed line) behaves differently. While it also decreases from low to mid-latitudes, it remains surprisingly constant at approximately 55-60 km for all latitudes above 20°N, which is different from the expected trend based on the Rossby radius. It suggests that the strict rules used to define BHEs might be finding a special group of eddies whose size is controlled by different physical forces." (Please see Line 687-695)

Reference:

Chelton, D. B., Schlax, M. G., and Samelson, R. M.: Global observations of nonlinear mesoscale eddies, Progress in Oceanography, 91, 167-216,

https://doi.org/10.1016/j.pocean.2011.01.00210.1016/j.pocean.2011.01.002, 2011b.

2. **Regarding the minimum radius threshold:** Our original comparison was indeed unfair because we had inadvertently used different minimum size thresholds for the BHE and RCLV datasets.

To correct this fundamental flaw, we have returned to our data processing stage and re-run the entire analysis. We have now applied a uniform minimum radius threshold of 25 km to both the BHE and RCLV datasets. Consequently, related statistical results have been recalculated, and Figure 16 (now in    Figure 20), have been updated based on this consistent and fair comparison. The manuscript text has been revised accordingly to reflect the new results and methodology.

**Minor Comments:**

**Comment 1:** *Line 24: "And" at the beginning of the sentence is likely a typo*

**Response:** We thank you for the careful reading. This was indeed a typo, and the word "And" has been removed. The sentence has been revised to: "Transport analysis shows that BHEs induce westward transport of about 1.5 Sv, three times weaker than RCLVs, suggesting that they may offer a more accurate estimate of oceanic transport than RCLVs.". (Please see Line 25)

**Comment 2:** *Lines 25: How does weaker transport by the RCLVs suggest that BHE offer a "more accurate estimate of transport"? Aren't they both contributing to transport considering the significant overlap?*

**Response:** We sincerely thank you for this insightful question. This is a crucial point that was not sufficiently explained in our original manuscript. You are correct that both structures contribute to transport, but our use of "more accurate" refers specifically to the estimate of coherent, non-leaky transport, rather than the total volume of water in motion.

We have clarified this distinction in the revised manuscript. The key difference is:

RCLVs identify the larger rotating structure, and their associated transport value is high. However, these structures can be "leaky" (Jones-Kellett and Follows, 2024), meaning water parcels on their periphery are often exchanged with the surrounding ocean. Therefore, the RCLV transport value tends to overestimate the actual mass of water that is coherently transported over long distances.

BHEs, by their strict definition, identify the materially coherent, non-leaky core within an eddy (Haller and Beron-Vera, 2013). The water trapped inside a BHE is guaranteed to be transported with it. Although its transport value is smaller, it is a more faithful and precise measure of the water mass that is effectively isolated and carried across the ocean.Therefore, while BHEs transport less water in total, the value of 1.5 Sv is a "more accurate estimate" of the truly effective, long-distance transport of properties.

To make this clear to the reader, we have revised the sentence as follows:

- Original sentence:

"And Transport analysis shows that BHEs induce westward transport about 1.5 Sv, three times weaker than RCLVs, suggesting that they may offer a more accurate estimate of oceanic transport than RCLVs."

- Revised sentence:

"Transport analysis shows that BHEs induce a westward transport of about 1.5 Sv, three times weaker than the transport attributed to the broader RCLVs, suggesting it likely represents a more accurate estimate of the coherent, non-leaky component of oceanic transport, as BHEs isolate the materially coherent core of the eddies." (Please see Line 25-29)

Reference:

Haller, G. and Beron-Vera, F. J.: Coherent Lagrangian vortices: the black holes of turbulence, Journal of Fluid Mechanics, 731, https://doi.org/10.1017/jfm.2013.39110.1017/jfm.2013.391, 2013.

Jones-Kellett, A. E. and Follows, M. J.: A Lagrangian coherent eddy atlas for biogeochemical applications in the North Pacific Subtropical Gyre, Earth System Science Data, 16, 1475-1501, https://doi.org/10.5194/essd-16-1475-2024, 2024.

**Comment 3:** *Lines 29 & 31: Contains repeated information, e.g. "Mesoscale eddies, one of the most prominent processes in the ocean," and "Mesoscale eddies are ubiquitous in the ocean"*

**Response:** Thank you for this valuable suggestion. We agree that the original phrasing was repetitive. We have revised and combined these two sentences to improve conciseness and flow. The revised sentence now reads:

- Original sentence:

"Mesoscale eddies, one of the most prominent processes in the ocean, typically range from several tens to hundreds of kilometers in spatial scale and can persist for periods of a few weeks to several years (Chelton et al., 2011a; Chaigneau et al.,2008). Mesoscale eddies are ubiquitous in the ocean and play a crucial role in the transport of heat, salinity, and other various variables, further influencing on the marine material cycle, large-scale water body transport, and biological activities (Chen et al., 2012; Chen et al., 2021; Faghmous et al., 2015; Dong et al., 2014)."

- Revised sentence:

"Mesoscale eddies, one of the most prominent processes in the ocean, typically range from several tens to hundreds of kilometers in spatial scale and can persist for periods of a few weeks to several years (Chelton et al., 2011a; Chaigneau et al., 2008). In the North Pacific, mesoscale eddies generally persist for about 4 weeks to one year, with longer-lived structures occasionally exceeding this range, particularly

in energetic regions such as the Kuroshio Extension (Chelton et al., 2011a; Chen and Han, 2019). These eddies play a crucial role in the transport of heat, salinity, and other various variables, further influencing on the marine material cycle, large-scale water body transport, and biological activities (Chen et al., 2012; Chen et al., 2021; Faghmous et al., 2015; Dong et al., 2014). " (Please see Line 33-39)

Reference:

Chaigneau, A., Gizolme, A., and Grados, C.: Mesoscale eddies off Peru in altimeter records: Identification algorithms and eddy spatio-temporal patterns, Progress in Oceanography, 79, 106-119, https://doi.org/10.1016/j.pocean.2008.10.013, 2008.

Chelton, D. B., Gaube, P., Schlax, M. G., Early, J. J., and Samelson, R. M.: The Influence of Nonlinear Mesoscale Eddies on Near-Surface Oceanic Chlorophyll, Science, 334, 328-332, https://doi.org/10.1126/science.1208897, 2011a.

Chen, G. X., Gan, J. P., Xie, Q., Chu, X. Q., Wang, D. X., and Hou, Y. J.: Eddy heat and salt transports in the South China Sea and their seasonal modulations, Journal of Geophysical Research-Oceans, 117, https://doi.org/10.1029/2011JC007724, 2012.

Chen, G., and Han, G.: Contrasting short-lived with long-lived mesoscale eddies in the global ocean, Journal of Geophysical Research: Oceans, 124, 3149–3167, https://doi.org/10.1029/2019JC014983, 2019.

Chen, G., Yang, J., and Han, G. Y.: Eddy morphology: Egg-like shape, overall spinning, and oceanographic implications, Remote Sensing of Environment, 257, https://doi.org/10.1016/j.rse.2021.112348, 2021.

Dong, C. M., McWilliams, J. C., Liu, Y., and Chen, D. K.: Global heat and salt transports by eddy movement, Nature Communications, 5, https://doi.org/10.1038/ncomms4294, 2014.

Faghmous, J. H., Frenger, I., Yao, Y. S., Warmka, R., Lindell, A., and Kumar, V.: A daily global mesoscale ocean eddy dataset from satellite altimetry, Scientific Data, 2, https://doi.org/10.1038/sdata.2015.28, 2015.

**Comment 4:** *Lines 74-75: "This approach demonstrates greater efficiency compared to previous Lagrangian eddy identification methods and enhances the objectivity of eddy boundary identification." This seems to be the major assumption driving this analysis, however, citations for this conclusion are not provided, and the other Lagrangian methods have not yet been introduced in the introduction at this point (lines 102-125).*

**Response:** We sincerely thank you for this insightful and crucial comment. You are absolutely correct that our statement in lines 74-75 was misleading and, in fact, contradictory to our main thesis. Our intention was to highlight that while the geodesic-based methods (the "approach" we referred to) provide a more objective and theoretically rigorous framework compared to earlier heuristic Lagrangian methods

(such as FTLE/FSLE, mentioned in lines 53-54), they are not more computationally efficient.

The high computational cost of these methods is the primary limitation that prevents the generation of long-term, large-scale Black Hole Eddy (BHE) datasets, which is the central motivation for our study. The later section (lines 102-125) discusses the evolution of Lagrangian data availability, which further emphasizes why such a BHE dataset does not yet exist despite advances in data collection.

To correct this critical error and clarify our logic, we have completely rewritten the sentence in question. The revised sentence now reads:

- Original sentence:

"This approach demonstrates greater efficiency compared to previous Lagrangian eddy identification methods and enhances the objectivity of eddy boundary identification. Therefore, the process of solving BHE is complex (Karrasch and Schilling, 2020), but they still maintain superior coherence in the long-term transportation of flow fields. But due to the inherent complexity of the algorithm designed for detecting BHE, an efficient detection algorithm for such eddies has yet to be developed."

- Revised sentence:

"This enhances the objectivity of eddy boundary identification, representing a significant theoretical advance over previous heuristic Lagrangian methods. And the process of solving BHE is complex (Karrasch and Schilling, 2020), but they still maintain superior coherence in the long-term transportation of flow fields. However, the practical implementation of these algorithms remains computationally intensive." (Please see Line 88-92)

Reference:

Karrasch, D. and Schilling, N.: Fast and robust computation of coherent Lagrangian vortices on very large two-dimensional domains %J The SMAI Journal of computational mathematics, 6, 101-124, https://doi.org/10.5802/smai-jcm.63, 2020.

**Comment 5:** *Lines 107: "Lagrangian-averaged vorticity deviation (LAVD) method"; I suggest adding a citation to the original study, Haller et al. 2016 (https://doi.org/10.1017/jfm.2016.151)*

**Response:** Thank you for this very helpful suggestion and for providing the reference. The reviewer is correct that we should cite the original paper that introduced the LAVD method. We have now added the citation for Haller et al. (2016) at the appropriate location. (Please see Line 125)

Reference:

Haller, G., Hadjighasem, A., Farazmand, M., and Huhn, F.: Defining coherent vortices objectively from the vorticity, Journal of Fluid Mechanics, 795, 136-173, https://doi.org/10.1017/jfm.2016.151, 2016.

**Comment 6:** *Figure 3: Why are yellow particles drawn outside of the contours in the second and third*

*timesteps, which gives the impression of eddy leakiness rather than coherence?*

**Response:** Thank you for this important observation. You are absolutely correct that our original Figure 3 was misleading. By showing particles outside the eddy boundary at later times, it incorrectly gave the impression of eddy leakiness, which contradicts the fundamental definition of a Black Hole Eddy.

Our true intention with this figure was not to illustrate the physical coherence of a single eddy, but rather to provide a schematic for the tracking algorithm detailed in the main text, which links a sequence of independently identified BHEs over time. We realize now that our original visualization was deeply flawed and confusing.

To address this fundamental issue, we have completely redesigned Figure 3 and rewritten its caption. The revised figure now correctly illustrates the tracking process without the misleading visual of escaping particles. The new caption explicitly states that the algorithm links a BHE identified at one time step with a new candidate BHE at the next, clarifying the logic of our methodology. (Please see Line 434-435)

[Figure]

**Figure 3. Schematic of the BHEs tracking algorithm.**

**Comment 7:** *Figure 6: Are these the boundaries of eddies at age 30-day or with total 30-day lifetimes?*

**Response:** Thank you for this excellent question, which highlights an ambiguity in our original figure caption. To be precise, the boundaries shown are neither for eddies of 'age 30-day' nor for those with 'total 30-day lifetimes'. They represent the coherent material boundaries identified on a specific start date (4 June 2021) that remain isolated from their surroundings over a subsequent 30-day forward integration period.

To eliminate this confusion, we have revised the caption for Figure 6 to be more explicit. The revised caption now reads: "Figure 8. Comparison between identified Black Hole Eddies (BHEs) and Rotationally

Coherent Lagrangian Vortices (RCLVs) from Liu et al. (2022) over the North Pacific on 1 March 2019, over a detection time interval T = 30 days with the Finite-Time Lyapunov Exponent (FTLE) field superimposed. (a) Anticyclonic BHEs are outlined in red, cyclonic BHEs in blue, and RCLVs in black. (b, c) Zoomed-in views of two representative regions.". (Please see Line 504-508)

***Comment 8:*** *Figs 11 & 12: I am confused about how the features relate to each other in these panels. Are the authors arguing this is an evolution of a single eddy or two different eddies in each row? I can't tell which color corresponds to which feature due to how small the figure caption labels are.*

**Response:** Thankyou for pointing out the significant lack of clarity in our presentation of these case studies. You are correct that the original figure layout was confusing. Our intention was to present separate examples for anticyclonic and cyclonic Naked BHEs, and we realize that our original presentation was flawed and insufficient.

To resolve this, and to provide more comprehensive and unambiguous evidence, we have completely redesigned our presentation and expanded it into four separate figures. We now dedicate two figures to the anticyclonic case study and two figures to the cyclonic case study.

This new format allows us to clearly demonstrate the coherence of these 'Naked' BHEs and to explicitly show that they are not enclosed by corresponding contours in either the Eulerian (ADT) or the RCLV (LAVD) fields. Specifically:

New Figure 11 shows the evolution of the anticyclonic NBHE overlaid on ADT contours, demonstrating its absence in the Eulerian field.

New Figure 12 shows the same anticyclonic NBHE, but overlaid on LAVD contours, demonstrating its absence in the RCLV field.

New Figures 13 and 14 repeat this same detailed analysis for the cyclonic case study.

We have also enlarged the legends in all new figures to ensure they are clear and readable. Here are the revised figures and corresponding captions.  (Please see Line 600-610)

- Orignal Figure:

[Figure]

**Figure 2. Positions of particles (colored dots) inside the BHE boundary and the SLA eddy bouadry every 14 days. The ADT fields are overlaid using black contours with solid lines for positive values and dashed lines for negative values.**

[Figure]

**Figure 3. Positions of particles (colored dots) inside the BHE boundary and the RCLVs bouadry every 14 days. The background is the LAVD fields.**

● Revised Figure:

[Figure]

**Figure 13. Evolution of an anticyclonic Naked Black Hole Eddy (red particles). The background contours show the Absolute Dynamic Topography (ADT) field.**

[Figure]

**Figure 14. The same anticyclonic Naked Black Hole Eddy from Figure 11, here overlaid on Lagrangian Averaged Vorticity Deviation (LAVD) contours.**

[Figure]

**Figure 15. Evolution of a cyclonic Naked Black Hole Eddy (blue particles), overlaid on ADT contours.**

[Figure]

**Figure 16. The same cyclonic Naked Black Hole Eddy from Figure 13, here overlaid on LAVD contours.**

**Comment 9:** *Line 525: Double citation typo*

**Response:** Thank you for pointing out this typo. We have corrected the repeated citation in Line 525. (Please see Line 691)

**Comment 10:** *Lines 551-552: The assumptions made here of constant size with depth do not align with the structures previously reported for Lagrangian eddies (see Deogharia et al. 2024; https://doi.org/10.1038/s41598-024-61744-6), which should at least be discussed.*

**Response:** Thank you for this very important and constructive comment, and for bringing the recent Deogharia et al. (2024) paper to our attention. You are correct that our volume calculation is based on a simplified model that assumes a constant vertical structure, which does not capture the complex 3D nature of Lagrangian eddies reported in recent literature.

To address this, we have added sentences that explicitly acknowledges the limitations of our approach. We state that our formula is a first-order approximation, cite the work of Deogharia et al. (2024) as an example of studies showing more complex structures, and provide the rationale for using the simplified method in the context of our large-scale, altimetry-based study. We believe this addition makes the limitations of our method transparent to the reader and properly situates our findings within the evolving understanding of Lagrangian eddy structures.

The newly added sentences now read:

"We note that our volume calculation is a simplified parameterization. Recent study (Deogharia et al., 2024) has revealed the complex 3D structure of Lagrangian eddies, which our surface altimetry-based analysis cannot resolve. Therefore, we use this established method to provide a first-order volume estimate, sufficient for the climatological scope of our study." (Please see Line 729-733)

Reference:

Deogharia, R., Gupta, H., Sil, S. et al.: On the evidence of helico-spiralling recirculation within coherent

cores of eddies using Lagrangian approach. Scientific Report, 14, 11014, https://doi.org/10.1038/s41598-024-61744-6, 2024.

**Comment 11:** *Figure 18: The caption has a typo, "distribution distributions"*

**Response:** Thank you for pointing out this typo. We have corrected the repeated word in the caption for Figure 18 (now in Figure. 22). (Please see Line 758)

**Comment 12:** *Line 617: "Unlike Eulerian eddies and RCLVs, which experience significant boundary deformation and mixing with the surrounding water during motion": it is misleading to couple Eulerian eddies with RCLVs in this way, considering that the BHEs are only marginally different than RCLVs, but both RCLVs and BHEs are substantially more coherent than Eulerian eddies.*

**Response:** We sincerely thank you for this important point. The reviewer is absolutely correct that it was misleading to group Eulerian eddies and RCLVs together. We recognize that both RCLVs and BHEs are advanced Lagrangian concepts representing materially coherent vortices, and they stand in contrast to the less coherent, flow-through nature of Eulerian eddies.To correct this logical flaw, we have completely restructured the sentence. The goal of the new phrasing is to first contrast Eulerian eddies with the family of coherent Lagrangian vortices (which includes both RCLVs and BHEs), and then to highlight the specific features of BHEs that are central to our analysis.

The revised sentence now reads:

- Original sentence:

"Unlike Eulerian eddies and RCLVs, which experience significant boundary deformation and mixing with the surrounding water during motion, BHE maintains a well-defined boundary and preserves their structure without significant filamentation or mixing with surrounding waters that effectively encloses and transports the water mass within."

- Revised sentence:

"Unlike traditional Eulerian eddies, whose boundaries are not material and allow for significant exchange with surrounding waters, Lagrangian vortices such as RCLVs and BHEs are defined by materially coherent boundaries that effectively trap the water mass within. Among these Lagrangian structures, BHEs are characterized by exceptionally stable boundaries that preserve their structure without significant filamentation or mixing during their motion." (Please see Line 806-810)

---

## Author Comment (AC3)

**Responses to reviewer 3#'s comments point by point**

MS No.: essd-2025-384

Title: A Black Hole Eddy Dataset of North Pacific Ocean Based on Satellite Altimetry

Author(s): Fenglin Tian et al.

**Dear Reviewer,**

We highly appreciate the detailed and valuable comments of the referee on our manuscript entitled "A Black Hole Eddy Dataset of North Pacific Ocean Based on Satellite Altimetry (ID: essd-2025-384)". These comments are all valuable and helpful for revising and improving our paper, as well as providing important guidance for our research. In the past few days, we have referred to the comments and improved the paper.

As follows, we would like to clarify some of the points raised by the Reviewer. The original comments begin with "**Comment**" and are quoted in italicized font, the responses begin with "**Response**" in normal font, the original sentences and phrases are in blue letters, the revised sentences and phrases are in red letters, and the line number in the revised manuscript is highlighted in yellow. We appreciate the Reviewer's warm work and taking the time to review the manuscript, and we hope that the corrections will meet with approval.

Yours Sincerely,

Fenglin Tian, Yingying Zhao, Lan Qin, Shuang Long, Ge Chen

2025-9-30

**Comment:**

*The paper is describing operational methods for the identification of Lagrangian structures (Black Hole Eddies) and transport analysis. Authors are underlining the computational aspect of their methodology.*

*Vortices are analyzed by using Cauchy - Green strain tensor in two dimensions, calculating distances from grid points in four directions and introducing auxiliary points.*

*Lorentz transformation, extremal curves with the shortest possible path length or as the curves that depart as little as possible from straightness (they are "locally straight") - i.e. geodesic, are important elements for the analysis. It is assumed that the absolute velocity vanishes.*

*I must confess that I find it difficult to follow the logical development of the procedures used for the detection and spatial evolution of BHEs.*

*My first doubt lies in the authors' development of the strain tensor in two dimensions, when they later admit that the flow field is three-dimensional. The transition to polar coordinates is unclear to me. I understand that this step could make the algorithm more efficient, but no explanation is provided.*

*The impression that emerges from reading the text is that the authors have taken some interesting developments from the literature, but how they are implemented in the context of the publication is somewhat cumbersome.*

**Response:**

Thank you for your detailed review and for providing such valuable and constructive feedback on our manuscript. Your professional insights are incredibly helpful, and we believe that addressing them will significantly improve the clarity and rigor of our paper. We have carefully considered all your comments and plan to revise the manuscript accordingly.

Below are our point-by-point responses and our plan for revision:

**Comment 1:** "*I must confess that I find it difficult to follow the logical development of the procedures used for the detection and spatial evolution of BHEs.*"

**Response:** Thank you for your valuable feedback. We completely agree that the clarity of the Methods section is of utmost importance for readers to understand our work.

We recognize that while we provided a flowchart (Figure 1) in our original manuscript to outline the overall framework, it may have only presented the steps at a high level. We understand that it may not have sufficiently revealed the complex computational logic within each step, nor the explicit link between the theoretical formulas and the practical implementation. This likely contributed to the difficulty in following the logical development of our methodology.

To thoroughly address this, and in response to your comment about the implementation being

"somewhat cumbersome", we have taken a key improvement measure in the revised manuscript: we have added a detailed algorithm pseudocode (Algorithm 1) to the Appendix A section.

We believe this new pseudocode perfectly complements the existing flowchart and will greatly enhance the clarity of our methodology.

The flowchart shows what we do, while the pseudocode clearly explains how we do it. It breaks down each high-level step into a concrete, logical sequence of computational operations. The pseudocode serves as a bridge between abstract mathematical formulas and their concrete implementation. We have included comments alongside key steps in the pseudocode that directly reference the corresponding equation numbers in the text, , which outlines the numerical procedure used to extract BHEs via null geodesic calculation. The pseudocode explicitly includes the main steps performed by the PYTHON code with GPU parallelization. This addition aims to make the implementation transparent and reproducible for users. we have added the pseudocode as follows: (Please see Line 868-870)

Appendix A: GPU-Accelerated Pseudo-Code for BHE Detection Using Null Geodesics

Algorithm 1 provides a brief summary of the main steps performed by the PYTHON code for BHE detection based the GPU-accelerated null geodesics algorithm.

**Algorithm 1. BHE detection based the GPU-accelerated null geodesics algorithm.**

**Input:**

- u(x, y, t), v(x, y, t): 2D velocity field in time window $[t_0, t_1]$

- Range of stretch rate $\lambda \in [0.9, 1.1]$ with step size $\Delta\lambda$

- Grid of initial polar angle $\theta \in [0, 2\pi]$

- dt: integration time step

- grid resolution: Resolution of the grid

- domain bounds: Domain boundaries for the particle seeding

**#Step 1: Load preprocessed velocity field (u, v)**

VelocityField,velocity_field = load_velocity_data("u_field.nc", "v_field.nc");

**#Step 2: Initialize particle grid and allocate on GPU(see equations (5) and (6))**

Grid grid = initialize_uniform_grid(resolution=1/32.0);

Particle* d_particles = allocate_particles_on_GPU(grid);

**#Step 3: Integrate particle trajectories using RK4 (GPU parallelism)**

RK4_integrate<<<gridDim, blockDim>>>(velocity_field, d_particles, time_step=0.1, total_steps);

**#Step 4: Compute Cauchy-Green strain tensor on GPU (see equation (3))**

compute_CG_tensor<<<gridDim, blockDim>>>(d_particles, d_C_tensor, grid);

**#Step 5: Perform eigendecomposition for each C tensor (on GPU) (see equation (4))**

eigendecompose_C_tensor<<<gridDim, blockDim>>>(d_C_tensor, d_lambda1, d_lambda2, d_xi1, d_xi2);

**#Step 6: Extract null geodesics using ZeroSet condition (each thread for φ at fixed λ) (see equation (12))**

for λ in linspace (0.9, 1.1, 9):    #λ from 0.9 to 1.1 with step of 0.025

extract_null_geodesics<<<gridDim_lambda,blockDim_phi>>>(d_C_tensor,λ,d_phi_array,d_null_flags;

**#Step 7: Check which geodesics are closed based on φ = 2π**

check_geodesic_closure<<<gridDim_geo,blockDim>>>(d_null_flags,d_phi_diff_array,d_closed_flags);

**#Step 8: Integrate valid seed points inside tensor field to generate nested LCS(see equations (13) and (14))**

integrate_seeds_in_tensor<<<gridDim_seed,blockDim>>>(d_C_tensor,d_seeds,d_nested_curves,target_phi=2π);

**#Step 9: Transfer nested curves to CPU for clustering and BHE boundary extraction**

download_curves_to_host(d_nested_curves, nested_curves);

clusters = cluster_curves_by_centroids(nested_curves);

for cluster in clusters:

    areas = compute_area_of_each_curve(cluster);

    max_curve = find_curve_with_max_area(areas);

    save_BHE_boundary(max_curve);

**Output:**

  - A closed contour representing the boundary of Black Hole Eddy
* * *
**Comment 2:** "*My first doubt lies in the authors' development of the strain tensor in two dimensions, when they later admit that the flow field is three-dimensional.*"

**Response:** Thank you for highlighting this crucial point of ambiguity. You are correct to question the use of "three-dimensional", and we appreciate the opportunity to clarify our terminology and its context.

Our analysis is performed in two spatial dimensions (longitude and latitude) because our input data consists of satellite-derived surface geostrophic velocities. This dataset is inherently two-dimensional. The confusion arises from two specific sentences in our manuscript where our terminology was imprecise. We will correct both instances to ensure clarity and consistency. Below, we explain the issue with each sentence and provide a comparison of the original and revised text.

1. The first instance appears in our description of the algorithm's mathematical formulation. The term "three-dimensional" here refers to the mathematical state space of the solver (two spatial coordinates, x and y, and an angle variable, $\phi$), not the physical dimensions of the ocean flow. While this is a mathematically accurate description of the method from Serra and Haller (2017), we now recognize it is highly ambiguous for the reader. We will revise the sentence to clarify this context.

- Original sentence:

  "...the proposed approach formulates a unified three-dimensional initial value problem...",

- Revised sentence:

  "...formulates a unified initial value problem in a three-dimensional state space (two spatial coordinates and an angle variable), enabling fully automated and robust vortex boundary detection (Serra and Haller, 2017)" (Please see Line 257)

2. The second instance is the primary source of the confusion and is incorrectly phrased in a fluid dynamics context. Our intention was to describe a flow field that is two-dimensional in space but evolves over time. We will completely rephrase this to accurately describe the system.

- Original sentence:

  "Given the three-dimensional nature of the flow field-encompassing longitude, latitude, and time..."

- Revised sentence:

  "As particles move through the unsteady, two-dimensional flow field (defined in longitude, latitude, and time), interpolation is required to estimate their trajectories." (Please see Line 323-324)

Reference:

Serra, M. and Haller, G.: Efficient computation of null geodesics with applications to coherent vortex detection, Proceedings of the Royal Society a-Mathematical Physical and Engineering Sciences, 473, https://doi.org/10.1098/rspa.2016.0807, 2017.

**Comment 3:** *The transition to polar coordinates is unclear to me. I understand that this step could make the algorithm more efficient, but no explanation is provided.*

**Response:** Thank you for this insightful feedback. We have expanded our explanation to clarify its fundamental importance to our methodology. The approach from Serra et al. (2017), which we refer to

as the geodesic-based BHE identification algorithm, simplifies the calculation of closed null geodesics. Its key innovation is the introduction of polar coordinates, which transforms the challenge from solving a variational problem to solving a more straightforward functional problem. This simplification is what enables the efficient identification of vortex boundaries. In the revised text, we now explicitly state that this is the reason for the transition.

- Original sentence:

"... Each discrete $\lambda$ yields a corresponding geodesic calculation. In the process of calculating geodesics and extracting null geodesic contours, the original algorithm is inefficient, as it requires significant time to compute and traverse each contour individually. However, since these contours are independent of one another, we can enhance the computation process by employing a GPU parallel acceleration algorithm. In this approach, each thread corresponds to a single geodesic line, allowing for simultaneous calculations and significantly reducing processing time. Utilizing a polar coordinate system, the partial derivatives of the tensor field calculated in equation are used as inputs, the computation steps for the null geodesic field adhere to the following equation: ..."

- Revised sentence:

"...with each discrete $\lambda$ yielding a corresponding geodesic calculation. The traditional approach of calculating these geodesics and extracting null geodesic contours individually is computationally inefficient. To overcome this problem, we adopt the key innovation from Serra et al. (2017) by introducing a polar coordinate system. This transforms the challenge from solving a complex variational problem into a more straightforward functional problem. The primary advantage is that a vortex boundary-a closed loop by definition-can be rigorously identified by testing if the rotation angle,$\phi$, completes a full $2\pi$ rotation. This mathematical simplification is particularly well-suited for parallelization, as it reframes the problem as a large set of independent calculations. We leverage this structure by employing a GPU-based parallel algorithm. In this approach, each thread is assigned the task of computing a single geodesic line, allowing for thousands of simultaneous calculations and thus reducing the overall processing time. Following this parallelized approach within the polar framework, the partial derivatives of the tensor field are used as inputs, and the computation steps for the null geodesic field adhere to the following equation: ..." (Please see Line 365-377)

**Comment 4:** *The impression that emerges from reading the text is that the authors have taken some interesting developments from the literature, but how they are implemented in the context of the publication is somewhat cumbersome.*

**Response:**

Thank you for this insightful feedback. We agree that the connection between the theoretical concepts and our specific computational implementation needs to be strengthened. To address these points together and resolve the overall impression of a "cumbersome" implementation, we have significantly revised our Methods section to provide a clearer and more detailed account of our computational approach.

Our revisions include two new schematic diagrams (Fig. 2 and Fig. 3) and a detailed algorithm pseudocode (now in Appendix A), which work together to bridge the gap between theory and practice. First, to demystify the specific computational details, we have added two new figures.

Figure 2 now illustrates our grid configuration, showing how we enhance simulation accuracy by densifying the original 1/4° velocity field grid by a factor of 8 using main and auxiliary nodes for the Cauchy-Green strain tensor calculation.  (Please see Line 303-306)

[Figure]

**Figure 2. Schematic diagram of the spatial relationship between a main grid point and its surrounding auxiliary points in the Cauchy-Green strain tensor field. The black dot represents an auxiliary grid point $x_0$, $\delta x_1$ and $\delta x_2$ denotes the distance between the auxiliary grid point (green) and the main grid point (red).**

Figure 3 provides a clear schematic of our CUDA thread allocation logic, explaining how the grid and block sizes are determined based on the input grid and GPU capacity. Once the size of the initial grid and the number of threads per Block in the GPU are inputted, the size of the Block to be allocated for each Grid can be determined (Fig. 3). (Please see Line 351-353)

[Figure]

**Figure 3. Thread allocation diagram.**

Complementing these diagrams, the new pseudocode in Appendix A provides the overarching algorithmic logic (**see response1**). It breaks down abstract concepts (like the Cauchy-Green strain tensor and null geodesics) into a clear sequence of computational steps (e.g., loops, conditional statements, numerical integration). To make the connection explicit, we have included comments within the pseudocode that refer back to specific equation numbers in the text.